# Impacts of reductions in non-methane short-lived climate forcers on future climate extremes and the resulting population exposure risks in eastern and southern Asia

Yingfang Li[1], Zhili Wang[1*], Yadong Lei[1], Huizheng Che[1], Xiaoye Zhang[1]

[1]State Key Laboratory of Severe Weather and Key Laboratory of Atmospheric Chemistry of CMA, Chinese Academy of Meteorological Sciences, Beijing, 100081, China

*Correspondence to*: Zhili Wang (wangzl@cma.gov.cn)

**Abstract.** Non-methane short-lived climate forcers (SLCFs), including aerosols, ozone, and their precursors, are important climate forcings and primary air pollutants. Future stringent SLCF emissions controls to mitigate air pollution will substantially impact regional climate change. Here, we investigate the changes in future climate extremes and resulting population exposure risks in Asia during 2031–2050 in response to non-methane SLCF emissions reductions using multi-model ensemble (MME) simulations under two shared socioeconomic pathway (SSP) scenarios (SSP3-7.0 and SSP3-7.0-lowNTCF) with different air quality control measures from the Aerosol and Chemistry Model Intercomparison Project (AerChemMIP), which is endorsed by Coupled Model Intercomparison Project phase 6 (CMIP6). The MME results show that future reductions in non-methane SLCF emissions lead to an increase of $0.23 \pm 0.16$ W m$^{-2}$ in global annual mean effective radiative forcing, thereby magnifying the greenhouse gas (GHG)-induced global surface warming by $0.19 \pm 0.1$ K during 2031–2050. In terms of the entire study area of Asia, the additional warming caused by the non-methane SLCF reductions increases the hottest days (TXx) by $0.3 \pm 0.1$ K, the percentage of warm days (TX90p) by $4.8 \pm 2.2$ %, the number of tropical nights (TR) by $1.7 \pm 0.8$ days, the warm spell duration (WSDI) by $1.0 \pm 0.4$ days, the number of heavy precipitation days (R10) by $1.0 \pm 0.5$ days, the maximum consecutive 5-day precipitation (RX5day) by $1.0 \pm 0.3$ mm, and the total wet-day precipitation (R95p) by $16.4 \pm 7.3$ mm during 2031–2050. In terms of sub-regional changes in temperature extremes, the largest regional increases of TXx, TX90p, and WSDI occur in northern India (NIN) and northern China (NC). Relatively large increases in TR are projected in NC and the Sichuan Basin (SCB), reaching $5.1 \pm 2.5$ days and $4.9 \pm 3.3$ days, respectively. For precipitation extremes, the regional changes are greatest in southern China (SC), particularly southwestern China (SWC), where reductions of non-methane SLCF emissions increases R10 by $2.5 \pm 1.9$ days, RX5day by $2.5 \pm 1.5$ mm, and R95p by $37.5 \pm 22.6$ mm. Moreover, the populations exposed to temperature and precipitation extremes increase most sharply in NIN, reaching $(32.2 \pm 11.4) \times 10^{7}$ person-days and $(4.6 \pm 6.1) \times 10^{6}$ person-days during 2031–2050, respectively, followed by NC and SCB. Our results highlight the significant impacts of non-methane SLCF reductions on future climate extremes and related exposure risks in eastern and southern Asia, which are comparable to the impact associated with increased GHG forcing in some regions.

## 1 Introduction

Short-lived climate forcers (SLCFs), also known as near-term climate forcers (NTCFs), include aerosols (e.g., sulfates, black carbon (BC), organic carbon (OC), ammonium salts, and nitrate), ozone, methane ($CH_4$), and their precursors (e.g., sulfur dioxide ($SO_2$), nitrogen oxides (NOx), carbon monoxide (CO), and ammonia) (Myhre et al., 2013). Since the beginning of the industrial era, human activities have led to a significant increase in emissions of SLCFs (Hoesly et al., 2018). Non-methane SLCFs, including aerosols, ozone, and their precursors, affect not only climate but also air quality. Previous studies have revealed significant influences of non-methane SLCFs on global and regional climate as well as human health (Liao et al., 2015; Forster et al., 2021; Apte et al., 2015; Malley et al., 2017; Xie et al., 2016a).

Increasing and accumulating GHGs emissions from human activities since the industrial revolution are the main drivers of global warming, but care must be taken in attributing an acceleration in human-induced global warming to these changes in SLCFs emissions (Jenkins et al., 2022). According to the Sixth Assessment Report of the Intergovernmental Panel on Climate Change (IPCC, AR6), global average effective radiative forcing (ERF) since the beginning of the industrial revolution (1750–2019) is –1.1 (–1.7 to –0.4) W m$^{-2}$ for aerosols and +0.47 (+0.24 to +0.71) W m$^{-2}$ for ozone (Forster et al., 2021). Over the past few decades, climate change caused by non-methane SLCFs has had an important impact on natural ecosystems and human societies in Asia (Ramanathan et al., 2007; Li et al., 2016). North eastern of India witnessed cooler maximum temperatures due to increased aerosols (Freychet et al. 2019). Surface cooling and tropospheric atmospheric thermal responses associated with aerosol forcing can weaken the East Asian summer monsoon, thereby suppressing precipitation in the East Asian monsoon region (Li et al., 2016; Xie et al., 2016b; Wang et al., 2019, 2020; Mu and Wang, 2021). The increase in aerosols since the 1950s may have led to weakening of summer precipitation in South Asia (Bollasina et al., 2011) and the anomalous precipitation pattern described as "southern flood/northern drought" in eastern China (Li et al., 2016). Regional aerosol-cloud interactions may be the main factor causing extreme precipitation variability in India and China during 1979–2005 (Lin et al., 2018). In terms of ozone, Chang et al. (2009) noted that tropospheric ozone increased average surface air temperature (SAT) by 0.43 °C and decreased average precipitation by 0.08 mm day$^{-1}$ in eastern China during 1971–2000. The simulation results of Li et al. (2018) showed that East Asian summer monsoon circulation was enhanced over southern China but weakened over northern China during 2001–2010 due to tropospheric ozone forcing.

Emissions controls implemented by governments, which aim to mitigate global warming and improve air quality, have led to significant changes in non-methane SLCF emissions that will affect future climate change (Naik et a., 2021). Meanwhile, the important role of non-methane SLCFs in historical climate change in Asia (Bollasina et al., 2011; Dong et al., 2019; Wang et al., 2019) suggests that SLCF emissions reductions may have major implications for future regional climate and its extremes (Wang et al., 2016; Zhang et al., 2018; Wilcox et al., 2020). In previous research, Turnock et el. (2020) made a first assessment of historical and future changes in air pollutants and Zanis et el. (2022) presented an analysis of the effect of climate change on surface ozone discussing the related penalties and benefits around the globe using five Earth system models (ESMs) from the Coupled Model Intercomparison Project phase 6 (CMIP6). But the assessed impact of reducing non-methane SLCF

emissions on future climate change has been limited to the effect of aerosol forcing associated with incomplete interactive tropospheric chemistry schemes in global climate models. Future aerosol emissions reductions could greatly aggravate the

warming effect caused by greenhouse gas (GHG) forcing (Lin et al., 2016; Hienola et al., 2018), particularly in eastern and southern Asia, and may increase surface temperatures by 0.5°C in some regions (Westervelt et al., 2015; Chuwah et al., 2016) and cause a significant increase in extreme high-temperature events (Wang et al., 2016; Samset et al., 2018; Luo et al., 2020). In eastern Asia, under global warming of 1.5 °C and 2 °C, the average temperature in China is expected to grow at a faster rate than the global mean, and there is a strong warming in the Tibetan Plateau (You et al. 2020). When studying changes in local

climate between 1.5 °C and 2 °C of global warming, non-GHGs influences need to be considered (King et al. 2018; You et al. 2020). The effect of projected reductions in anthropogenic aerosol emissions over eastern Asia caused an increase in summer temperatures and raised the likelihood of extreme hot summers (King et al. 2018). Future aerosol reductions may also shift the tropical rain belt northwards, enhancing precipitation in the Asian monsoon region (Westervelt et al., 2018; Zanis et al., 2020) and shifting both the mean and extreme values of precipitation toward larger values (Zhao et al., 2018). Asia is a region where

extreme precipitation is particularly sensitive to aerosol reductions (Lin et al., 2016; Samset et al., 2018).

The precursors of aerosols and ozone are homologous and overlapping (e.g., NOx, volatile organic compounds (VOCs)) (Myhre et al., 2013). Implementation of air pollution control measures inevitably causes simultaneous changes in their radiative forcings. Therefore, considering the impacts of changes in non-methane SLCFs on future climate, particularly on regional extreme climate events, is essential. However, few studies in this field have been conducted to date. A recent study considering

emissions reductions of aerosols, ozone, and their precursors showed that future non-methane SLCF reductions will not only improve air quality but also increase global mean temperature, precipitation, and climate extremes in the mid-21st century, with more marked warming and wetting trends in some regions, particularly Asia and the Arctic (Allen et al., 2020). However, there were some limitations in that study. First, Allen et al. (2020) only considered three extreme indicators including hottest day, wettest day, and consecutive dry days to examine the effects of future non-methane SLCF reductions on climate extremes.

These three indicators were not enough to represent climate extremes, especially the lack of some indicators related to human health. For example, tropical night usually occurs in combination with extended periods of heat (particularly in extra-tropical regions) and have been suggested to be problematic for human health (Weisskopf et al., 2002; Patz et al., 2005) and the maximum consecutive 5-day precipitation can be used as an indicator of flooding and related hazards (Frich et al., 2002; Sillmann et al., 2013). Second, climate extremes pose a serious threat to human health and well-being (Bras et al., 2021;

Tellman et al., 2021). Quantifying avoided population exposure to climate extremes associated with future non-methane SLCF reductions is valuable for future policymaking on climate change mitigation and adaptation, especially in these densely populated and industrially developed regions of Asia, which is lacking in Allen et al. (2020).

With the continued global warming caused by GHG emissions, the frequency of climate extremes increases (Zhou and Qian, 2019) will increasingly become a threat to human health, economic stability, and environmental sustainability. As a

densely populated and climate-vulnerable region, Asia is at high risk for future population exposure to climate extremes. In the coming decades, the population exposed to extreme heat in East Asia will continue to increase under most future scenarios

(An et al., 2020). The population exposure to extreme precipitation over the Indus River Basin is projected to increase by 72.4%, 122.7%, and 87.6%, respectively, with global warming of 1.5°C, 2.0°C, and 3.0°C (Zhao et al., 2021). The population affected by extreme precipitation in China is projected to increase by nearly 22% by the end of the 21st century under the representative concentration pathway–shared socioeconomic pathway (RCP-SSP) scenario RCP4.5-SSP2, with the largest absolute increase in eastern China (Chen and Sun, 2020). With increasing warming in the future, dry conditions across China will be further aggravated, and an additional 12.9 million people will be exposed to droughts under the 1.5°C global warming scenario (Chen et al., 2018; Chen and Sun, 2019). This raises the issue of whether reductions in non-methane SLCF emissions will further exacerbate the intensity and frequency of extreme climate events and the resulting risk of population exposure.

With continuous development, some models have evolved from the original climate system model to more complete ESMs that considers the coupling of Earth system processes (including atmosphere, ocean, land, sea and ice, and biogeochemical cycles) and can provide a more realistic and complete analysis of the whole Earth system. Furthermore, the Aerosols and Chemistry Intercomparison Project (AerChemMIP) was established by the CMIP6 to comprehensively assess the impacts of SLCFs on climate and air quality (Collins et al., 2017), which can further elucidate changes in atmospheric chemical composition and their interactions with changes in global and regional climate. Here, we quantify the impacts of non-methane SLCF emissions reductions on extreme climate changes and the resulting population exposure risks during 2031–2050 in eastern and southern Asia (here defined as 0–60°N, 70–150°E) using historical simulations and two scenario simulations (SSP3-7.0 and SSP3-7.0-lowNTCF) with state-of-the-art ESMs from the AerChemMIP (Collins et al., 2017).

The remainder of this paper is organized as follows. Section 2 describes the models, simulations, and calculations of climate extremes and related population exposure. Section 3 contains the results. Finally, our discussion and conclusions are presented in Section 4.

## 2 Methods

### 2.1 Future emissions scenarios

Simulations under two AerChemMIP scenarios, SSP3-7.0 and SSP3-7.0-lowNTCF, were performed to assess the impacts of non-methane SLCF changes on climate and air quality (Collins et al., 2017; Allen et al., 2020). The SSP3-7.0 scenario includes no climate policies related to GHG reductions and weak air pollution controls ($\sim$7.0 W m$^{-2}$ at 2100). In this scenario, the future climate change is approximatively influenced by GHG forcing only. TheSSP3-7.0-lowNTCF scenario uses the same GHG forcing path but with strong air pollution controls, and thus future climate change in this scenario is influenced by a combination of GHG forcing and reduction of non-methane SLCF emissions. As shown in Figure 1, the two simulations include the same variation in carbon dioxide ($CO_2$) and $CH_4$, both of which continue to increase in the future. However, non-methane SLCFs exhibit different changes in these two scenarios. Global emissions of non-methane SLCFs are maintained at a stable level or vary slightly after 2020 under the SSP3-7.0 scenario, with a decrease of 1% in $SO_2$ and increases of 7–13% in other SLFCs by 2050. Global emissions of all non-methane SLCFs show downward trends under the SSP3-7.0-lowNTCF

scenario, ranging from 10% for BC to 55% for $SO_2$ by 2050. Assuming that the joint impact of GHGs and non-methane SLCFs

are approximated as a linear superposition of their respective impacts, the impact of non-methane SLCF mitigation can be

quantified as the difference between the results of simulation under the two scenarios (SSP3-7.0-lowNTCF minus SSP3-7.0)

(Allen et al., 2020).

## 2.2 Models and simulations

Seven CMIP6 ESMs with interactive representation of tropospheric aerosols and atmospheric chemistry were used to

135 conduct the SSP3-7.0 and SSP3-7.0-lowNTCF simulations (Table 1), with integration periods of 2015–2055 and individual

models integrated through 2100. We used the daily and monthly outputs for both scenarios from 2015 to 2050. Historical

climate simulations (Historical) for 1995–2014 from the same models were used as the reference period. In addition to coupled

ocean-atmosphere simulations, these models were also used for Atmospheric Model Intercomparison Project (AMIP)-style

simulations with the time-varying sea surface temperature (SST) outputs from SSP3-7.0 simulations under the two emissions

scenarios to quantify the ERF associated with future changes in atmospheric composition (Collins et al., 2017). All models

used here provided SST simulations for calculating ERF and they provided the same number of realizations as stated in Table

1. ERF is a measure of the extent to which forcing agents affect climate, defined as the change in net radiative flux at the top

of the atmosphere caused by a disturbance that allows for changes in atmospheric temperature, water vapor, and clouds, but

keeps global SST unchanged (Myhre et al., 2013; Forster et al., 2016; Pincus et al., 2016). At present, there are few models

carrying out both SSP3-7.0 and SSP3-7.0-lowNTCF experiments as mentioned above, thus, we directly used the seven-model

ensemble mean to investigate the changes of climate extremes in response to future SLCFs emission reductions.

To evaluate the performance of the models, a gridded daily maximum and minimum temperature and daily precipitation

dataset obtained from the National Oceanic and Atmospheric Administration (NOAA) Climate Prediction Center (CPC) is

used here. This dataset was constructed using optimal interpolation methods based on approximately 16,000 station and

150 satellite observations (Chen et al., 2020b). It spans the period from 1979 to the present with a resolution of $0.5° \times 0.5°$. All

model outputs as well as observations were interpolated into a common grid ($1° \times 1°$) through bilinear interpolation except

precipitation data, which used first-order conservative interpolation. We also quantify the confidence of the MME relative to

each individual model using sign agreement and Signal-to-noise ratio ($S_{NR}$). The details of the calculation of $S_{NR}$ can be found

in Supplementary method. We compared the simulated results with the observational climate extremes during 1995–2014. In

general, the multi-model ensemble (MME) can reasonably reproduce the observed spatial distribution of extreme temperature

and precipitation indices. For the extreme temperature indices, the maximums obtained from both the MME and observations

are found in eastern China and southern Asia, especially for the simulated absolute extreme indices (the hottest day, tropical

night) (Figs. 2a, 2b, 2g and 2h), which are generally consistent with the observations in spatial distribution with limited

difference in magnitude. Relative to the absolute extreme indices, the percentile and duration indices show large differences

between the MME and observation (Figs. 2d, 2e, 2j and 2k). Previous studies also shown that both CMIP5 and CMIP6 models

perform relatively unsatisfactorily in simulating spatial patterns of the duration and percentile indices (Fan et al., 2020; Guo

et al., 2021). For heavy precipitation, the maximum consecutive 5-day precipitation and total wet-day precipitation, the climatological mean is well captured by MME, although it tends to produce overestimates especially over southeastern Qinghai-Tibet Plateau and the Indo-China Peninsula (Figs. 3a-i). In addition, the MME underestimates the consecutive dry days in northwest China and along Mongolia (Fig.3l), which is consistent with previous studies (Zhu et al., 2021; Kim et al., 2020). Although the MME produce some regional biases with respect to observation, such biases will be reduced when considering the difference between the two segments of time (Sillmann et al., 2013; Chen et al., 2020). In this study, we focused on the changes in the future (2031–2050) relative to the reference period (1995–2014), so the results of the MME for future climate extreme projections can be considered representative.

## 2.3 Climate extremes indices

Following the definitions of the Expert Team for Climate Change Detection and Indices, six extreme temperature indices and four extreme precipitation indices representing intensity and frequency were employed in this study (Table 2). Due to incomplete data provided by CESM2-WACCM and GISS-E2-1-G in the AerChemMIP, the extreme temperature indices were calculated with the remaining five models, and the extreme precipitation indices were calculated with the six models other than GISS-E2-1-G. We focused on the changes in extreme climate indices in the future (2031–2050) relative to the reference period (1995–2014), as this is when reductions in aerosol and ozone precursor emissions and the resulting changes in climate are expected to be significant.

## 2.4 Population exposure

Population exposure to climate extremes can be estimated by multiplying the frequency of extreme events by the number of people, and this method has been widely employed to assess future climate risks (Jones et al., 2015). Gridded population datasets for 2000 and 2040 under SSP3 were used to represent the population during the reference and future periods, respectively (Jones and Oneill 2016). These population datasets were summed for each grid box of $1° \times 1°$ to match the resolution of the climate data.

Population exposures to extreme heat and heavy precipitation are calculated as the population exposed multiplied by the number of days with a daily maximum temperature greater than the 90th percentile of the reference period and by the number of days with daily precipitation greater than the 95th percentile of the reference period, respectively (Chen et al., 2020; Sun et al., 2022). The exposure change (Δexpo) could be decomposed into three parts, namely, changes driven by the climate effect, the population effect, and their interaction:

$$\Delta expo = (c + \Delta c) \cdot (p + \Delta p) - c \cdot p = c \cdot \Delta p + p \cdot \Delta c + \Delta c \cdot \Delta p , \tag{1}$$

where c and p are the number of extreme days and population in the baseline period, respectively. Δc and Δp are the changes in climate extreme days and population in the future with respect to the baseline period. Thus, the terms of c·Δp, p·Δc,

and $\Delta c \cdot \Delta p$ represent the population, climate, and population-climate interaction effects, respectively, on exposure changes, allowing for attribution analysis of changes in population exposure.

## 3 Results

**3.1 Changes in global ERF and SAT**

Figure 4 shows the spatial patterns of ERF under the SSP3-7.0 and SSP3-7.0-lowNTCF scenarios and their changes associated with reductions in non-methane SLCF emissions during 2031–2050 relative to 1995–2014. As GHG concentrations increase and SLCF concentrations decrease, more long-wave radiation is absorbed and less short-wave radiation is reflected, resulting in positive global average ERF values under both scenarios (Figs. 4a and b). It is worth noting that the global spatial

distribution of ERFs shows poor inter-model agreement, with large areas where there is disagreement of the sign with at least one of the models and $S_{NR}$ less than 1 (Supplementary methods and Fig. S2), especially over the ocean. Non-methane SLCF emissions reductions lead to positive ERF over most of the globe, particularly in the Northern Hemisphere (Fig. 2c). Increases in ERF of approximately 1 W m$^{-2}$ occur mainly in Eurasia north of 40°N, the northern Pacific Ocean, northern America, and the northern Atlantic Ocean. The largest increases of more than 1.5 W m$^{-2}$ are found over India, southeastern Asia, and

surrounding oceans with relatively good inter-model agreement (Fig. 4c and S2c). Although the reduction of SLCFs leads to a decrease in the aerosol optical depth (AOD) on a global scale (Fig. S1), not all SLCFs behave the same way and it is reductions in non-BC aerosols that are driving the increase in ERF in the SSP3-7.0-lowNTCF scenario. Besides, some negative ERFs remain over northern China, the Sichuan Basin, Arabian Peninsula, and northern Africa, as the increase in low cloud cover induced by changes in regional precursor concentrations leads to greater reflection of shortwave radiation (Yang et al.,

2017; Zhang et al., 2018). The multi-model ensemble (MME) global annual mean ERF induced by non-methane SLCF emissions reductions is +0.23 W m$^{-2}$, with positive values obtained from all models except GFDL-ESM4 (Table 3).

Figure 5 shows the MME global changes in SAT under each scenario. Increases in SATs occur worldwide during 2031–2050 under both the SSP3-7.0 and SSP3-7.0-lowNTCF scenarios (Fig. 5a and b). The SAT increases are greater in the Northern Hemisphere and on land than in the Southern Hemisphere and ocean areas due to greater precursor emissions in the Northern

Hemisphere and the high heat capacity of the ocean. Regionally, the greatest warming of more than 1.5 K is found in central and northern Asia and in northern North America, particularly in the Arctic, where warming is greater than 2.5 K. The models agree on the sign of the change across regions and have high level of confidence (Fig. 5a, b and Fig. S3a, b). Previous studies have also shown that the Arctic warms at a much higher rate than the rest of the globe in recent decades (Screen and Simmonds, 2010; Pithan and Mauritsen, 2014). Moreover, such warming is exacerbated by future non-methane SLCF emissions reductions

in most regions of the world, particularly in the Northern Hemisphere with great inter-model consistency (Fig. 5c and 3c). Previous studies have shown that reduction of sulfate aerosols is the dominant driver of future SAT increases in response to aerosol forcing, particularly in the Northern Hemisphere (Baker et al., 2015; Samset et al., 2018). Future reductions in non-methane SLCF emissions causes additional warming of more than 0.2 K throughout Eurasia, the northern Pacific, and northern

North America. In particular, warming in the Arctic will exceed 0.6 K (Fig. 5c). The reduction of non-methane SLCF emissions results in an average increase of 0.19 K in global mean SAT in the MME results, ranging from 0.06 K to 0.29 K across different models (Table 3). Furthermore, the increase in global mean SAT is consistent with the increase in global mean ERF except for GFDL-ESM4. However, warming and positive ERF do not correspond exactly in some regions, such as the Arctic and northern China (Figs. 4c and 5c), likely due to climate feedbacks, remote teleconnection, and other processes. In general, under the SSP3-7.0-lowNTCF scenario, the combined reductions of aerosols, ozone, and their precursors causes further warming, suggesting that the cooling effect of ozone reduction is somewhat weaker than the warming effect of aerosol reduction.

## 3.2 Changes in temperature extremes in eastern and southern Asia

Figure 6 shows the time series of changes in annual mean extreme temperature indices averaged across the entire study area under the SSP3-7.0 and SSP3-7.0-lowNTCF scenarios from 2015 to 2050 relative to the reference period. All extreme temperature indices in the entire study area consistently increase under both scenarios, with a larger increase under the SSP3-7.0-lowNTCF scenario. By 2050, the hottest day and warm days increase by 2.2 ± 0.7 K (mean ± standard deviation) and 27.7 ± 13.1 % under the SSP3-7.0 scenario and by 2.6 ± 0.6 K and 34.9 ± 13.3 % under the SSP3-7.0-lowNTCF scenario, respectively. Compared to the hottest day and warm days, larger increases occur in the coldest day and warm nights, which increase by 2.8 ± 0.8 K and 35.9 ± 14.9 % under the SSP3-7.0 scenario and 3.2 ± 1.0 K and 43.1 ± 15.6 % under the SSP3-7.0-lowNTCF scenario, respectively. These changes indicate that future nighttime warming will be more apparent than daytime warming under both scenarios. Compared to the SSP3-7.0 scenario, non-methane SLCF emissions reductions result in varying degrees of increase in the temperature extremes indices, and such increases will become more pronounced as future time passes.

Figure 7 shows the spatial patterns of changes in the hottest day, warm days, tropical night, and warm spell duration over the entire study area during 2031–2050 relative to the reference period. Consistent increases occur in these extreme temperature indices under both the SSP3-7.0 and SSP3-7.0-lowNTCF scenarios. For the hottest day, in the SSP3-7.0 and SSP3-7.0-lowNTCF scenarios, the warming in most regions exceeds 1.5 K, and the warming is greater at higher latitudes under both scenarios with all models agree on the sign and $S_{NR}$ larger than 1 (Fig 7a, b and Fig. S4a, b). Compared to the SSP3-7.0 scenario, non-methane SLCF reduction leads to an additional increase of about 0.2 K across most of the entire study area's continent. Local effects of non-methane SLCF reductions are particularly large due to the short lifetimes of these contaminants. For example, future reductions in non-methane SLCFs cause the hottest day increases of more than 0.4 K in northern India and northern China with great inter-model consistency, where anthropogenic emissions are high (Fig. 7c).

Similarly, warm days increases under both scenarios (Figs. 7d and e). In contrast to the changes in SAT (Fig. 5) and the hottest day, the increase in warm days is more pronounced (>40%) at lower latitudes. This difference arises because lower daily temperature variation at low latitudes could result in more days above the 90th percentile of the reference period. The largest increases in warm days with non-methane SLCF reductions are found over northern India and eastern China with relatively high agreement among models and confidence level (Figs. 7f and Figs S4f), where increases are more than 8% and around 6%, respectively (Fig. 7f). In these regions, the effects of non-methane SLCFs are comparable to those of GHG forcing

(Fig. 7d and f). The hottest day and warm days indicate an increase in extreme daytime heat events, which not only increases the risk for heatstroke but also exacerbates ozone pollution, posing a further threat to human health (Gosling et al., 2009; Pu et al., 2017).

Concomitant with the increases in the coldest day and warm night (Fig 6), tropical night is also projected to increase over most study area (red areas in Fig. 7g and h) in the future with some difference among models, but the agreement among models is relatively good in regions where tropical night increases, such as eastern China, western northern China, southern Asia and southern eastern Asia. An increase of more than 40 days occurs in parts of southern India, the Indo-China Peninsula, and Indonesia, followed by eastern and northern China and Mongolia, with increases greater than 30 days, under the SSP3-7.0 scenario (Fig. 7g). The spatial pattern of tropical night changes under SSP3-7.0-lowNTCF is similar to that under SSP3-7.0, but the magnitude of the changes is larger, especially in eastern China, western northern China, and Indo-China Peninsula where models show good agreement (Fig. 7g, h and Fig. S4g, h). Non-methane SLCF reductions cause robust increases in tropical night of more than 5 days in the Indo-China Peninsula, northern and northeastern China, and the Korean Peninsula (Fig. 7i). Notably, tropical night indicates that the temperature at night remains above 20°C. High nighttime temperatures can cause insomnia and abnormalities in thermoregulation, which may increase health risks, particularly for elderly or sick individuals (Fischer and Schär, 2010; Gosling et al., 2009).

With the increase in average temperature and values of extreme temperature indices described above, warm spell duration is also projected to increase across the entire study area under both scenarios (Fig. 7j and k). Similar to warm days, the increase in warm spell duration is more apparent at lower latitudes. The largest increments occur over the Qinghai-Tibet Plateau, southern China, and the Indo-China Peninsula. Compared to the SSP3-7.0 scenario, the increase in warm spell duration is greater under the SSP3-7.0-lowNTCF scenario, with increases of more than 12 days in the Qinghai-Tibetan Plateau, the Indo-China Peninsula, and along the coast of southern China. Non-methane SLCF reductions are associated with increases in warm spell duration of more than 2 days over most study area and more than 4 days in parts of northern China, coastal areas of southern China, northern India, and central and northern Asia (Fig. 7l). Notably, the confidence level of the MME in these regions is relatively high, except for some small areas (Fig. S4l), but the variation in sign among models is large (Fig. 7l).

Overall, although ozone reductions cause cooling, the combined reductions of aerosols, ozone, and their precursors have an additional warming effect on Earth's climate system, exacerbating the surface warming and extreme temperature events caused by GHGs. These results are consistent with studies considering only aerosol reductions (Wang et al., 2016; Samset et al., 2018). Spatially, non-methane SLCF reductions cause the largest increases in extreme temperature indices in northern India and eastern China with good agreement among models and high level of confidence (Fig. S4). In these regions, the human and environmental effects of temperature extremes are more significant due to high population densities and industrial development. Therefore, we selected northern India (NIN), northern China (NC), southern China (SC), and the Sichuan Basin (SCB) (Fig. 8a) for quantitative assessment of changes in regional mean extreme temperature indices in response to non-methane SLCF reductions.

The domain-averaged values of the hottest day, warm days, tropical night, and warm spell duration across the entire study area have increases of $1.4 \pm 0.4$ K, $37.4 \pm 11.3$ %, $8.9 \pm 2.3$ days, and $9.1 \pm 2.0$ days under the SSP3-7.0 scenario, respectively (Fig. 8). Considering the effects of non-methane SLCF reductions, these extreme temperature indices show increases of $1.7 \pm 0.4$ K, $42.2 \pm 11.0$ %, $10.6 \pm 2.8$ days, and $10.1 \pm 1.7$ days under the SSP3-7.0-lowNTCF scenario, respectively. Future reductions in non-methane SLCFs result in additional increases of $0.3 \pm 0.1$ K in the hottest day, $4.8 \pm 2.2$ % in warm days, $1.7 \pm 0.8$ days in tropical night, and $1.0 \pm 0.4$ days in warm spell duration for the entire study area. The hottest day, warm days, and warm spell duration all show their largest regional increases in NIN and NC due to non-methane SLCFs reduction, reaching $0.5 \pm 0.2$ K, $8.5 \pm 2.9$ %, and $2.8 \pm 1.1$ days and $0.4 \pm 0.2$ K, $5.9 \pm 2.7$ %, and $3.0 \pm 1.3$ days, respectively. The models agree on the sign of the change in tropical night across regions, and the largest increases occur in NC and the SCB, at $5.1 \pm 2.5$ and $4.9 \pm 3.3$ days, respectively. Although the MME changes in regional mean extreme temperature indices are consistently positive, some differences are found among the models, which may be attributed to differences in model resolution, the types of SLCFs included in each model, and parameterization schemes (Allen et al., 2020; Wei et al., 2019).

## 3.3 Changes in precipitation extremes in eastern and southern Asia

Figure 9 shows the time series of changes in annual mean precipitation indices averaged across the entire study area under the SSP3-7.0 and SSP3-7.0-lowNTCF scenarios from 2015 to 2050 relative to the reference period. Similar to the extreme temperature changes described above, the extreme precipitation changes are larger under the SSP3-7.0-lowNTCF scenario than the SSP3-7.0 scenario, and this difference increases gradually after 2030. However, precipitation projections have larger uncertainties and disagreement across models. By 2050, the values of heavy precipitation days, the maximum consecutive 5-day precipitation, and total wet-day precipitation have increased by $1.6 \pm 0.6$ days, $2.7 \pm 1.0$ days, and $46.7 \pm 16.4$ mm under the SSP3-7.0 scenario, respectively. The increases in precipitation decrease drought events, resulting in a reduction in maximum consecutive dry days of $2.4 \pm 1.5$ days. Non-methane SLCFs mitigation causes an increase in extreme precipitation indices after 2035 except consecutive dry day, which shows a complex signal in the future.

Figure 10 shows the spatial patterns of extreme precipitation indices over the entire study area. Considering the spatial distribution of the changes in extreme precipitation indices, we selected NIN, NC, SC, and southwestern China (SWC) (Fig. 11a) for quantitative analysis of extreme precipitation changes. Heavy precipitation days, the maximum consecutive 5-day precipitation, and total wet-day precipitation represent future changes in the intensity and frequency of extreme heavy precipitation. Under the SSP3-7.0 scenario, heavy precipitation days decreases by approximately 2 days in central India, the Indo-China Peninsula, and by more than 4 days in the southeastern Qinghai-Tibet Plateau, SWC, and parts of Indonesia, while it increases in all other regions of the entire study area (Fig. 10a). Under the SSP3-7.0-lowNTCF scenario, heavy precipitation days increases by $1.5 \pm 1.0$ days across the entire study area (Fig. 11b), with the largest increases in western China and near the Himalayas. However, it is worth noting that the consistency among models is poor (Fig. 10b and Fig. S5b). The reductions in non-methane SLCFs cause an increase of $1.0 \pm 0.5$ days in average heavy precipitation days across the entire study area (Fig. 11b). For the selected regions, robust increases in heavy precipitation days occur in SWC, SC, reaching $2.5 \pm 1.9$ days

and 1.5 ± 0.8 days, respectively (Fig. 10c and Fig. 11b). In addition, increases of more than 3.0 days in heavy precipitation days are also found in India, the Indo-China Peninsula, and Indonesia, but again, consistency in these regions among models is poor, and this pattern is exactly opposite of that of GHG forcing (Fig. 10a and c). By contrast, previous research suggested that reduction of aerosols alone will reduce R10 in these regions during 2031–2050 (Wang et al., 2016; Zhao et al., 2018).

The maximum consecutive 5-day precipitation increases across most of eastern and southern Asia under both future scenarios (Figs. 10d and e), with average increases of 1.7 ± 0.9 mm and 2.7 ± 1.0 mm under the SSP3-7.0 and SSP3-7.0-lowNTCF scenarios, respectively (Fig. 11c). The largest increases among the selected regions occur in SC, at 2.9 ± 2.4 mm and 4.5 ± 2.8 mm, respectively (Fig. 11c). Future reductions in non-methane SLCFs leads to an increase of 1.0 ± 0.3 mm in the maximum consecutive 5-day precipitation in the entire study area (Fig. 11c). Regionally, these effects of non-methane SLCF reductions are much greater than the effects of GHG forcings alone. For example, non-methane SLCF mitigation yields an increase of 2.5 ± 1.5 mm in the maximum consecutive 5-day precipitation in SWC, compared to 0.5 ± 2.1 mm under the SSP3-7.0 scenario (Fig. 11c). In addition, non-methane SLCF reductions cause an increase of more than 2 mm in the maximum consecutive 5-day precipitation in the southeastern Qinghai-Tibet Plateau, the western Hengduan Mountains, SWC and the middle and lower reaches of the Yangtze River, whereas decreases of more than 3 mm occur in the southwestern part of the Indo-China Peninsula. However, there are large differences among models (Fig. 10f and Fig. S5f). The variation in the maximum consecutive 5-day precipitation can be used as an indicator of flooding and related hazards (Frich et al., 2002; Sillmann et al., 2013). Our results suggest that heavy precipitation associated with natural disasters will be aggravated in some parts of eastern and southern Asia in the future due to non-methane SLCF reductions.

The spatial pattern of changes in total wet-day precipitation is consistent with that of the maximum consecutive 5-day precipitation. With or without strong air quality controls, total wet-day precipitation increases across most of eastern and southern Asia, reaching 38.3 ± 21.5 mm and 54.7 ± 22.4 mm, respectively (Fig. 10g, h and Fig. 11d). The increases in total wet-day precipitation exceed 120 mm in the Himalayas, southeastern Qinghai-Tibet Plateau, western Hengduan Mountains, and Indonesia under the SSP3-7.0-lowNTCF scenario (Fig. 10h). Additional increases in total wet-day precipitation caused by reduced emissions of non-methane SLCFs are apparent in the southeastern Qinghai-Tibet Plateau, SWC, and the middle and lower reaches of the Yangtze River (>40 mm), while decreases occur in the southwestern part of the Indo-China Peninsula (<50 mm) with lower consistency among models. Total wet-day precipitation also decreases (<10 mm) along the coast of SC (Fig. 10i), with effects greater than or comparable to GHGs, even opposite in some regions. This finding is in opposition to previous results based on only aerosol reduction (Wang et al., 2016; Zhao et al., 2018).

In contrast to the three extreme precipitation indices described above, consecutive dry days represents the variability of extreme drought. In the entire study area north of 30°N, the increase in total precipitation reduces consecutive dry days, particularly at high latitudes. However, consecutive dry days increases in NIN, the southeastern Tibetan Plateau, and the southern Yangtze River (Fig. 10j and k) with low inter-model agreement. Overall, consecutive dry days decreases by 0.02 ± 0.8 days and 0.4 ± 0.5 days under the SSP3-7.0 and SSP3-7.0-lowNTCF scenarios, respectively (Fig. 11e), with $S_{NR}$ less than 1, which means model projections have a low level of confidence (Fig. S5j and k). Future reductions in non-methane SLCFs

contribute to decreases in consecutive dry days in northwestern and northeastern China, near the Hengduan Mountains, and the Indo-China Peninsula. Consecutive dry days decreases by $1.4 \pm 1.2$ days in SWC due to increase in the frequency and intensity of heavy precipitation (Fig. 11e). The sign of the results shows relatively good agreement and the $S_{NR}$ is larger than

1 among models in SWC and NC (Fig. 10i and Fig. S5i). Similarly, the changes in consecutive dry days due to non-methane SLCF reductions in some regions are comparable to the impacts of GHG forcing. Also, for most regions, the models do not all agree on the sign of the responses and have low level of confidence. The increases in consecutive dry days in India and eastern China are accompanied by increases in the frequency and intensity of extreme precipitation, which may be related to the probability distribution of future precipitation. A decrease in light rainfall and increase in heavy rainfall will lead to

simultaneous increases in consecutive dry days and extreme precipitation (Wang et al., 2016). Zhu et al. (2021) showed that extreme precipitation in SC increases while droughts become more severe under the SSP3-7.0 and SSP5-8.5 scenarios, suggesting that the future precipitation probability distribution function (PDF) distribution in SC could be more heterogeneous under high SSP scenarios.

### 3.4 Population exposure to climate extremes in eastern and southern Asia

Extreme weather and climate events pose serious threats to human health, economic stability, and environmental sustainability, particularly in densely populated areas. Sustained warming of the climate will exacerbate climate extremes and the associated risks (Diffenbaugh et al., 2017; Sun et al., 2021). Based on the analysis presented above, reduction of non-methane SLCFs will further increase the frequency, intensity, and duration of extreme temperature and precipitation occurrences in eastern and southern Asia. Next, we analyze the changes in population exposure to warm days and total wet-

375    day precipitation under both future scenarios (Fig. 12). It is important to emphasize that there are still uncertainties among models, especially in future projections of extreme precipitation, the difference in model simulations could also lead to uncertainties in population exposure.

Considering population growth and increased frequency of climate extremes, the total population exposed to warm days and total wet-day precipitation show similar spatial patterns under the SSP3-7.0 and SSP3-7.0-lowNTCF scenarios, with the

largest increases in India and eastern China (Fig. 12a, b, d, and e). Quantitatively, with the same population growth, a significantly greater population will be exposed to extreme temperature than to extreme heavy precipitation due to the higher frequency of extreme temperature events than extreme heavy precipitation events in the future. Under the SSP3-7.0 scenario, the largest increases in the population exposed to extreme temperature occur in India and eastern China, with a maximum increase greater than $6 \times 10^8$ person-days, while the increases in the population exposed to extreme precipitation exceed $4.5 \times$

$10^7$ person-days in India and are relatively small in eastern China ($>1.5 \times 10^7$ person-days) (Fig. 12e). Compared to the SSP3-7.0 scenario, future non-methane SLCF reduction will increase the population exposed to extreme temperature by more than $3.5 \times 10^8$ person-days in NIN, followed by more than $1.5 \times 10^8$ person-days in NC and the SCB (Fig. 12c). In addition, future non-methane SLCF reductions will significantly increase the population exposed to extreme precipitation in India, the SCB, and the middle and lower reaches of the Yangtze River, reaching values greater than $6 \times 10^6$ person-days (Fig. 12f).

Future changes in population exposure to climate extremes are affected by climate, population, and climate-population interactions according to equation (1) presented above. Under the SSP3 scenario, the population in India increases significantly by 2040 relative to 2000. This change is particularly sharp in NIN, which experiences an increase of more than 5 million people, followed by more than 1 million people in the SCB, NC and SC (data not shown). We attribute the changes in population exposure to three major factors (Figs. 13 and 14).

The increases in population exposed to extreme high temperatures are dominated by changes in climate factors under both the SSP3-7.0 and SSP3-7.0-lowNTCF scenarios in the four selected regions (Fig. 13). This result suggests that climate change caused by non-methane SLCF is the primary driver of population exposure to extreme temperature events, followed by changes in the climate-population interaction factor, with population change contributing the least. Future non-methane SLCF reductions increase total population exposure to warm days by $(32.2 \pm 11.4) \times 10^7$ person-days in NIN, $(10.3 \pm 5.2) \times 10^7$ person-days in NC, $(7.5 \pm 6.9) \times 10^7$ person-days in the SCB, and $(6.3 \pm 4.7) \times 10^7$ person-days in SC, respectively.

For population exposure under extreme precipitation, the climate factor is also the largest contributor to the increase in population exposure, followed by the climate-population interaction factor and the population factor in NC, the SCB, and SC (Fig. 14b-d). By contrast, the population factor becomes dominant in NIN, indicating that population growth is the main cause of increased population exposure in that region, followed by changes in the climate factor and climate-population interaction factor (Fig. 14a). Notably, in the SCB, greater changes in population exposure are caused by non-methane SLCF reductions than by the continued increase in GHGs (Fig. 14c). Future non-methane SLCF reductions increase total population exposure to total wet-day precipitation by $(4.6. \pm 6.1) \times 10^6$ person-days in NIN, $(2.0 \pm 1.9) \times 10^6$ person-days in NC, $(3.6 \pm 3.8) \times 10^6$ person-days in the SCB, and $(2.0 \pm 2.2) \times 10^6$ person-days in SC. Also, the inter-model variation is large, which arises from differences in model simulations of extreme precipitation.

## 4 Discussion and conclusions

This study quantitatively assesses the impacts of reductions in emissions of non-methane SLCFs (including aerosols, ozone, and their precursors) on eastern and southern Asian climate extremes and the associated population exposure risks during 2031–2050 using MME simulations under the SSP3-7.0 and SSP3-7.0-lowNTCF scenarios from AerChemMIP. Our results show that non-methane SLCF reductions alone will exacerbate the warming effect caused by GHGs, resulting in increases in extreme temperature and precipitation events compared to the standard SSP3-7.0 scenario. These results are consistent with previous studies that have considered only aerosol reductions (Wang et al., 2016; Samset et al., 2018; Luo et al., 2020). Future reductions in non-methane SLCFs during 2031–2050 are projected to cause a global mean ERF of $0.23 \pm 0.16$ W m$^{-2}$ and an increase of $0.19 \pm 0.1$ K in global average SAT relative to 1995–2014 based on the MME results.

The additional warming caused by the reduction of non-methane SLCF emissions increases domain-averaged the hottest day by $0.3 \pm 0.1$ K, warm days by $4.8 \pm 2.2$ %, tropical night by $1.7 \pm 0.8$ days, and warm spell duration by $1.0 \pm 0.4$ days across the entire study area. The models agree on the sign of the change across regions, the hottest day, warm days, and warm

spell duration show their largest increases in NIN and NC, reaching $0.5 \pm 0.2$ K, $8.5 \pm 2.9$ %, and $2.8 \pm 1.1$ days and $0.4 \pm 0.2$ K, $5.9 \pm 2.7$%, and $3.0 \pm 1.3$ days, respectively. The increase in warm days due to non-methane SLCF mitigation is comparable to that caused by GHG emissions in some regions. The largest increases of tropical night are found in NC and the SCB, reaching $5.1 \pm 2.5$ and $4.9 \pm 3.3$ days, respectively. Overall, the warming trends driven by the combined reductions of aerosols, ozone, and their precursors are similar to the trends driven only by aerosol reductions. Such strong local effects of SLCFs on temperature extremes were also revealed in other high emission and population density regions such as Europe and the United States (Sillmann et al., 2013a; Samset et al., 2018; Luo et al., 2020). In addition, population exposure to warm days associated with future non-methane SLCF reductions is greatest in NIN, at $(32.2 \pm 11.4) \times 10^7$ person-days.

For extreme precipitation, future non-methane SLCF reductions increase domain-averaged heavy precipitation days by $1.0 \pm 0.5$ days, the maximum consecutive 5-day precipitation by $1.0 \pm 0.32$ mm, and total wet-day precipitation by $16.4 \pm 7.3$ mm in the entire study area. For heavy precipitation days, the maximum consecutive 5-day precipitation, and total wet-day precipitation, there is a robust increase in the southeastern Qinghai-Tibetan Plateau, SWC and the middle and lower reaches of the Yangtze River, with relatively good inter-model agreement and high confidence. In SWC, these extreme precipitation indices increase by $2.5 \pm 1.9$ days, $2.5 \pm 1.5$ mm, and $37.5 \pm 22.6$ mm, and these increases are greater or comparable to the effects of GHGs. By contrast, consecutive dry days decreases by $1.4 \pm 1.2$ days in SWC due to increases in the frequency and intensity of heavy precipitation events. Notably, our results show a large uncertainty and model disagreement in simulating future extreme precipitation. Previous studies also indicate that there are still large uncertainties in the model's simulations of extreme precipitation, which may attribute to model resolution and natural variation (Li et al., 2014; Deser et al., 2012). Besides, large differences are found at the regional scale in response to reductions of non-methane SLCFs and only aerosols, with changes in opposite directions observed in some regions. This discrepancy may be attributed to ozone and aerosol interactions, which may be further assessed via model simulations with both coupled and uncoupled experiments in the future. The spatial pattern of changes in the total population exposed to total wet-day precipitation due to non-methane SLCF reductions is largely consistent with the distribution of warm days, with the largest increases in NIN reaching $(4.6 \pm 6.1) \times 10^6$ person-days. Similarly, the inter-model variation in population exposure is large, which arises from differences in model simulations of extreme precipitation.

Compared with previous assessment by Allen et al. (2020), our study provides some new insights for the effects of future non-methane SLCF emissions on regional climate change. Firstly, although extreme temperature indices are all increasing in the future due to the reduction of non-methane SLCFs, warm days and warm spell duration vary spatially opposite to the hottest day, indicating that the warming of future temperature extremes is greater at higher latitudes, while the increase in the frequency and duration of extreme temperature occurrences is more pronounced at lower latitudes. As for extreme precipitation, changes in both heavy precipitation days and total wet-day precipitation in some areas are opposite in sign to previous results considering only aerosol reduction, revealing the importance of considering aerosol and ozone interactions. More importantly, we analyze the changes in tropical night and the maximum consecutive 5-day precipitation. The former represents the variation of nighttime temperature extremes that are important for human health. The latter is usually used as

an indicator of flooding, suggesting that heavy precipitation associated with natural disasters will be aggravated in the future due to non-methane SLCFs reduction in certain areas, especially in SC and SWC. Secondly, population exposure is an effective indicator for climate change risk assessment. Compared to SSP3-7.0 scenario, the reduction of non-methane SLCFs will result in additional exposure of millions of people to extreme events, and up to tens of millions in densely populated areas, such as

northern India, which is an indicator of human health risk and also valuable for future policy making on climate change mitigation and adaptation. Thirdly, Allen et al. (2020) used nine models, including five Aer+O3 models and four Aer-only models, but we used seven Aer+O3 models. The more Aer+O3 models may better reflect the effect of considering the combined aerosol and ozone changes simultaneously. Finally, Pendergrass et al. (2019) have shown that the response of extreme precipitation to warming varies widely in climate models, especially in the tropics. The rate of response increases with warming

is not linear (Pendergrass et al., 2019), as shown in Allen et al. (2020) that some of the extreme indices were not well fitted such as the wettest day and CDD, with lower value of $R^2$ or even lack significance. Freychet et al. (2019) suggested that radiation-driven aerosol emission impacts on local surface temperature and precipitation were not linear and could be mitigated or cancelled by the local dynamics. Our method of subtracting the mean between two periods may, to some extent, provide a more intuitive representation of the changes in the extreme indices in absolute terms.

Notably, GHGs share many common sources with SLCFs, such as the combustion of fossil fuels. Emission reductions for shorter-lived GHGs, such as methane, can partially mask or offset the warming caused by emission reductions of non-methane SLCFs over decades, providing benefits for both climate change mitigation and air quality on nearly all decadal to centennial time scales (Allen et al., 2021; Shindell and Smith, 2019). Increasing and accumulating anthropogenic GHG emissions are the main driving factor in shaping the increase and intensification of extreme high temperatures globally and

regionally. By the end of the century, the differences in temperature and precipitation between the different aerosol reduction scenarios are negligible in the context of ambitious $CO_2$ reductions (Hienola et al., 2018; Wilcox et al., 2020), and global warming is significantly less under strong mitigation scenarios with both climate policies and air quality controls (i.e., SSP1-1.9 and SSP1-2.6) than under the SSP3-7.0-lowNTCF scenario with only strong air quality control measures (Naik et a., 2021). Even if the impact of non-methane SLCFs is negligible on centennial time scales, they may be important for regional and

global climate in the coming decade, especially for Asia where aerosol has played an important role in historical changes, in particular for precipitation (Wilcox et al., 2020). The important impact of SLCFs on climate change in the short term does not mean that reducing air pollutants is harmful to the climate, but rather that it generates additional warming to the climate, amplifying the temperature and extreme precipitation caused by GHGs changes (Wang et al., 2016; Luo et al., 2020). In the short term, the differences in regional climate change under different future emission scenarios depend strongly on changes in

emissions of SLCFs, especially before net $CO_2$ emissions (and co-emitted aerosol emissions) become very low in the first half of this century (Hienola et al., 2018; Wilcox et al., 2020). Warming is most obvious in the strong mitigation scenarios (i.e., SSP1-1.9 and SSP1-2.6) because of the rapid reduction in aerosols. In the SSP3-7.0 scenario, aerosols do not decrease until mid-century, but increases in methane and ozone contribute to the net warming in 2040. The warming is similar in magnitude to the SSP1 scenario, where aerosol reductions are the primary driver (Wilcox et al. 2020; Naik et al. 2021). Also, the

conclusion of Shindell and Smith (2019) that there is no conflict between climate and air quality objectives may not hold when using a full coupled global climate model and when investigating changes beyond global mean temperature (Wilcox et al., 2020). Although it is difficult to improve air quality alone without reducing GHGs in reality, this is the case for some air quality policies, such as flue gas desulfurization in coal-fired power plants, denitrification, restaurant grease pollution control, and improved vehicle emission standards. These advanced end-of-pipe control measures may involve only the reduction of air pollutants (Rafaj et al., 2014; Hordijk and Amann, 2007). For example, European countries have taken specific measures to reduce air pollutant emissions, especially through the application of advanced end-of-pipe emission control technologies, resulting in a significant decline in $SO_2$ and NOx emissions in Western Europe after the 1970s, compared to a constant growth rate of $CO_2$ emissions (Rafaj et al., 2014; Hordijk and Amann, 2007). For China, the State Council implemented the Air Pollution Prevention and Control Action Plan in 2013 and the Three-Year Action Plan to Win the Blue Sky Defense War implemented in 2018 both introduced a series of aggressive industrial clean air policies (Zheng et al., 2018; Cheng et al., 2019). The reality in China's air pollution prevention and control policies in recent years is that China's treatment is still dominated by end-of-pipe measures, as source treatment often requires large investments to ensure energy efficiency and is not conducive to maintaining the competitiveness of Chinese industry (Wu et al., 2019), in which case it may only result in rapid reductions in air pollutants. Besides, emissions reductions aimed at achieving global carbon neutrality will inevitably result in further reductions in SLCF emissions, as demonstrated in the SSP1-1.9 and SSP1-2.6 scenarios (Gidden et al., 2019), which may lead to greater impacts of SLCFs on climate. The maximum technical potential in SSP3-7.0-lowNTCF scenario refers to the currently existing end-of-pipe technologies, with faster technological progress and stronger air quality action, greater emission reductions may also be possible, which may cause greater impacts on climate. Additionally, cleaner air may already have increased the warming effect of $CO^2$ emissions over the past two decades, and this will get worse as air pollution continues to be controlled (Quaas et al., 2022; McKenna et al., 2021). Finally, large biases may exist in SLCF emissions at the regional scale in global emissions scenarios due to insufficient consideration of local environmental policies (Tong et al., 2020). For example, the SSP-RCP global emissions scenarios used in CMIP6 do not fully consider the rapid pollution controls enacted in China since 2013 under the Air Pollution Prevention and Control Action Plan. Consequently, CMIP6 database underestimate emission reductions obtained from China's Action Plan (Wang et al., 2021; Tong et al., 2020), which means that the impact of emission reductions that have already happened may be underestimated, thus may lead to underestimation of the impact of SLCF emissions reductions in China. This study highlights the importance of reductions in emissions of non-methane SLCFs for future climate change and population exposure risk in eastern and southern Asia in the short term and suggests that current and future policy decisions about air pollution emissions have the potential for a large near-term impact on temperature and precipitation extremes. Air pollution is dangerous to human health, and there is no doubt that we need clean air, but more importantly efforts to reduce GHGs need to be doubled in order to simultaneously mitigate climate change and improve air quality (Quaas et al., 2022; McKenna et al., 2021).

*Data availability.* The CMIP6 Historical and AerChemMIP model data can be freely downloaded from https://esgf-node.llnl.gov/projects/cmip6. The observed gridded data can be downloaded at the website of https://psl.noaa.gov/data/gridded/index.html. Gridded population datasets for the base year and future under SSP3 scenario are publicly available at https://www.cgd. ucar.edu/iam/modeling/spatial-population-scenarios.

*Author contributions.* ZW conceived the study. YL performed the analysis. YL, ZW, and YL led the paper writing. All authors provided comments and contributed to the text.

*Competing interests.* The authors declare that they have no conflict of interest.

*Acknowledgments.* This study was supported by the National Natural Science Foundation of China (42275042 and 41875179) and the Science and technology development fund of CAMS (2022KJ004).

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

| Model | Institution | Country | Atmospheric resolution (Longitude ×Latitude) | realizations |
|-------|-------------|---------|-----------------------------------------------|--------------|
| BCC-ESM1 | Beijing Climate Centre (BCC) | China | 2.81°×2.81° | 3 |
| CESM2-WACCM | National Center for Atmospheric Research (NCAR) | United States | 1.25°×0.94° | 3 |
| GISS-E2-1-G | Goddard Institute for Space Studies (GISS) | United States | 2.0°×2.5° | 3 |
| GFDL-ESM4 | Geophysical Fluid Dynamics Laboratory (GFDL) | United States | 1.25°×1.0° | 1 |
| EC-Earth3-AerChem | European consortium of meteorological services, research institutes and high-performance computing centers (EC-Earth consortium) | European consortium | 0.70°×0.70° | 2 |
| MRI-ESM2-0 | Meteorological Research Institute (MRI) | Japan | 1.87°×1.87° | 3 |
| UKESM1-0-LL | Met Office Hadley Centre (MOHC) | United Kingdom | 1.88°×1.88° | 3 |

**Table 1: Summary of CMIP6 models used in this study.**

| Index Name | Index Definition | Units |
|---|---|---|
| The hottest day (TXx) | Annual average of the monthly maximum value of daily maximum temperature | K |
| The coldest day (TNn) | Annual average of monthly minimum value of daily maximum temperature | K |
| Warm days (TX90p) | Percentage of days with daily maximum temperature > 90% percentile | % |
| Warm nights (TN90p) | Percentage of days with daily minimum temperature > 90% percentile | % |
| Tropical nights (TR) | Annual count of days when daily minimum temperature > 20°C. | days |
| Warm spell duration (WSDI) | Annual count of days with at least 6 consecutive days when daily maximum temperature > 90th percentile | days |
| Maximum consecutive 5-day precipitation (RX5day) | Monthly maximum consecutive 5-day precipitation | mm |
| Total wet-day precipitation (R95p) | Annual total precipitation on days when daily precipitation >95th percentile. | mm |
| Heavy precipitation days (R10) | Annual count of days when precipitation ≥ 10mm | days |
| Consecutive dry days (CDD) | The maximum length of dry spell, the maximum number of consecutive days with precipitation < 1mm | days |

**Table 2: Definitions of the extreme climate indices used in this study.**

| Models | ERF (W/m$^2$) | SAT (K) |
|---|---|---|
| BCC-ESM1 | 0.23 | 0.20 |
| CESM2-WACCM | 0.41 | 0.29 |
| EC-Earth3-AerChem | 0.50 | 0.23 |
| GFDL-ESM4 | -0.03 | 0.24 |
| GISS-E2-1-G | 0.19 | 0.06 |
| MRI-ESM2-0 | 0.12 | 0.13 |
| UKESM1-0-LL | 0.20 | 0.22 |
| MMM | 0.23 | 0.19 |

**Table 3. Global effective radiative forcing (ERF) and surface air temperature (SAT) responses to non-methane SLCFs mitigation. MMM is the multi-model mean.**

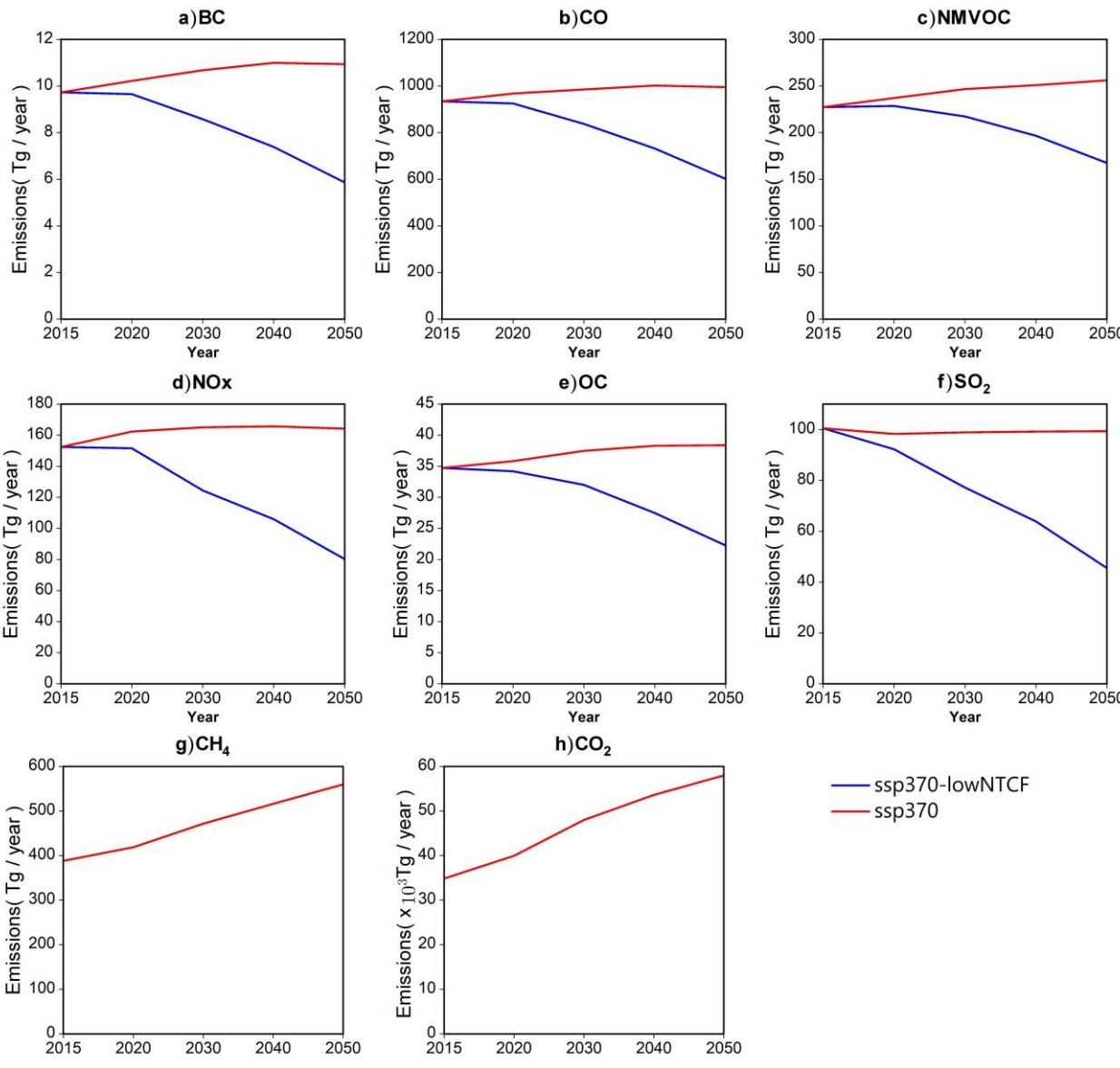

**Figure 1: Time series of global annual emissions of (a) BC, (b) CO, (c) NMVOC, (d) NOx, (e) OC, (f) SO₂, (g) CH₄, and (h) CO₂ from 2015-2050 under the SSP3-7.0 (red solid line) and SSP3-7.0-lowNTCF (blue solid line) scenarios. Only the red line is visible in (g,h) as ssp370 and ssp370-lowNTCF overlap.**

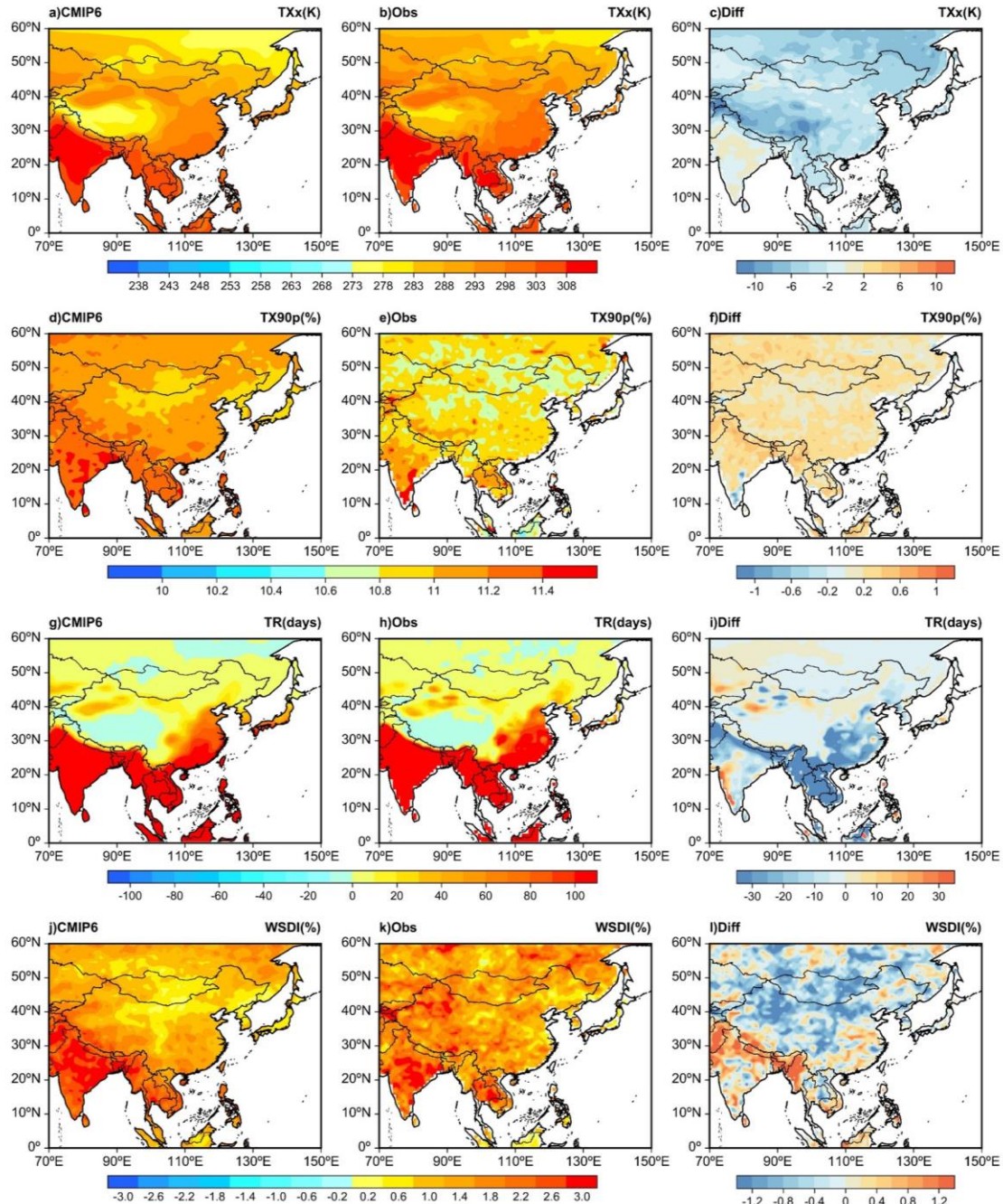

**Figure 2: The annual mean of the hottest day (TXx), warm days (TX90p), tropical nights (TR), and warm spell duration (WSDI) over study area during 1995-2014 for CMIP6 multi-model mean (left column) and gridded observations (middle column). The right column represents difference between CMIP6 and observations, so the red area in this column indicates that model is over-projecting.**

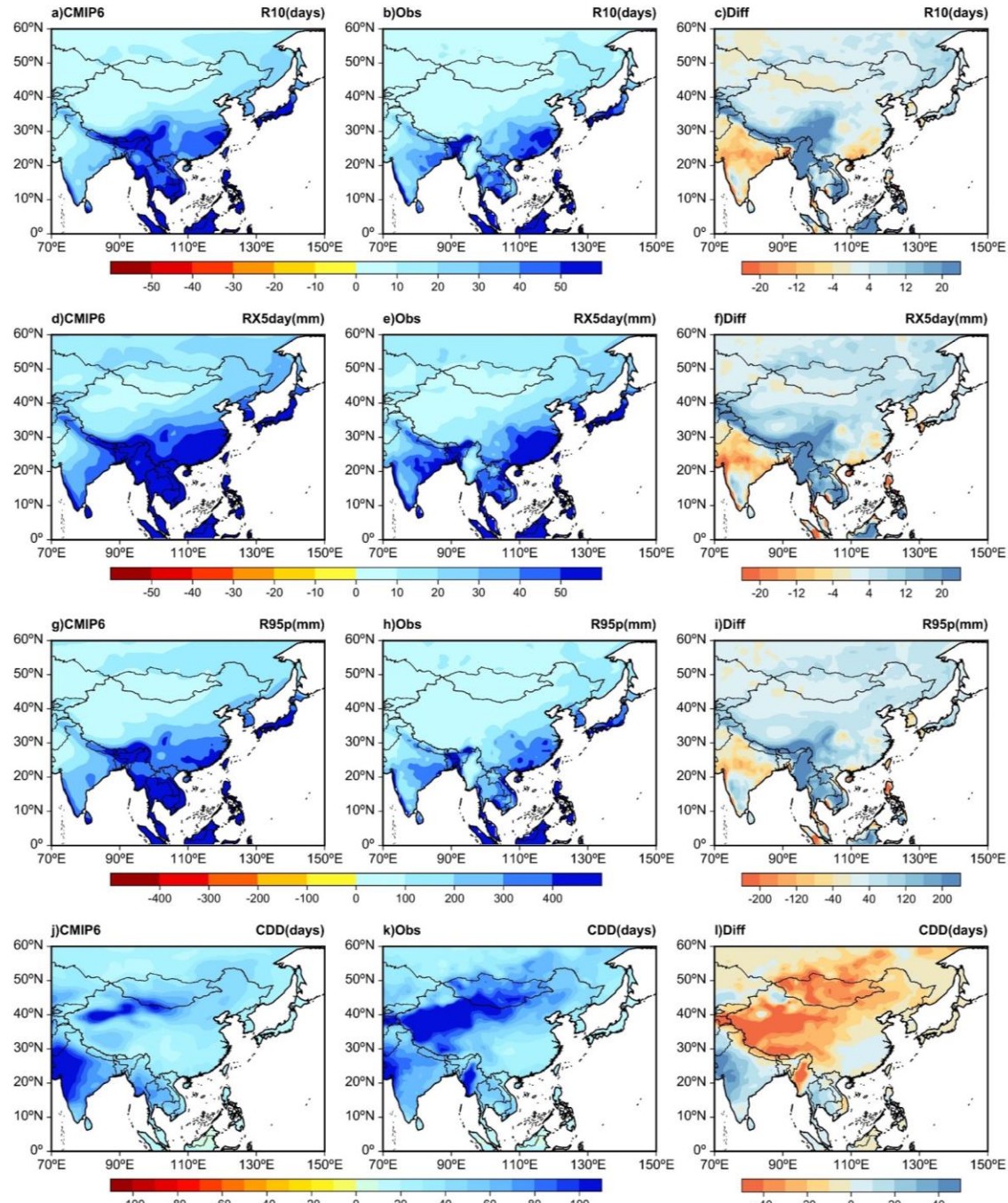

**Figure 3: The annual mean of heavy precipitation (R10), maximum consecutive 5-day precipitation (RX5day), total wet-day precipitation (R95p), and consecutive dry days (CDD) over study area during 1995-2014 for CMIP6 multi-model mean (left column) and gridded observations (middle column). The right column represents difference between CMIP6 and observations.**

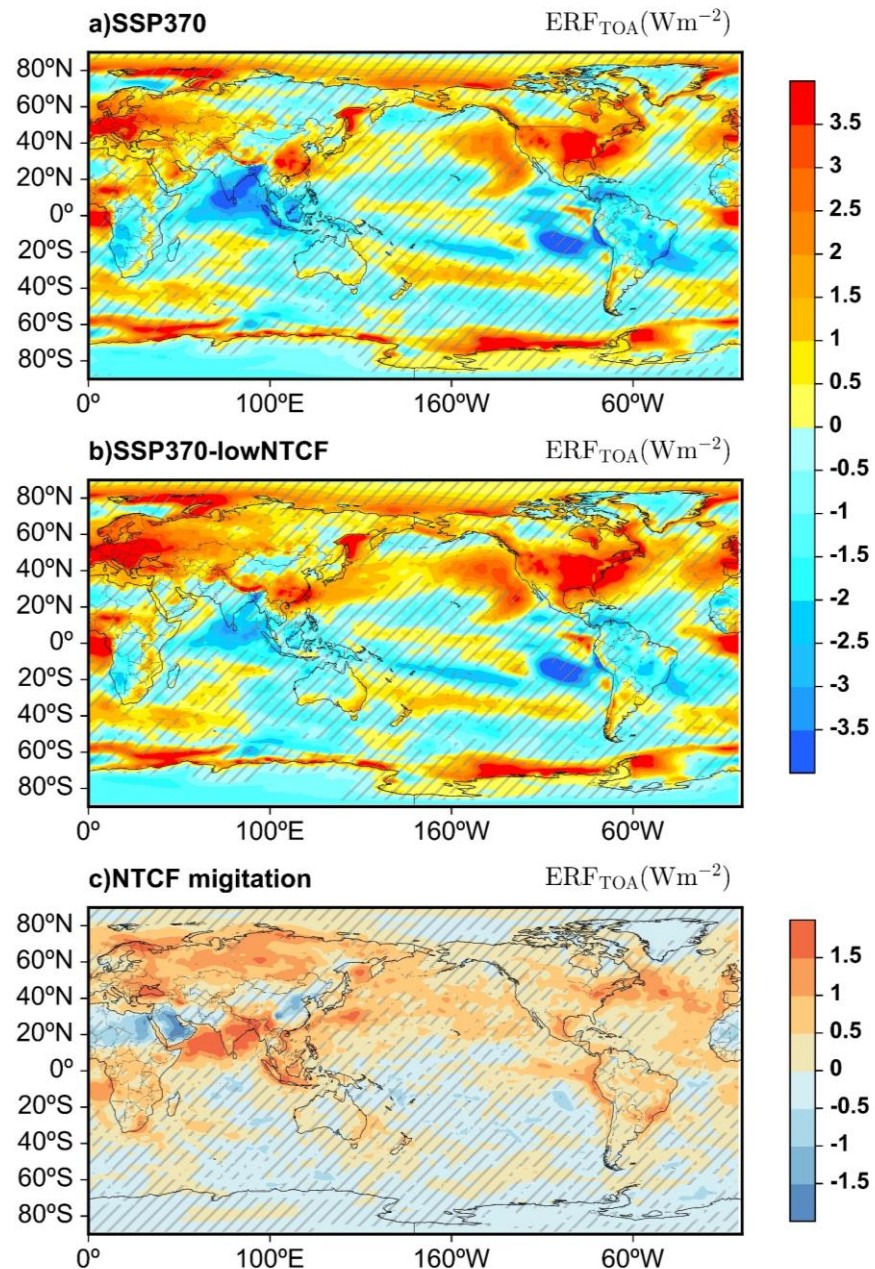

**Figure 4: Spatial patterns of effective radiative forcing (ERF) during 2031-2050 under the (a) SSP3-7.0 and (b)SSP3-7.0-lowNTCF scenarios and (c) only caused by the non-methane SLCFs reductions relative to 1995-2014 (units: W m⁻²). The areas with hatching indicate that at least one out of seven models disagree on the sign.**

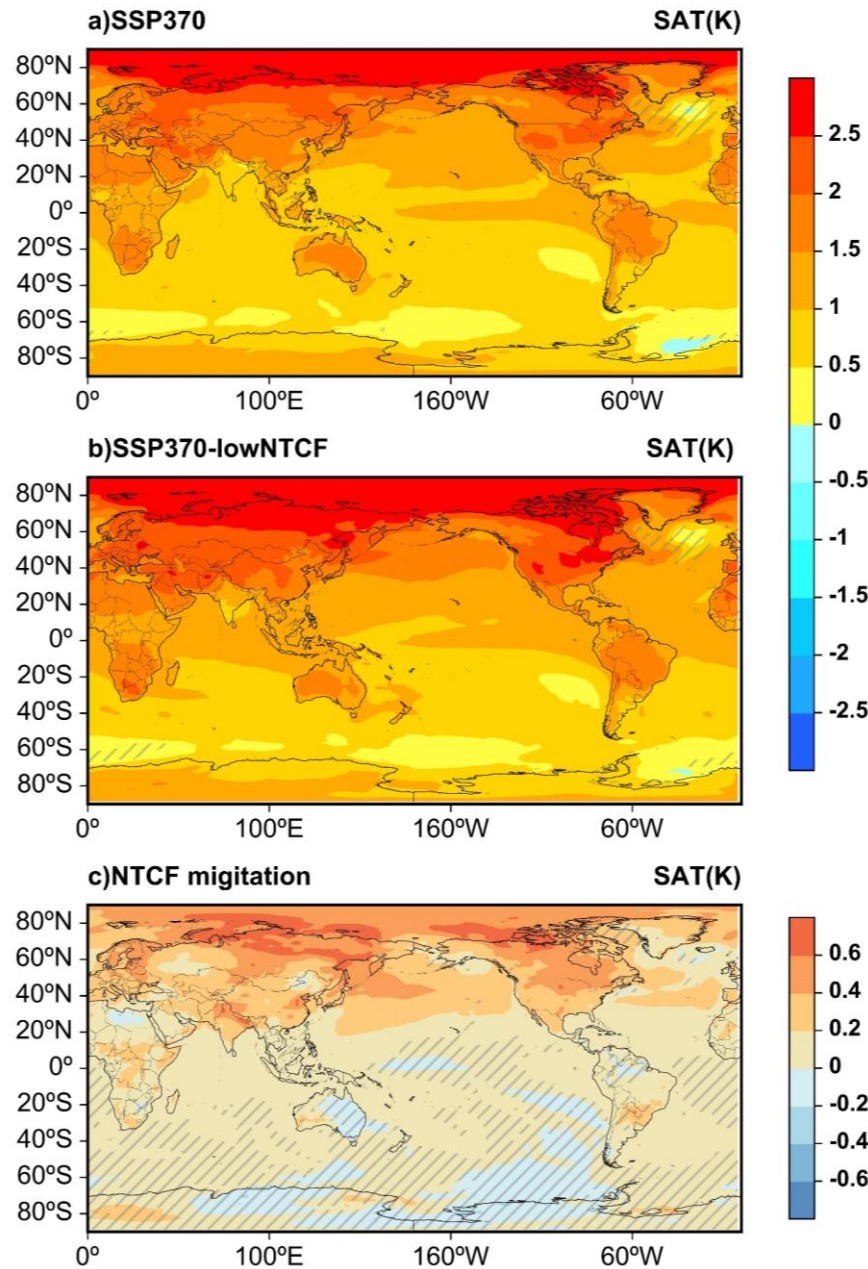

**Figure 5: Spatial patterns of changes in surface air temperature (SAT) during 2031-2050 under the (a) SSP3-7.0 and (b) SSP3-7.0-lowNTCF scenarios and (c) only caused by the non-methane SLCFs reductions relative to 1995-2014surface air (units: K). The areas with hatching indicate that at least one out of seven models disagree on the sign.**

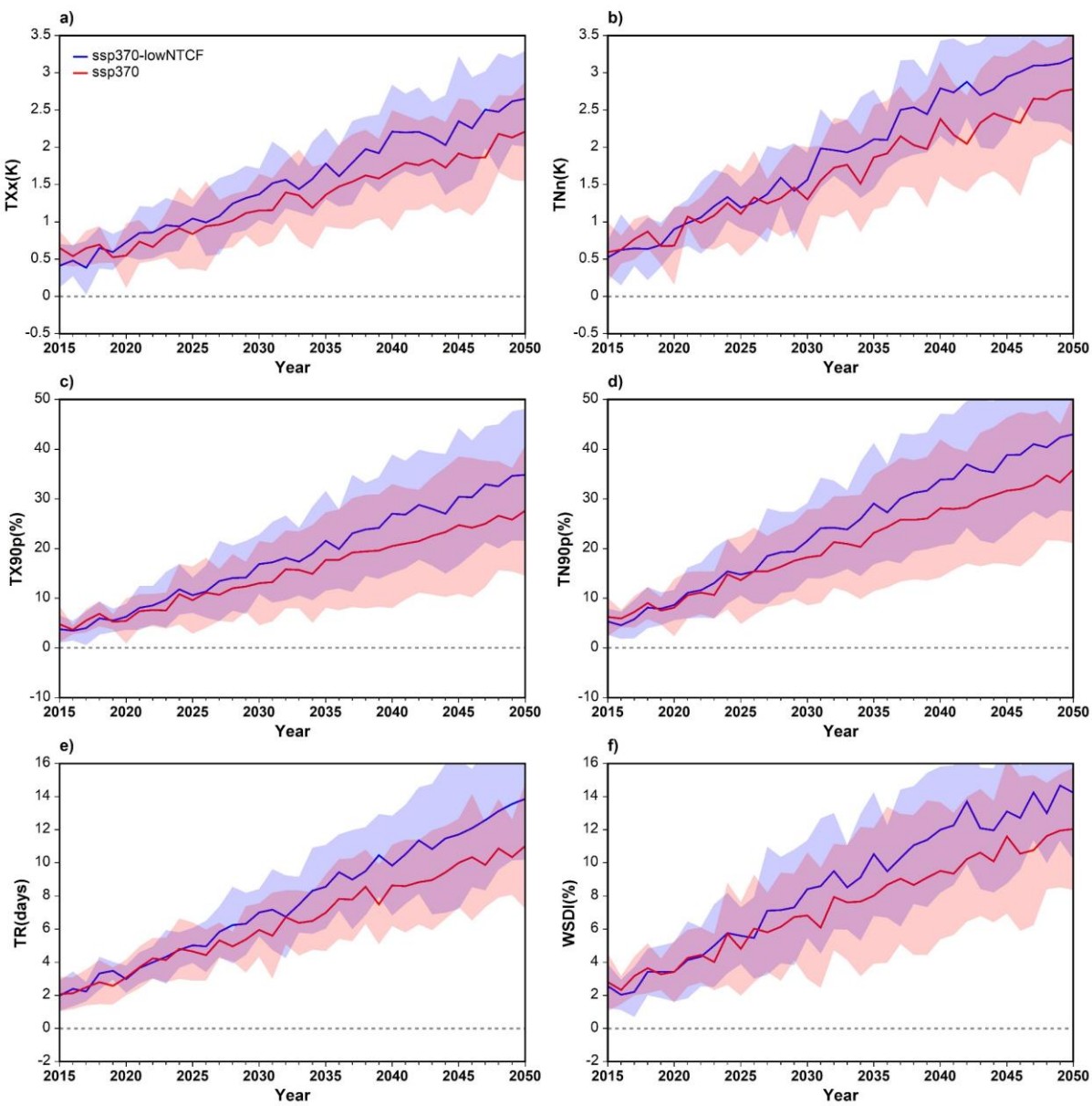

**Figure 6: Time series of changes in annual mean the hottest day (TXx), warm days (TX90p), tropical nights (TR), and warm spell duration (WSDI) averaged over Asia under the SSP3-7.0 (red) and SSP3-7.0-lowNTCF (blue) scenarios from 2015 to 2050 relative to 1995-2014. The red and blue shading represents the ± 1 standard deviations across models.**

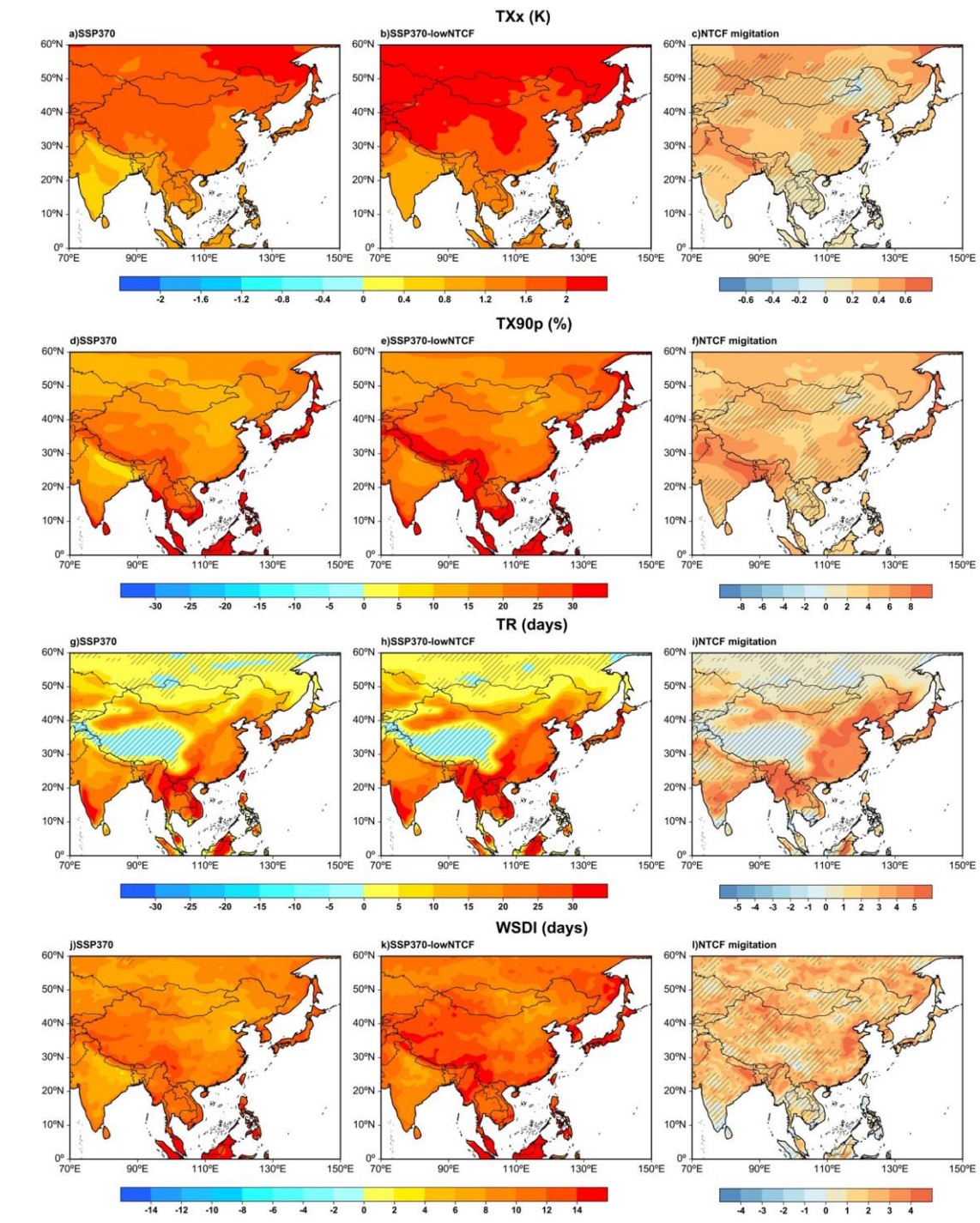

**Figure 7: Spatial patterns of changes in the hottest day (TXx), warm days (TX90p), tropical nights (TR), and warm spell duration (WSDI) during 2031-2050 in Asia under the SSP3-7.0 (left column) and SSP3-7.0-lowNTCF (middle column) scenarios relative to 1995-2014. The right column represents changes caused by the non-methane SLCFs mitigation. The areas with hatching indicate that at least one out of five models disagree on the sign.**

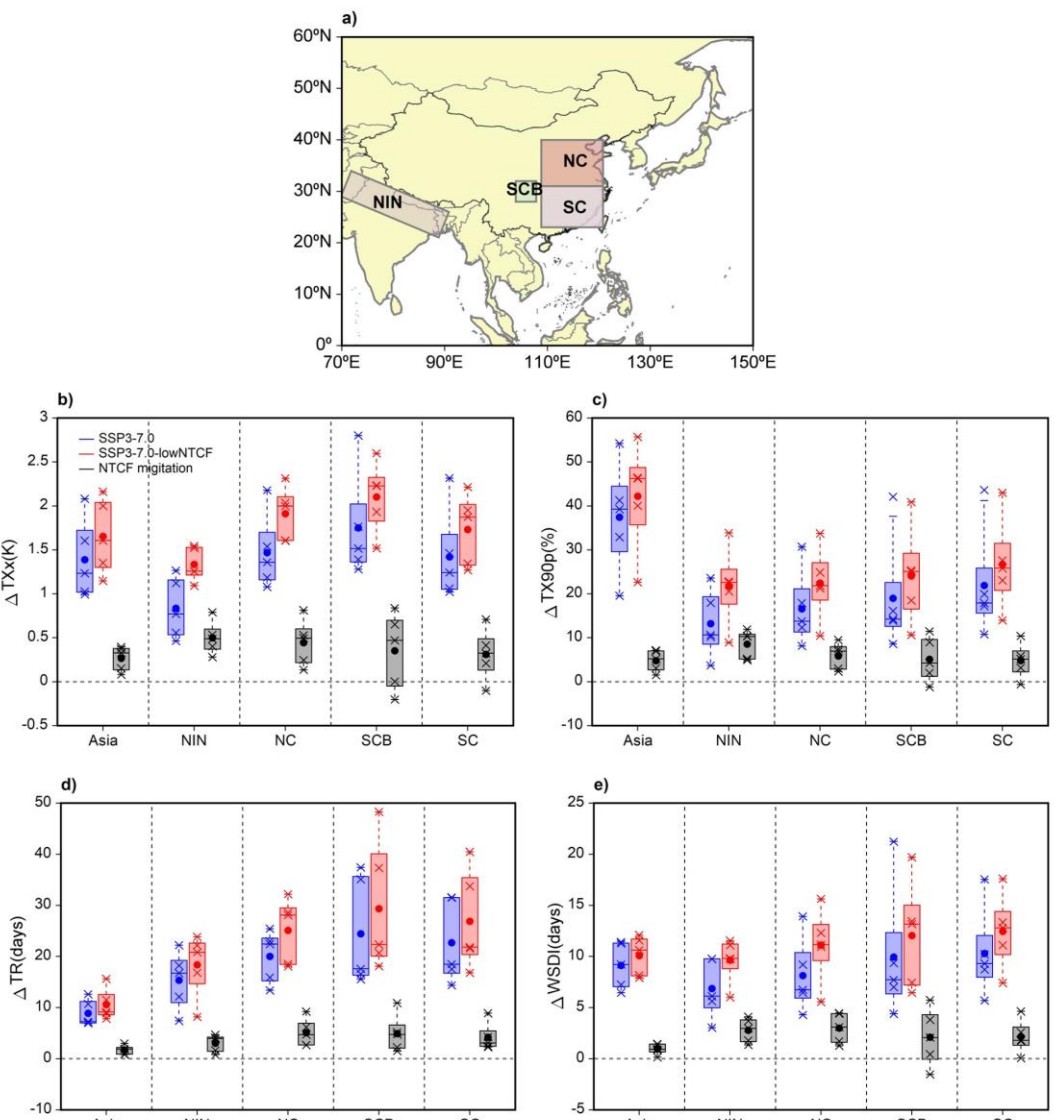

Figure 8: Changes in the hottest day (TXx), warm days (TX90p), tropical nights (TR), and warm spell duration (WSDI) averaged over NIN, NC, SC, and SCB during 2031-2050 relative to 1995-2014 under the SSP3-7.0 (red) and SSP3-7.0-lowNTCF (blue) scenarios. The gray bars represent the changes caused by the non-methane SLCFs reductions. The star symbols indicate the individual models, and the boxes indicate the spread from the 25th to the 75th percentile of the models, with the center lines representing the median and the dot representing the mean values.

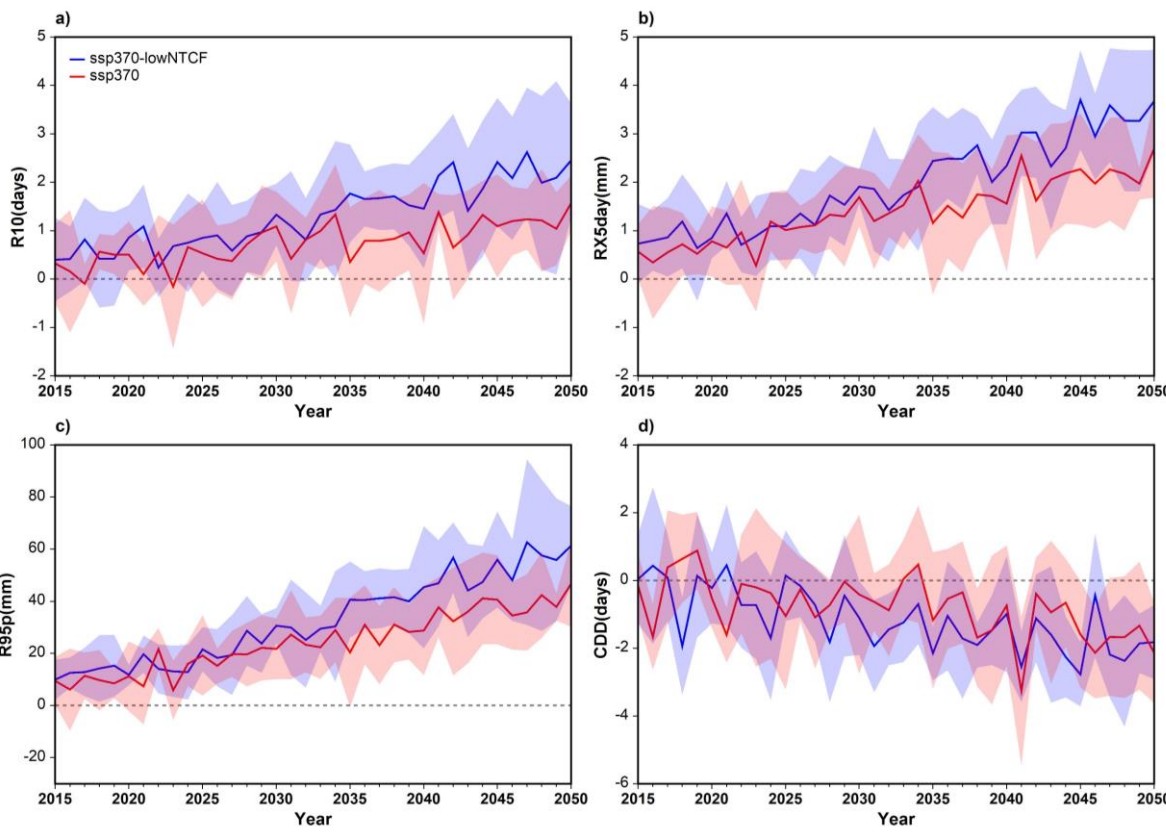

**Figure 9: Time series of changes in annual mean heavy precipitation (R10), maximum consecutive 5-day precipitation (RX5day), days, total wet-day precipitation (R95p), and consecutive dry days (CDD) averaged over Asia under the SSP3-7.0 (red) and SSP3-7.0-lowNTCF (blue) scenarios from 2015 to 2050 relative to 1995-2014. The red and blue shading represents the ± 1standard deviations across models.**

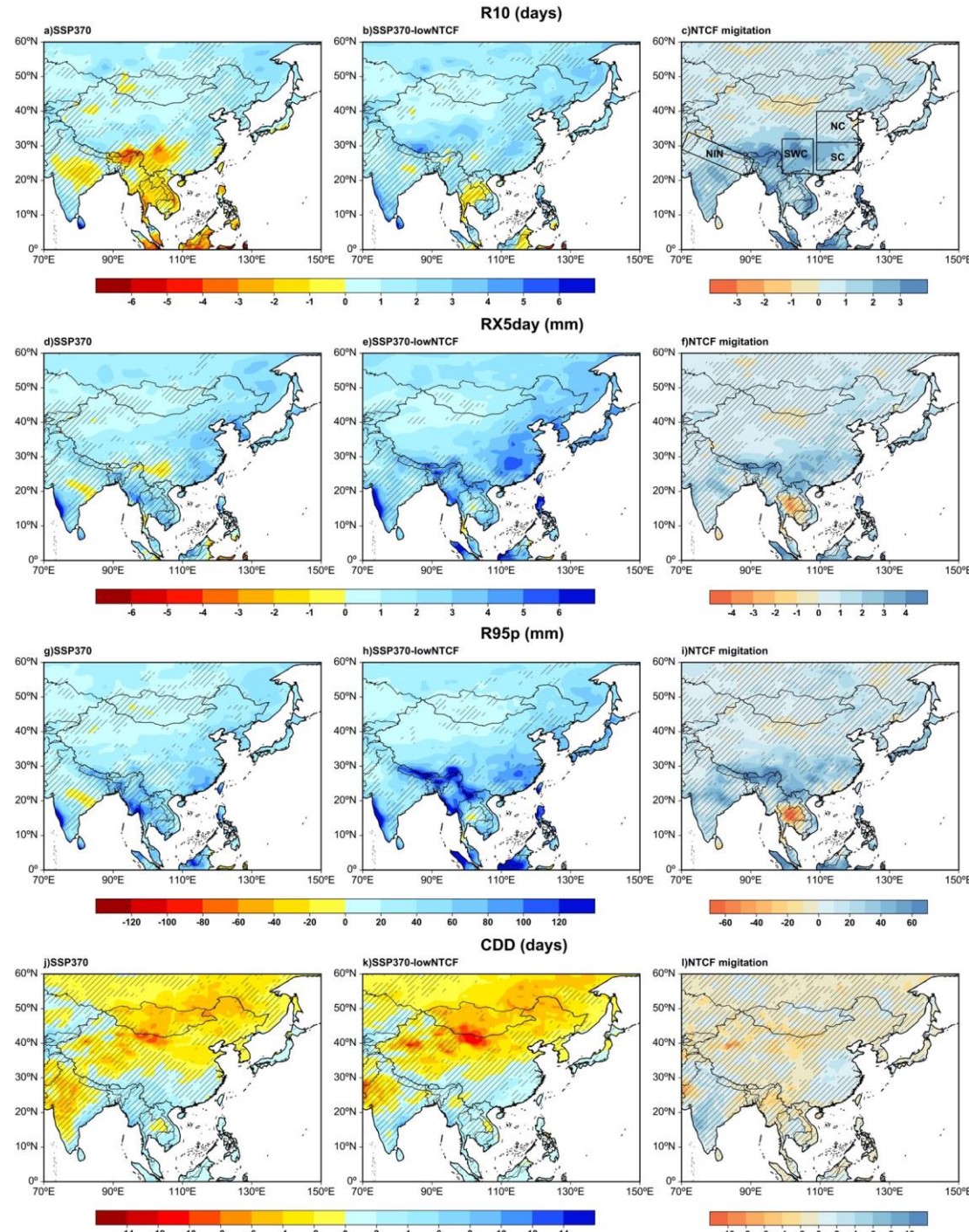

**Figure 10: Spatial patterns of changes in heavy precipitation (R10), maximum consecutive 5-day precipitation (RX5day), total wet-day precipitation (R95p), and consecutive dry days (CDD) during 2031-2050 in Asia under the SSP3-7.0 (left column) and SSP3-7.0-lowNTCF (middle column) scenarios relative to 1995-2014. The right column represents changes caused by the non-methane SLCFs mitigation. The areas with hatching indicate that at least one out of six models disagree on the sign.**

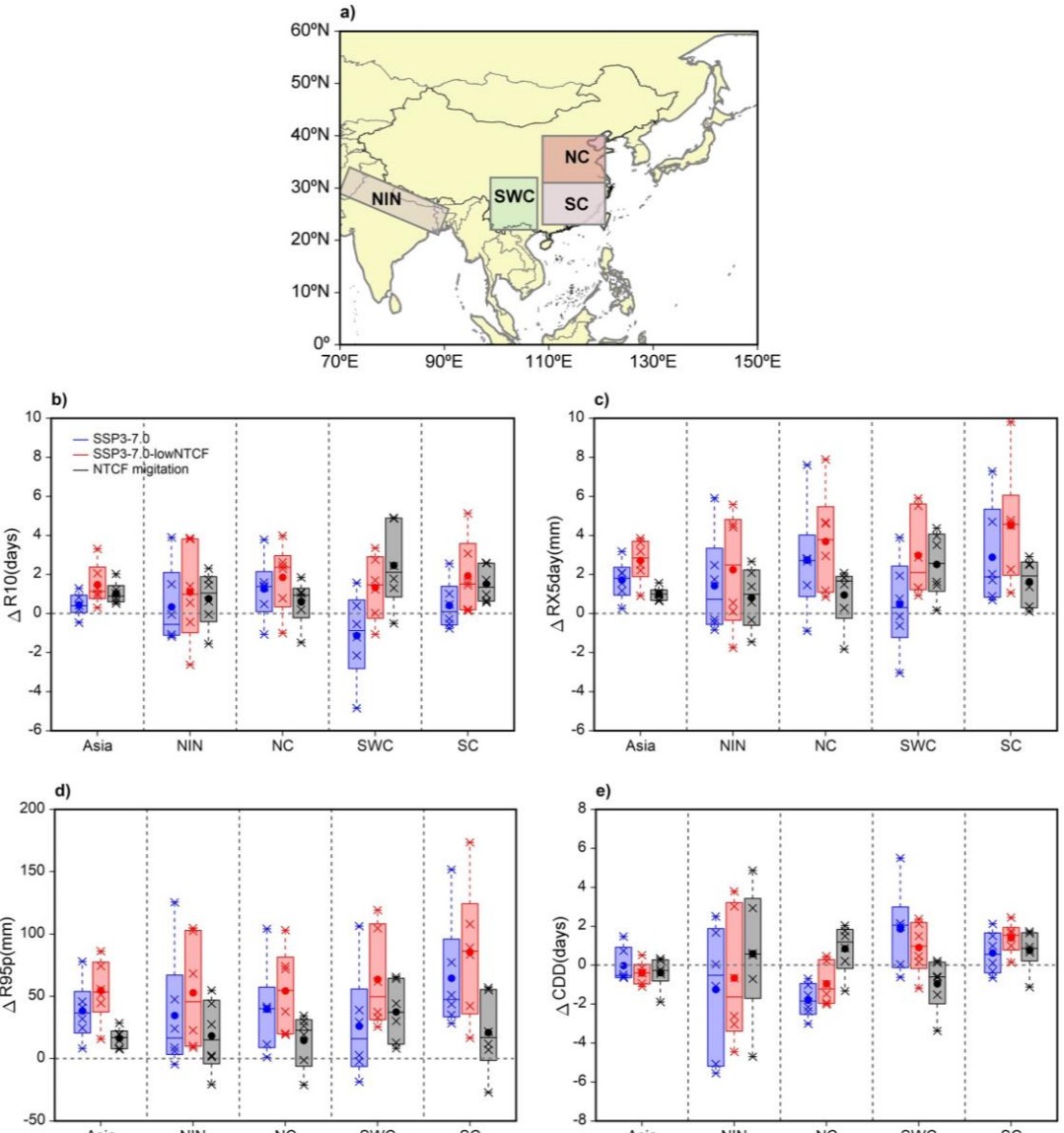

**Figure 11: Changes in heavy precipitation (R10), maximum consecutive 5-day precipitation (RX5day), days, total wet-day precipitation (R95p), and consecutive dry days (CDD) averaged over NIN, NC, SC, and SCB during 2031-2050 relative to 1995-2014 under the SSP3-7.0 (red) and SSP3-7.0-lowNTCF (blue) scenarios. The gray bars represent the changes caused by the non-methane SLCFs reductions. The star symbols indicate the individual models, and the boxes indicate the spread from the 25th to the 75th**

**percentile of the models, with the center lines representing the median and the dot representing the mean values.**

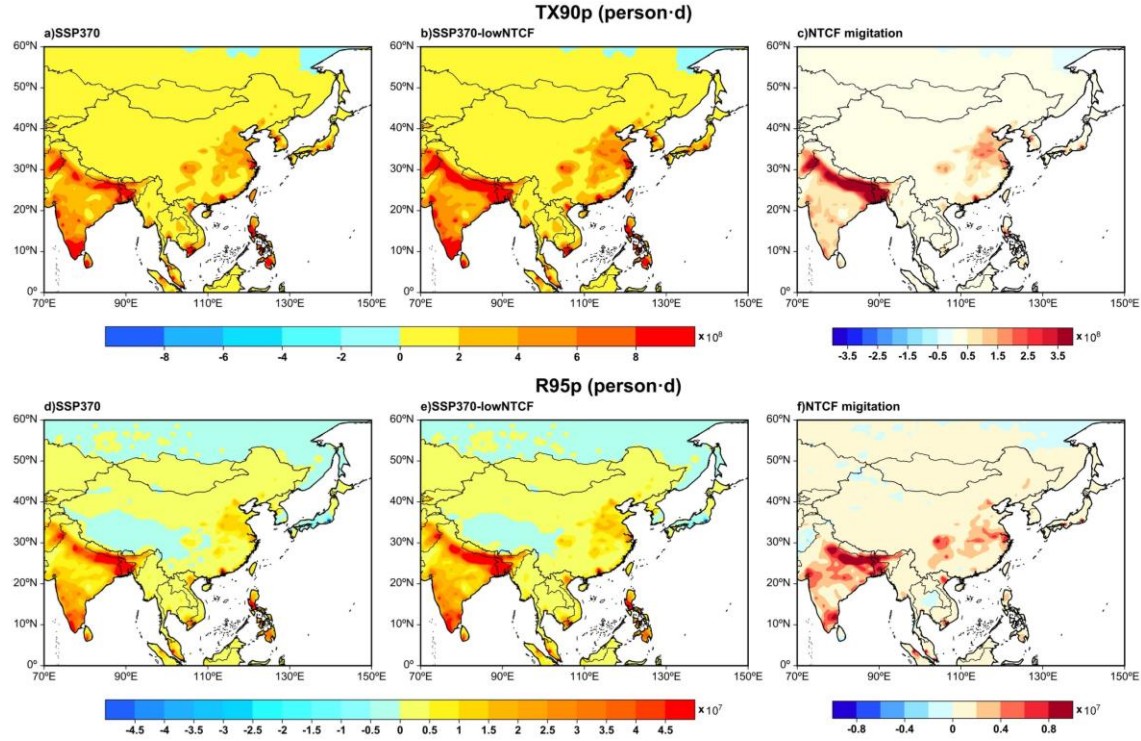

**Figure 12: Spatial patterns of changes in population exposed to warm days (TX90p) and total wet days (R95pdays) in Asia during 2031-2050 under the SSP3-7.0 (left) and SSP3-7.0-lowNTCF (middle) relative to 1995-2014 (units: person-days). The right column represents the changes caused by the non-methane SLCFs reductions.**

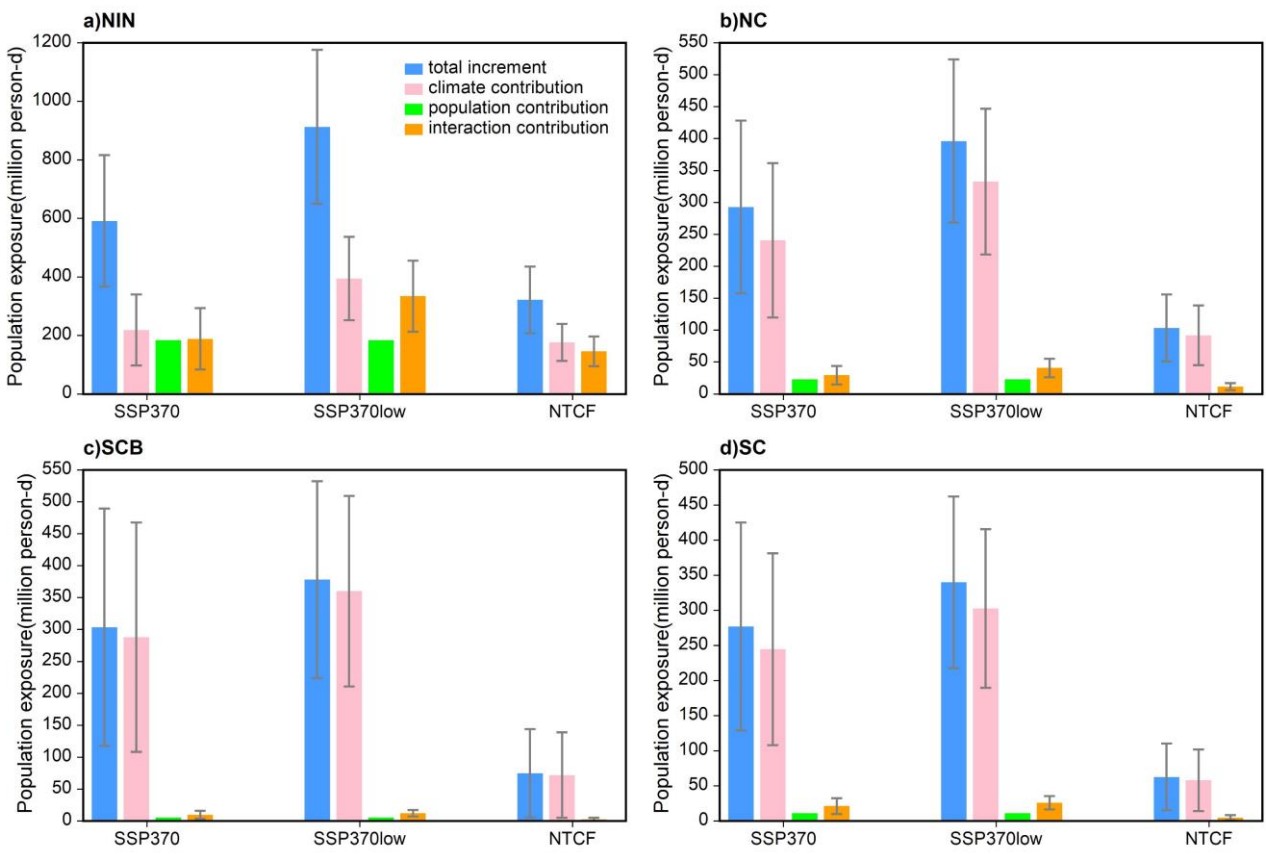

**Figure 13: The total changes of population exposure to warm days (TX90p) averaged over northern India (NIN), northern China (NC), the Sichuan Basin (SCB) and southern China (SC) under the SSP3-7.0 and SSP3-7.0-lowNTCF scenarios and its changes driven by the non-methane SLCFs reductions. The pink, green, and orange bars represent the changes driven by climate change, population change, and population-climate interaction, respectively. The error bars denote the ± 1 standard deviations across models.**

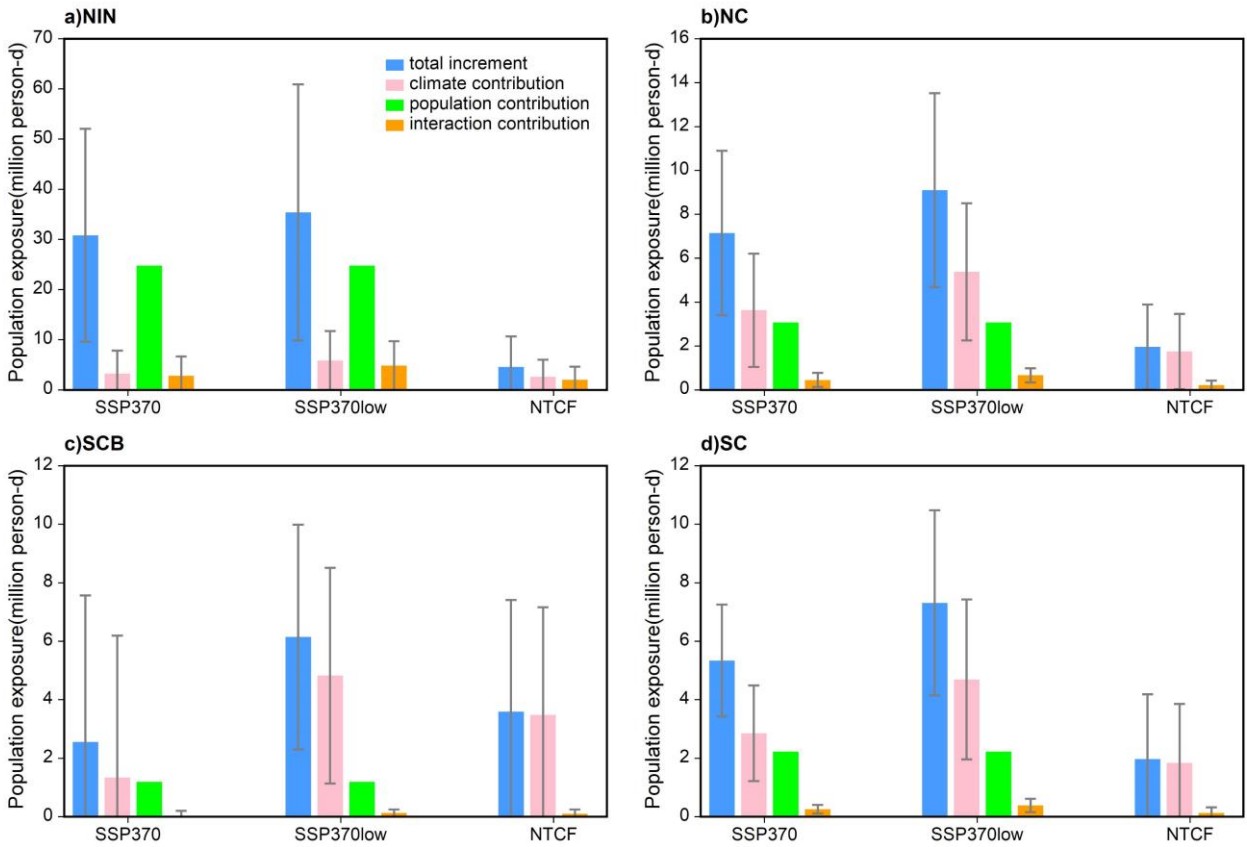

**Figure 14: The total changes of population exposure to wet days (R95pdays) averaged over northern India (NIN), northern China (NC), the Sichuan Basin (SCB) and southern China (SC) under the SSP3-7.0 and SSP3-7.0-lowNTCF scenarios and its changes driven by the non-methane SLCFs reductions. The pink, green, and orange bars represent the changes driven by climate change, population change, and population-climate interaction, respectively. The error bars denote the ± 1 standard deviations across models.**