# Peer review of "Impacts of reductions in non-methane short-lived climate forcers on future climate extremes and the resulting population exposure risks in eastern and southern Asia"

_Atmospheric Chemistry and Physics, 2022_

## Author Comment (AC1)

**Response to Referee #1:**

We are grateful to the referees for their time and energy in providing helpful comments and guidance that have improved the manuscript. In this document, we describe how we have addressed the reviewer's comments. Referee comments are shown in black and author responses are shown in blue text.

Summary:

Li et al. use CMIP6 simulations (SSP3-7.0 and SSP3-lowNTCF, part of AerChemMIP) to compare the impacts of increasing emissions of greenhouse gases+aerosols (SSP3-7.0) as opposed to reduced emissions of aerosols (Near-Term Climate Forcers, or NTCFs) and continued emissions of greenhouse gases (SSP3-lowNTCF, which is an idealized simulation that implements future air quality standards that do not reduce GHG emissions). The authors first compare global impacts of SSP3-7.0, SSP3-lowNTCF, and the difference between the two experiments (under the assumption that studying the difference isolates the climate impact of reduced non-methane NTCFs), then specifically South/East Asia. The authors report changes in Effective Radiative Forcing (ERF) and increased climate extremes. I found this paper to be generally well written and the text easy to follow, and the results are interesting and potentially important if policymakers implement short-term air pollution controls only (without reducing any greenhouse gas emissions). My main concerns are outlined below, followed by specific line comments.

Before I continue my review, I should note that I am relatively new to the research area of NTCF simulations, but I have more experience in the area of population exposure, CMIP ensemble analysis, and climate impacts of emissions. I am not a statistician or climate extremes expert, so I have focused my comments on the overall scientific quality of the paper, novelty of the work, clarity of the writing/figures, and overall climate impacts assessed.

- Thank you for your positive evaluations. All the questions and concerns have been carefully answered and the paper has been revised accordingly.

Main Concerns:

1. I am less familiar with the literature related to AerChemMIP, so I went back to re-read a few of the papers Li et al. cite in the introduction. Specifically, I read Allen et al. (2020), who conducted a very similar analysis (compared SSP3-7.0 to SSP3-lowNTCF simulations) and parts of the IPCC AR6 WG1 Chapter 6.

My main concern is that many of the results presented here are either quantitatively or qualitatively similar to those in Allen et al.: Allen et al. compare CMIP6 SSP3-7.0 and SSP3-lowNTCF, and already make the main points this manuscript makes- specifically, climate extremes (temperature, precipitation) are intensified with reduced aerosol forcing, and this change in extremes is strongly felt over parts of southern and eastern Asia. Some (but not all) of the climate extreme indices are even already presented in Allen et al.

I do recognize that there is some new material here because Li et al. assess a wider range of climate extreme indices, examine population-weighted climate impacts, and focus on Asia. However, I didn't notice any outstanding new findings, but instead more details on findings similar to those already published. The authors simply acknowledge that Allen et al. examine 'trends', but don't really acknowledge very much beyond that, and even re-present similar information (e.g., changes in ERF), but using a slightly different method (change in means over two time periods instead of trends, etc).

**Response:** Thanks for your constructive comments! As suggested, we clarified our motivation and discussed the novelty of this study compared with Allen et al. (2020) in the revised paper. The following information are added in Introduction and Discussion sections:

(i) "However, there were some limitations in that study. First, Allen et al. (2020) only considered three extreme indicators including hottest day, wettest day, and consecutive dry days to examine the effects of future non-methane SLCF reductions on climate extremes. These three indicators were not enough to represent climate extremes, especially the lack of some indicators related to human health. For example, tropical night (TR) usually occurs in combination with extended

periods of heat (particularly in extra-tropical regions) and have been suggested to be problematic for human health (Weisskopf et al., 2002; Patz et al., 2005) and the maximum consecutive 5-day precipitation (RX5day) can be used as an indicator of flooding and related hazards (Frich et al., 2002; Sillmann et al., 2013). Second, climate extremes pose a serious threat on human body (Bras et al., 2021; Tellman et al., 2021). Quantifying avoided population exposure to climate extremes associated with future non-methane SLCF reductions is valuable for future policymaking on climate change mitigation and adaptation, especially in these densely populated and industrially developed regions of Asia, which is lacked in Allen et al. (2020)." Lines 84-93

(ii) "Compared with previous assessment by Allen et al. (2020), our study provides some new insights for the effects of future non-methane SLCF emissions on regional climate change. Firstly, although extreme temperature indices are all increasing in the future due to the reduction of non-methane SLCFs, TX90p and WSDI vary spatially opposite to TXx, indicating that the warming of future temperature extremes is greater at higher latitudes, while the increase in the frequency and duration of extreme temperature occurrences is more pronounced at lower latitudes. As for extreme precipitation, changes in both R10 and R95p in some areas are contrary to previous results considering only aerosol reduction, revealing the importance of considering aerosol and ozone interactions. More importantly, we analyze the changes in TR and RX5day. The former represents the variation of nighttime temperature extremes that are important for human health. The latter is usually used as an indicator of flooding, suggesting that heavy precipitation associated with natural disasters will be aggravated in the future due to non-methane SLCFs reduction. Secondly, population exposure can provide a well assessment of future climate change risk. The reduction of non-methane SLCFs will result in the exposure of millions of people to extreme events, and up to tens of millions in densely populated areas, such as northern India, which is an indicator of human health risk and also valuable for future policymaking on climate change mitigation and adaptation, Thirdly, Allen et al. (2020) used nine models, including five Aer+$O_3$ models and four Aer-only models, but we used seven Aer+$O_3$ models. The more Aer+$O_3$ models may better reflect the effect of considering the combined aerosol and ozone changes simultaneously. Finally, Pendergrass et al. (2019) have shown that the response of extreme precipitation to warming varies widely in climate models, especially in the tropics. The rate of response increases with warming is not linear but

non-linear (Pendergrass et al., 2019), as shown in Allen et al. (2020) that some of the extreme indices were not well fitted. Freychet et al. (2019) suggested that radiation-driven aerosol emission impacts on local surface temperature and precipitation were not linear and could be mitigated or cancelled by the local dynamics. Our method of subtracting the mean between two periods may, to some extent, provide a more intuitive representation of the changes in the extreme indices in absolute terms." Lines 426-446

2. A critique of how this experiment (SSP3-lowNTCF) is analyzed and presented here: there seems to be a history of 'idealized emissions reduction modeling' experiments that have led to misconceptions related to the impacts of aerosol reductions in the scientific literature and in the public/media (e.g., see Shindell and Smith, 2019, Nature, who argue against the realism of immediate 'zero emissions' and the associated spike in warming).

Admittedly, SSP3-lowNTCF is a more 'realistic' scenario that doesn't implement immediate 'zero emissions' – However, I went back to the IPCC AR6 WG1 Chapter 6.6 and 6.7, and the IPCC specifically states that SSP3-lowNTCF is 'an idealized simulation of a very ambitious air-quality policy where the maximum technical potential of existing end-of-pipe technologies is explored'. How realistic is it that nations will only implement air quality, and absolutely no GHG reductions? And how are the results presented here to be interpreted (esp. by policymakers, the public, and/or the media)? I understand that there is scientific value in distinguishing among GHG vs aerosol reductions, but how is this information to be used, and what is the context?

I bring these issues up because I think the Discussion/Conclusion could be strengthened by at least a qualitative comparison with SSP1 (which in my understanding is where the reduced aerosol emissions come from in the SSP3-lowNTCF experiments)- what are the benefits of reducing both GHG+NTCF emissions instead of reduced NTCF emissions alone? Without context, the results presented here make me think 'Air pollution reductions are harmful- they will worsen climate extremes'- Is this the takeaway the authors intend?

IPCC AR6 Chapter 6.7 ends with some discussion of the contextualization of these idealized results in comparison with SSP1 (see for example, Shindell and Smith, 2019, Nature, or how Allen et al. also contextualizes the changes in extremes with the reduced air pollution exposure, etc.).

**Response:** Thanks for your constructive suggestions! As suggested, we discussed the realism of SSP3-lowNTCF scenario and added some qualitative comparison with SSP1 in the revised paper as follows:

[revised manuscript text omitted]

3. Clarity of figures:

In several figures showing time series, there are no y axis labels or x axis labels (e.g., Figure 4, others). Also, there are acronyms in the figure subplot titles that are not defined in the figure caption- it is difficult to find what these are without searching through the text/tables, so please define all acronyms in the caption. I understand the extremes are defined in a table, but it would be helpful to not have to flip back and forth to determine what they are.

**Response:** In the revised paper, we added axis labels and defined all acronyms in the captions.

In many of the maps showing changes, the same colormap is used to show changes over time and differences among experiments in the same figure, which is visually confusing.

**Response:** In the revised paper, we used different colormap, makes it look more visually appealing in terms of magnitude.

Several figures showing time series include red and green lines, which will be indistinguishable for a red/green colorblind reader.

**Response:** As suggested, the green lines were replaced with gray lines in Figure 1, Figure 4 and Figure 7 in the revised paper.

Several of the figures with maps show dots/stippling at locations where >60% of models 'agree on sign of change'. I don't find stippling 'significant' locations in this case to be particularly helpful visually because the dots cover almost all of the map (yes, CMIP6 models show the globe warms under SSP3- why is this stippled when this is my default expectation, and has been reported before?). Perhaps more importantly, the stippling obscures the colors underneath so the reader cannot easily interpret the colors. Can the authors stipple throughout where there is disagreement?

**Response:** As suggested, the dotted regions indicate that the warming is not significant in Figure 3 in the revised manuscript.

[Figure]

Showing that CMIP6 models agree on sign of change for global warming is a frankly low bar/not a robust metric, as is the low >60% agreement threshold. Can the authors choose a more robust method (e.g., agree on magnitude of change, using something like coefficient of variation- e.g., Buzan and Huber, 2020: https://www.annualreviews.org/doi/abs/10.1146/annurev-earth-053018-060100), or at least where 75% or 90% or 95% of the models agree on sign?

**Response:** As suggested, we used a higher threshold (six out of seven models for SAT, four out of five models for the extreme temperature indices and five out of six models for the extreme precipitation indices) to estimate whether the changes of climate extremes are robust. All related figures were updated in the revised paper.

CMIP6 model validation in terms of climate extremes: I was surprised to see an evaluation of CMIP6 model results without any (as far as I could tell) mention/citation of an assessment of model performance- how well do these models simulate temperature + precipitation extremes if we are to rely of their projections of future extremes?

**Response:** As suggested, we added new Figure 2 to evaluate the performance of the CMIP6 MME in simulating temperature and precipitation extremes in the historical period (1995-2014). The following information are added in the revised paper:

"To evaluate the performance of the models, a gridded daily maximum and minimum temperature and daily precipitation dataset obtained from the National Oceanic and Atmospheric Administration (NOAA) Climate Prediction Center (CPC) is used here. This dataset was constructed using optimal interpolation methods based on approximately 16,000 station and satellite observations (Chen et al., 2020b). It spans the period from 1979 to the present and has a high level of resolution of $0.5° \times 0.5°$. All model outputs as well as observations were interpolated into a common grid ($1° \times 1°$) through bilinear interpolation except precipitation data, which used first-order conservative interpolation." Lines 148-153

"We compared the simulated results with the observational climate extremes during 1995–2014 (Fig. 2). In general, the CMIP6 MME can reasonably reproduce the observed spatial distribution of

extreme temperature and precipitation indices. For the extreme temperature indices, the maximums obtained from both the CMIP6 MME and observations are found in eastern China and southern Asia, especially for the simulated absolute extreme indices (TXx, TR) (Figs. 2a, 2b, 2e and 2f), which are generally consistent with the observations in spatial distribution with limited difference in magnitude. Relative to the absolute extreme indices, the percentile and duration indices show large differences between the CMIP6 MME and observation (Figs. 2c, 2d, 2g and 2h). Previous studies also shown that both CMIP5 and CMIP6 perform relatively unsatisfactorily in simulating spatial patterns of the duration and percentile indices (Fan et al., 2020; Guo et al., 2021). For R10, RX5day and R95p, the climatological mean is well captured by CMIP6 MME, although it tends to produce overestimates especially over southeastern Qinghai-Tibet Plateau and the Indo-China Peninsula (Figs. 2i-n). In addition, the CMIP6 MME underestimates the CDD in northwest China and along Mongolia (Figs. 2o and 2p), which is consistent with previous studies (Zhu et al., 2021; Kim et al., 2020). Although the CMIP6 MME produce some regional biases with respect to observation, such biases will be significantly reduced when considering the difference between the two segments of time (Sillmann et al., 2013; Chen et al., 2020). In this study, we focused on the changes in the future (2031–2050) relative to the reference period (1995–2014), so the results of the CMIP6 MME can be considered representative."    Lines 155-169

[Figure]

**Figure 2**: The annual mean of the hottest day (TXx), warm days (TX90p), tropical nights (TR), warm spell duration (WSDI), heavy precipitation (R10), maximum consecutive 5-day precipitation (RX5day), total wet-day precipitation (R95p), and consecutive dry days (CDD) over study area during 1995-2014 for CMIP6 multi-model mean (left column) and gridded observations (right column).

CMIP6 multi-model ensemble averaging: From my reading of the paper, the authors seem to have just averaged across an ensemble of CMIP6 models- should the models be selected/weighted according to performance, or perhaps according to independence (see for example Brunner et al., 2020: https://esd.copernicus.org/articles/11/995/2020/ who suggest that equally weighting CMIP6 models does not produce the same results as weighting them based on independence/performance)? Or are there too few models? Some mention/justification of simple ensemble averaging could be helpful.

**Response:** Thanks for your suggestion.

(i) In the revised paper, we clarified as follows:

"At present, there are few models carrying out both SSP3-7.0 and SSP3-7.0-lowNTCF experiments, thus, we directly used the CMIP6 multi-model ensemble (MME) mean to investigate the changes of climate extremes in response to future SLCFs emission reductions." Lines 145-147

(ii) In the revised paper, we added new Figure 2 to evaluate the performance of the CMIP6 MME and found that the CMIP6 MME can well capture the spatial pattern of temperature and precipitation extremes in the historical period (1995-2014). For detailed descriptions, please refer to the above response.

The authors have an ensemble of model simulations, so they should have a range of results in terms of changes in climate extremes, but the main results presented in the text and Abstract are presented as one number (e.g. 'regional average temperature on the hottest days (TXx) by 0.3 K)'- isn't there some range/spread in the results? This range is shown in the figures and mentioned occasionally in the text, but this nuance/uncertainty does not come out in the Abstract or in much of the results, where one mean/median number, with no range/uncertainty is presented. It would seem to me that the range or 1 or 2 sigma should be presented next to all of the mean/median results.

**Response:** As suggested, we added one sigma range to all of the mean results in the Abstract and Conclusions in the revised paper. For example,

"The additional warming caused by the non-methane SLCF reductions increases the hottest days (TXx) by $0.3 \pm 0.1$ K, the percentage of warm days (TX90p) by $4.8 \pm 2.2$ %, the number of tropical nights (TR) by $1.7 \pm 0.8$ days………." Lines 16-25

Line/Specific Comments:

Lines 11-15: 'Stringent…climate'- the authors mention that SLCF emissions reductions have been implemented, then in the next sentence state that they examine future impacts. The naïve reader could assume that the SSP3-lowNTCF simulations are realistic and are simply a continuation of past air quality/climate policy. Is this the case?

Also, this sentence conveys to me the idea that emissions reductions have already happened, but the next sentence claims to study future emissions reductions. I think some clarification could be helpful here.

**Response:** In the revised paper, we modified this sentence as follows:

"Future stringent SLCF emissions controls to mitigate air pollution will substantially impact regional climate change." Lines 9-10

Line 13: 'in Asia'- the boxes in the main text seem to be a sub-section of eastern/southern Asia, not all of Asia.

**Response:** Thanks for your suggestion. We have changed the title into "Impacts of reductions in non-methane short-lived climate forcers on future climate extremes and the resulting population exposure risks in eastern and southern Asia". In the revised paper, we defined the latitude and longitude of our chosen study area in Figure 2 as follows:

"The scope of eastern and southern Asia in this study is defined as 0-60 °N, 70-150 °E." Line 115

Line 14: 'SSP' – please define this acronym (authors define all other acronyms in Abstract, why not this one?).

**Response:** As suggested, we defined "SSP" as "shared socioeconomic pathway (SSP)" in the revised paper. Line 12

Lines 16-18: 'The MME results…': The authors report a change in ERF that seems to be half the magnitude of that presented in the Allen et al. paper cited in the Introduction (0.44 W/m2 over the entire time period in Allen et al.). Why the difference? And also, most of the paper is not about ERF, but climate extremes, so why is this part of the focus in the Abstract? Or is this necessary to explain why changes are observed? I don't understand why the authors need to re-report changes in ERF that were recently published unless they show something significantly new/different.

**Response:** Our ERF estimates of 0.23 was smaller than 0.44 in Allen et al. (2020), which is likely attributed two aspects: (i) The units are different in two studies. Allen et al. (2020) represents trend change with the unit of $Wm^{-2}$ $decade^{-1}$, but this study represents mean change of two periods with the unit of $Wm^{-2}$. (ii) The number of climate models are different in two studies. The ERF trend of $0.44$ $Wm^{-2}$ $decade^{-1}$ in Allen et al. (2020) is the average result of all models, including both the five $Aer+O_3$ models and the four AER-only models. But in this study, the results are averaged from the seven $Aer+O_3$ models. SLCFs affect climate by perturbing the balance of radiative energy balance at the top of the Earth's atmosphere, and ERF is a useful measure of the extent to which forcing elements affect climate.

Line 16: 'Regional average temperature'- please remind reader which region (transitioned from global ERF in previous sentence)

**Response:** As suggested, we defined "regional mean" as the entire study area in the revised paper. Line 17

Line 24: 'predicted'- here and elsewhere in the manuscript, isn't 'projected' the preferred term for changes in climate based on changes in boundary conditions/forcing?

**Response:** As suggested, we modified "predicted" as "projected" in the revised paper. Line 23, Line 256, Line 266 and Line 405

Lines 58-59: 'on future climate change has been limited to the effect of aerosol forcing associated with incomplete interactive tropospheric chemistry schemes in global climate models' - except in Allen et al., who used CMIP6 ESMs, like in this study.

**Response:** As suggested, we added some other studies for this sentence in the revised paper. Lines 60-63

Lines 74-75: 'focused on trends of climate variables, and its assessment of regional climate changes, particularly climate extremes, was insufficient.' – this seems subjective- the extreme indices they assessed showed qualitatively similar results to the ones presented here. The time series presented in Allen et al. also show similar information, so what makes this 'insufficient'?

**Response:** As suggested, we clarified our motivation in the revised paper as follows:

"However, there were some limitations in that study. First, Allen et al. (2020) only considered three extreme indicators including hottest day, wettest day, and consecutive dry days to examine the effects of future non-methane SLCF reductions on climate extremes. These three indicators were not enough to represent climate extremes, especially the lack of some indicators related to human health. For example, tropical night (TR) usually occurs in combination with extended periods of heat (particularly in extra-tropical regions) and have been suggested to be problematic for human health (Weisskopf et al., 2002; Patz et al., 2005) and the maximum consecutive 5-day precipitation (RX5day) can be used as an indicator of flooding and related hazards (Frich et al., 2002; Sillmann et al., 2013). Second, climate extremes pose a serious threat on human body (Bras et al., 2021; Tellman et al., 2021). Quantifying avoided population exposure to climate extremes associated with future non-methane SLCF reductions is valuable for future policymaking on climate change mitigation and adaptation, especially in these densely populated and industrially developed regions of Asia, which is lacked in Allen et al. (2020)." Lines 84-93

"Compared with previous assessment by Allen et al. (2020), our study provides some new insights for the effects of future non-methane SLCF emissions on regional climate change. Firstly, although extreme temperature indices are all increasing in the future due to the reduction of non-methane SLCFs, TX90p and WSDI vary spatially opposite to TXx, indicating that the warming of future

temperature extremes is greater at higher latitudes, while the increase in the frequency and duration of extreme temperature occurrences is more pronounced at lower latitudes. As for extreme precipitation, changes in both R10 and R95p in some areas are contrary to previous results considering only aerosol reduction, revealing the importance of considering aerosol and ozone interactions. More importantly, we analyze the changes in TR and RX5day. The former represents the variation of nighttime temperature extremes that are important for human health. The latter is usually used as an indicator of flooding, suggesting that heavy precipitation associated with natural disasters will be aggravated in the future due to non-methane SLCFs reduction. Secondly, population exposure can provide a well assessment of future climate change risk. The reduction of non-methane SLCFs will result in the exposure of millions of people to extreme events, and up to tens of millions in densely populated areas, such as NIN, which is an indicator of human health risk and also valuable for future policymaking on climate change mitigation and adaptation. Thirdly, Allen et al. (2020) used nine models, including five Aer+$O_3$ models and four Aer-only models, but we used seven Aer+$O_3$ models. The more Aer+$O_3$ models may better reflect the effect of considering the combined aerosol and ozone changes simultaneously. Finally, Pendergrass et al. (2019) have shown that the response of extreme precipitation to warming varies widely in climate models, especially in the tropics. The rate of response increases with warming is not linear but non-linear (Pendergrass et al., 2019), as shown in Allen et al. (2020) that some of the extreme indices were not well fitted. Freychet et al. (2019) suggested that radiation-driven aerosol emission impacts on local surface temperature and precipitation were not linear and could be mitigated or cancelled by the local dynamics. Our method of subtracting the mean between two periods may, to some extent, provide a more intuitive representation of the changes in the extreme indices in absolute terms." Line 427-447

Line 76: Yes, the IPCC AR6 does present mostly a global overview in terms of temperature impacts (figure 6.23 in AR6 WG1), but Allen et al. already report that impacts are most intense in parts of Asia.

**Response:** As suggested, we clarified our motivation in the revised paper. Please see the response to the above comment.

Lines 137-139: 'Gridded population datasets for 2000 and 2040 under SSP3 were used to represent the population during the reference and future periods, respectively.' – which dataset? GPWv4 and SSP projections? Citation?

**Response:** As suggested, we added citation and clarified as follows:

"Gridded population datasets for 2000 and 2040 under SSP3 were used to represent the population during the reference and future periods, respectively (Jones and Oneill 2016)." Lines 178-180

Figure 2 and other figures with maps: see main comment about stippling significance, which covers information/colors presented on maps. Also, please use separate colormap to show differences in time periods vs differences among experiments to visually distinguish, unless there is a specific reason to use the same colormap in both.

**Response:** In the revised paper, we used different colormap, makes it look more visually appealing in terms of magnitude.

Lines 145-149: I am more familiar with attempting to distinguish among causes of impacts (changing climate or changing population) by assessing changes in population-weighted impacts using static (present) population, and comparing to results using dynamic population. Does this produce the same result as the equation/method here?

**Response:** The changes of future population exposure were attributed to the changes in population, climate, and population-climate interaction, which is well illustrated in Equation 1. We have revised the description. The difference of results using present and dynamic population only represents the contribution of population change to changes of population exposure.

Line 180 '3.2 Changes in temperature extremes in Asia' – I was a bit surprised not to notice a mention of how well these models simulate observed frequency etc of past, observed extremes. The authors jump straight into projected extremes- are there papers showing that these temperature and precipitation extreme indices are well captured by the climate models used here? See for example (Li et al.: https://journals.ametsoc.org/view/journals/clim/34/9/JCLI-D-19-1013.1.xml, Kim et al.,

2020: https://www.sciencedirect.com/science/article/pii/S2212094719302439; Yang et al., 2021: https://link.springer.com/article/10.1007/s00376-021-0351-4).

Related to this, Yang et al. (above) report that not all models perform equally well in terms of simulation of temperature extremes over China, and choose to analyze projections from a sub-selection of models that better simulate observed climate- have the authors considered doing this?

**Response:** At present, there are few models carrying out both SSP3-7.0 and SSP3-7.0-lowNTCF experiments, thus we directly used the CMIP6 multi-model ensemble (MME) mean to investigate the changes of climate extremes in response to future SLCFs emission reductions." Lines 145-147

In the revised paper, we added new Figure 2 to evaluate the performance of the CMIP6 MME and found that the CMIP6 MME can well capture the spatial pattern of temperature and precipitation extremes in the historical period (1995-2014). For detailed descriptions, please refer to the above response.

Figure 4: where is the region that the average encompasses? I don't think the region boxes are shown until Figure 6. How are these sub-regional boxes chosen? Are these SREX regions? Or did the authors just choose boxes to maximize signal after they ran the analysis? Also, the regional maps (like Figure 5) could be easier to interpret if the authors include country borders.

**Response:** (i) In the revised paper, we defined our study area in Figure 2. The regional mean time series in the entire study area were shown in Figure 4.

ii) We selected sub-regional boxes with the large signal, high emission, and population density.

iii) Accepted, we have added country borders in all regional maps.

Figure 7: Here and previous figure showing time series: please do not include red and green for red-green colorblind readers (why not red, blue, grey?).

**Response:** As suggested, the green lines were replaced with gray lines in Figure 1, Figure 4 and Figure 7 in the revised paper.

Line 245: 'the extreme precipitation changes are more significant' – how is significance determined? Do the authors mean 'larger' or something similar?

**Response:** Yes, the description "significant" were revised as "large" in the revised paper. Line 295

Line 250: Here and Figure 8: drawing boxes outlining the regions of interest on the anomaly maps would be helpful to determine where spatially the location is that the authors are discussing in the text.

**Response:** As suggested, we added the boxes in Figure 8 to represent the four selected sub-regions.

Line 293: 'future precipitation distribution in SC could be more heterogeneous under high SSP scenarios.' Spatial distribution or PDF distribution? Please be more specific, as I wasn't sure what a 'heterogeneous distribution is' if it's a histogram/PDF.

**Response:** It means the PDF distribution. A decrease in light rainfall and increase in heavy rainfall will lead to simultaneous increases in CDD and extreme precipitation, and this phenomenon may be even more pronounced under high SSP scenarios.

Figures 11/12: Typo in figure 'climte'

**Response:** Corrected as suggested.

Lines 329-333: 'climate factors under both the SSP3-7.0 and SSP3-7.0-lowNTCF scenarios in the four selected regions (Fig. 11). This result suggests that climate change is the primary driver of population exposure to extreme temperature events, followed by changes in the climate population interaction factor, with population change contributing the least.'

I am a bit confused about the wording here- how are we to distinguish SLCF from climate? I think a clarifying phrase and breaking up some long sentences could be helpful as authors explain.

**Response:** As suggested, we modified this sentence as "This result suggests that climate change caused by non-methane SLCFs is the primary driver of population exposure to extreme temperature events." Lines 385-386

Lines 346-347: 'in increases in extreme temperature and precipitation events' – this is relative to the background increases already experienced in SSP3-7.0, correct? Please clarify.

**Response:** Yes, we modified as this sentence as "…… resulting in increases in extreme temperature and precipitation events compared to the standard SSP3-7.0 scenario." Lines 402-403

Lines 368-378: See my main comment above: can the authors benchmark/qualitatively compare previously published findings of reductions of both GHG+NTCF under SSP1-2.6 or a similar experiment? What are the benefits of reducing both vs just reducing NTCFs, or the combined impacts? And I think a further discussion/emphasis of the realism of NTCF reductions only would be helpful- are countries going to only reduce SLCFs? Have they already? How realistic are these results in a real-world context?

**Response**:In the revised paper, we discussed as follows:

[revised manuscript text omitted]

---

## Author Comment (AC2)

**Response to Referee #2:**

We are grateful to the referees for their time and energy in providing helpful comments and guidance that have improved the manuscript. In this document, we describe how we have addressed the reviewer's comments. Referee comments are shown in black and author responses are shown in blue text.

Review of "Impacts of reductions in non-methane short-lived climate forcers on future climate extremes and the resulting population exposure risks in Asia" by Li et al.

This study uses AerChemMIP simulations to examine the effects of improved air quality through pollutant emissions reductions on projected climate extremes and associated population exposure in south and east Asia. A significant accelerated warming effect is found highlighting the importance of these short-lived forcings in policy-making and planning for future extremes.

The study will make a useful contribution to the literature and this is a very important area for analysis. I do have some significant concerns that I would ask the authors to consider.

- Thank you for your positive evaluations. All the questions and concerns have been carefully answered and the paper has been revised accordingly.

1. My first major concern is a technical one. The data are regridded using bilinear interpolation onto a 1-degree grid (L128). This is a much higher resolution than all but one of the models' native resolutions (Table 1). By interpolating to a higher resolution (in effect extrapolating) additional synthetic information is being added unintentionally and this could have a substantial effect on the extremes analysis in particular. My suggestion is to interpolate onto a common grid that is coarser (perhaps 2-degrees) and to use a different interpolation method for precipitation. I would recommend having a look at this webpage for useful discussion: https://climatedataguide.ucar.edu/climate-data-tools-and-analysis/regridding-overview.

**Response:** Thanks for your constructive suggestion!

(i) As suggested, we applied new interpolation method (first-order conservative) to interpolate precipitation data and updated the results for precipitation extremes (Figures 2, 7, 8, 9 and 12). The following information were added in the revised paper:

"All model outputs as well as observations were interpolated into a common grid ($1° \times 1°$) through bilinear interpolation except for precipitation data, which used first-order conservative interpolation." Lines 153-154

(ii) We selected hottest days (TXx) and heavy precipitation days (R10) as examples to compare the results between low ($2° \times 2°$) and high ($1° \times 1°$) resolutions. It is seen that there are limited differences of both spatial pattern and magnitude for temperature (Figure R1) and precipitation (Figure R2) extremes between low- and high-resolution results. Because our study focused on regional assessment, we choose to use the original results at high resolution ($1° \times 1°$) in the revised paper.

[Figure]

**Figure R1** Spatial distribution of TXx at different resolutions. (top: $2° \times 2°$ grid, bottom: $1° \times 1°$ grid).

[Figure]

**Figure R2** Spatial distribution of R10 at different resolutions. (top: 2°×2° grid, bottom: 1°×1° grid).

2. My other main concern is that there is no attempt at model evaluation apparent. I understand this may be challenging but given the analysis I would suggest some evaluation of extremes against an observational dataset over the recent period would be useful to benchmark whether the models are performing well enough.

**Response:** As suggested, we added new Figure 2 to evaluate the performance of the CMIP6 MME in simulating temperature and precipitation extremes in the historical period (1995-2014). The following information are added in the revised paper:

"To evaluate the performance of the models, a gridded daily maximum and minimum temperature and daily precipitation dataset obtained from the National Oceanic and Atmospheric Administration (NOAA) Climate Prediction Center (CPC) is used here. This dataset was constructed using optimal interpolation methods based on approximately 16,000 station and satellite observations (Chen et al., 2020b). It spans the period from 1979 to the present and has a high level of resolution of 0.5°×0.5°. All model outputs as well as observations were interpolated into a common grid (1°×1°) through bilinear interpolation except precipitation data, which used first-order conservative interpolation."
Lines 148-153

"We compared the simulated results with the observational climate extremes during 1995–2014 (Fig2). In general, the CMIP6 MME can reasonably reproduce the observed spatial distribution of extreme temperature and precipitation indices. For the extreme temperature indices, the maximums obtained from both the CMIP6 MME and observations are found in eastern China and southern Asia, especially for the simulated absolute extreme indices (TXx, TR) (Fig. 2a and b, Fig. 2e and f), which are generally consistent with the observations in spatial distribution with limited difference in magnitude. Relative to the absolute extreme indices, the percentile and duration indices show large differences between the CMIP6 MME and observation (Fig. 2c and d, Fig. 2g and h). Previous studies also shown that both CMIP5 and CMIP6 perform relatively unsatisfactorily in simulating spatial patterns of the duration and percentile indices (Fan et al., 2020; Guo et al., 2021). For R10, RX5day and R95p, the climatological mean is well captured by CMIP6 MME, although it tends to produce overestimates especially over southeastern Qinghai-Tibet Plateau and the Indo-China Peninsula (Fig. 2i-n). In addition, the CMIP6 MME underestimates the CDD in northwest China and along Mongolia (Fig. 2o and p), which is consistent with previous studies (Zhu et al., 2021; Kim et al., 2020). Although the CMIP6 MME produce some regional biases with respect to observation, such biases will be significantly reduced when considering the difference between the two segments of time (Sillmann et al., 2013b; Chen et al., 2020a). In this study, we focused on the changes in the future (2031–2050) relative to the reference period (1995–2014), so the results of the CMIP6 MME can be considered representative." Lines 155-169

[Figure]

**Figure 2**: The annual mean of the hottest day (TXx), warm days (TX90p), tropical nights (TR), warm spell duration (WSDI), heavy precipitation (R10), maximum consecutive 5-day precipitation (RX5day), total wet-day precipitation (R95p), and consecutive dry days (CDD) over study area during 1995-2014 for CMIP6 multi-model mean (left column) and gridded observations (right column).

3. There is other relevant literature on the role of aerosols in influencing the climate of this region (Freychet et al. 2019), including on accelerated warming and associated extremes (King et al. 2018; You et al. 2020).

**Response:** As suggested, the above three papers were discussed and cited in the revised paper.

"North-East of India witnessed cooler maximum temperatures due to increased aerosols (Freychet et al. 2019)." Lines 45-46.

"In eastern Asia, under global warming 1.5 ℃ and 2 ℃, China is expected to grow at a faster rate than the global mean, and there is a strong warming in the Tibetan Plateau and when studying changes in local climate between 1.5 ℃ and 2 ℃ of global warming, non-GHGs influences need to be considered (King et al. 2018; You et al. 2020). The effect of projected reductions in anthropogenic aerosol emissions over eastern Asia caused an increase in summer temperatures and raised the likelihood of extreme hot summers (King et al. 2018)." Lines 69-73.

4. Section 3.2. Some comparison of the changes in temperature extremes over Asia relative to other parts of the world would also be useful in reinforcing your point about the role of short-lived forcers in affecting local climate extremes.

**Response:** In the revised paper, we clarified as follows:

"In the SSP3-7.0 and SSP3-7.0-lowNTCF scenarios, the warming in most regions exceeds 1.5 K, and the warming is greater at higher latitudes under both scenarios, but the magnitude of the increase is larger under the SSP3-7.0-lowNTCF scenario than the SSP3-7.0 scenario. Such strong local effects of short-lived forcers to temperature extremes were also revealed in other high emission and population density regions (Sillmann et al., 2013a; Samset et al., 2018; Luo et al., 2020). The greatest changes in TXx, exceeding 5K, were simulated in RCP8.5 in such regions as South and North America, Eastern Europe, north-central Eurasia as well as Australia by the end of the 21st century (Sillmann et al., 2013a)." Lines 236-242

"For populated regions such as Europe, the United States and East Aisa, the TXx change in response to remove short-lived aerosol reductions is on average 25% stronger than global land-area mean (Samset et al., 2018). " Lines 246-248

5. Table 1: "America" should be the "United States" and "England" should be "United Kingdom".

**Response:** Corrected as suggested.

6. All map figures: Stippling where 60% of models agree (I'm assuming you mean five out of seven?) is quite a weak threshold for agreement that could be met by chance quite often. It might be more useful to just say the fraction of models and use a higher threshold (e.g. six out of seven).

**Response:** As suggested, we used a higher threshold (six out of seven models for SAT, four out of five models for the extreme temperature indices and five out of six models for the extreme precipitation indices) to estimate whether the changes of climate extremes are robust. Meanwhile, the inappropriate description "e.g., 60% of models" has been modified as "e.g., six out of seven models" in all figure captions. For example:

Figure 5: Spatial patterns of changes in the hottest day (TXx), warm days (TX90p), tropical nights (TR), and warm spell duration (WSDI)during 2031-2050 in Asia under the SSP3-7.0 (left column) and SSP3-7.0-lowNTCF (middle column) scenarios relative to 1995-2014. The right column represents changes caused by the non-methane SLCFs mitigation. The dotted regions indicate that at least four out of five models agree on the sign. Lines 796-799

**Reference:**

Chen, H., Sun, J., Lin, W., and Xu, H.: Comparison of CMIP6 and CMIP5 models in simulating climate extremes, Sci. Bull., 65, 1415-1418, https://doi.org/10.1016/j.scib.2020.05.015, 2020a.

Chen, H., Sun, J., and Li, H.: Increased population exposure to precipitation extremes under future warmer climates, Environ. Res. Lett., 15, https://doi.org/10.1088/1748-9326/ab751f, 2020b.

Fan, X., Miao, C., Duan, Q., Shen, C., and Wu, Y.: The Performance of CMIP6 Versus CMIP5 in Simulating Temperature Extremes Over the Global Land Surface, J. Geophys. Res. Atmos., 125, 1-16, https://doi.org/10.1029/2020JD033031, 2020.

Freychet, N., Tett, S. F. B., Bollasina, M., Wang, K. C., and Hegerl, G. C.: The Local Aerosol Emission Effect on Surface Shortwave Radiation and Temperatures, J. Adv. Model. Earth Syst., 11, 806-817, https://doi.org/https://doi.org/10.1029/2018MS001530, 2019.

Guo, D., Zhang, Y., Gao, X., Pepin, N., and Sun, J.: Evaluation and ensemble projection of extreme high and low temperature events in China from four dynamical downscaling simulations, Int. J. Climatol., 41, E1252-E1269, https://doi.org/10.1002/joc.6765, 2021.

Kim, Y. H., Min, S. K., Zhang, X., Sillmann, J., and Sandstad, M.: Evaluation of the CMIP6 multi-model ensemble for climate extreme indices, Weather Clim. Extrem., 29, 100269, https://doi.org/10.1016/j.wace.2020.100269, 2020.

King, A. D., Knutti, R., Uhe, P., Mitchell, D. M., Lewis, S. C., Arblaster, J. M., and Freychet, N.: On the Linearity of Local and Regional Temperature Changes from 1.5°C to 2°C of Global Warming, J. Clim., 31, 7495–7514, https://doi.org/10.1175/JCLI-D-17-0649.1, 2018.

Sillmann, J., Kharin, V. V, Zwiers, F. W., Zhang, X., and Bronaugh, D.: Climate extremes indices in the CMIP5 multimodel ensemble: Part 2. Future climate projections, J. Geophys. Res. Atmos., 118, 2473–2493, https://doi.org/https://doi.org/10.1002/jgrd.50188, 2013a.

Sillmann, J., Kharin, V. V., Zhang, X., Zwiers, F. W., and Bronaugh, D.: Climate extremes indices in the CMIP5 multimodel ensemble: Part 1. Model evaluation in the present climate, J. Geophys. Res. Atmos., 118, 1716-1733, https://doi.org/10.1002/jgrd.50203, 2013b.

You, Q., Wu, F., Shen, L., Pepin, N., Jiang, Z., and Kang, S.: Tibetan Plateau amplification of climate extremes under global warming of 1.5 ° C, 2 ° C and 3 ° C, Glob. Planet. Change, 192, 103261, https://doi.org/https://doi.org/10.1016/j.gloplacha.2020.103261, 2020.

Zhu, X., Lee, S. Y., Wen, X., Ji, Z., Lin, L., Wei, Z., Zheng, Z., Xu, D., and Dong, W.: Extreme climate changes over three major river basins in China as seen in CMIP5 and CMIP6, Clim. Dyn., 57, 1187-1205, https://doi.org/10.1007/s00382-021-05767-z, 2021.

---

## Author Response (AR2)

Response to Referee #1:

We are very grateful to the referee again in providing helpful comments and guidance that have improved the manuscript. In this document, we describe how we have addressed the reviewer's comments. Referee comments are shown in black and author responses are shown in blue text.

Overview:

The authors have mostly addressed the concerns I raised in the previous review, with a few exceptions. My major concerns are outlined below, followed by line comments.

Response: Thank you very much for your positive evaluations. All the questions and concerns have been carefully answered and the paper has been revised accordingly.

Major concerns:

1) Consistency of communication of findings:

In my initial review, I had suggested the authors stipple (add dots to) regions where the models disagree, but the authors only made this change to one figure. The other figures have stippling where the models agree- these dots cover the colors/shading in the locations where the changes are agreed upon by the models, effectively visually masking the relevant information. I suggest consistently using stippling throughout (in other words: stipple on all maps where models disagree, not where they agree on some maps and where they disagree on other maps- this can be confusing to the reader if the stippling changes from figure to figure).

Response: As suggested, we have stippled on all maps where models disagree in the revised manuscript.

The authors added country borders/boundaries to some of the maps, but not all of the maps- why did the authors only include this information on some of the maps and not others? I find the political boundaries/borders helpful for identifying which regions are impacted in which country when the

authors discuss the findings in the text.

Response: As suggested, we added country borders/boundaries to all the maps in the revised manuscript.

The authors are still using a very low significance criteria for stippling on the maps- most of the models must 'agree on sign of change' as the significance test. As I stated in my last review, I think in addition to agreeing on sign of change, some sort of signal: noise (or coefficient of variation or similar) would be a more robust/effective metric, particularly in the maps showing warming in ssp370 vs ssp370lowntcf- most of the globe warms almost everywhere over land in both experiments, so showing where the models all agree it warms as an agreement metric isn't particularly useful in my opinion. Agreeing on magnitude of changes would be more robust and meaningful.

Response: Thanks for your suggestion. Agreeing on sign of change was widely used in multi-model significance testing (Samset et al., 2018; Sillmann et al., 2019; Li et al., 2021). Also, as suggested, we added the information of signal-to-noise ratio, defined as the ratio of the absolute value of the multi-model ensemble (MME) mean to the deviation among the models, in the Supplements and corresponding description in the text to further describe the confidence level of the multi-model results. In addition, we also implemented statistical testing on the MME result (figures not shown). The regions with significance at ≥95 % confidence level from the t-test were broadly in line with those with signal-to-noise ratio larger than 1. So we only reserved the results of the signal-to-noise ratio in the revised manuscript.

2) Significance and uncertainty of findings, especially as they relate to precipitation changes.

In the text, I had suggested in my previous review that the authors present not only the multi-model mean/median when reporting findings, but also the spread or +/- 1 or 2 sigma. I appreciate the authors added this information in the Abstract and in some locations in the text, but they left out the uncertainty/model spread in many locations in the text. I suggest showing this uncertainty/spread throughout, both for consistency and transparency in terms of model agreement.

Response: As suggested, we added the corresponding uncertainty/spread in the text in the revised manuscript.

The authors have chosen to report several findings related to precipitation changes in the text with minimal discussion/mention (mentioned maybe once in the Discussion?) of the fact that the maps indicate most of the models don't even agree on the sign of the change (let alone the magnitude). Several of the precipitation results are reported in the text that seem to not be stippled in the maps, suggesting to me that the models don't agree, and I question how much we should trust the results. If the models don't agree on the sign of change in a region, I strongly suggest the authors mention this along with the results to warn the reader that the results should be interpreted with caution.

Response: As suggested, we mentioned that the sign of results shows poor agreement among models for the projection of extreme precipitation, especially in sub-regions in the revised manuscript as follows:

"Under the SSP3-7.0 scenario, heavy precipitation days decreases by approximately 2 days in central India, the Indo-China Peninsula, and by more than 4 days in the southeastern Qinghai-Tibet Plateau, SWC, and parts of Indonesia, while it increases in all other regions of the entire study area (Fig. 10a). However, it is worth noting that the consistency among models is poor." Lines 307-310

"In addition, increases of more than 3.0 days in heavy precipitation days are also found in India, the Indo-China Peninsula, and Indonesia, but again, consistency in these regions among models is poor, and this pattern is exactly opposite of that of GHG forcing (Fig. 10a and c)." Lines 314-316.

"In addition, non-methane SLCF reductions cause an increase of more than 2 mm in the maximum consecutive 5-day precipitation in the southeastern Qinghai-Tibet Plateau, the western Hengduan Mountains, SWC and the middle and lower reaches of the Yangtze River, whereas decreases of more than 3 mm occur in the southwestern part of the Indo-China Peninsula. However, there are large differences among models (Fig. 10f, Fig. S5f)." Lines 325-328

"Additional increases in total wet-day precipitation caused by reduced emissions of non-methane SLCFs are apparent in the southeastern Qinghai-Tibet Plateau, SWC, and the middle and lower reaches of the Yangtze River (>40 mm), while decreases occur in the southwestern part of the Indo-China Peninsula (<50 mm) with lower consistency among models." Lines 336-339

"However, consecutive dry days increases in NIN, the southeastern Tibetan Plateau, and the southern Yangtze River (Fig. 10j and k) with low inter-model agreement. Overall, consecutive dry days decreases by $0.02 \pm 0.8$ days and $0.4 \pm 0.5$ days under the SSP3-7.0 and SSP3-7.0-lowNTCF scenarios, respectively, with SNR less than 1, which means model projections have a low level of confidence. Future reductions in non-methane SLCFs contribute to decreases in consecutive dry days in northwestern and northeastern China, near the Hengduan Mountains, and the Indo-China Peninsula. Consecutive dry days decreases by $1.4 \pm 1.2$ days in SWC due to increase in the frequency and intensity of heavy precipitation (Fig. 11e). The sign of the results shows relatively good agreement and the SNR is larger than 1 among models in SWC and NC (Fig. 10i and Fig. S5i). Similarly, the changes in consecutive dry days due to non-methane SLCF reductions in some regions are comparable to the impacts of GHG forcing. Although for most regions, the models do not all agree on the sign of the responses, however, some robust features are still apparent." Lines 344-353

Related to this, the authors use wording such as 'significant increases are found' - If the authors are going to suggest that the results are 'significant', what is the significance test? If no test is used, can the authors use a test, such as a Mann-Whitney/Rank Sum test to determine if the medians/distributions are statistically distinguishable?

Response: Thanks for your constructive suggestion! We removed or replaced the word 'significant' and modified these sentences in the revised manuscript.

3) Proofreading/changing text after authors made changes:

There are several instances where the authors made changes to figures, but the figure numbers did not change (for example, there are two Figure 2s). Also, colors in line plots are changed from green to grey, but all captions still reference green lines that no longer exist. Please double check all of this, and that updated figure numbers in text match figure numbers in captions after fixing figure numbering.

Response: Thanks for your suggestion, we checked and updated the figure numbers in the revised manuscript.

Line comments:

Line 17: 'the entire study area'- suggest 'entire study area of Asia' or similar more specific wording to remind reader

Response: As suggested, we modified this sentence as "In terms of the entire study area of Asia". Line 17

Line 21: 'For temperature extremes…' To prepare reader for sub-regional reporting, I suggest starting with wording such as 'In terms of sub-regional changes in temperatures extremes, the largest increases in...'

Response: As suggested, we modified this sentence as "In terms of sub-regional changes in temperatures extremes…". Line 21

Line 45: 'North-East of India' - suggest 'North eastern' or 'The North East of'

Response: As suggested, the description "North-East of India" was revised as "North eastern". Line 44

Line 63: 'using the five Earth system models (ESMs)' - which five? (author use of the word 'the' seems to imply these were introduced previously?)- suggest removing 'the' unless authors have previously introduced these models

Response: As suggested, we removed "the" in this sentence. Line 61

Line 69: 'global warming 1.5 °C and 2 °C,' – suggest 'global warming of 1.5 °C and 2 °C'

Response: As suggested, we modified this sentence as "In eastern Asia, under global warming of 1.5 °C and 2 °C…". Line 68

Line 70: 'Plateau and when' – there are two complete phrases run together with 'and'- please separate the phrases into different sentences.

Response: As suggested, we separated the phrases into different sentences as follows:

"In eastern Asia, under global warming of 1.5 °C and 2 °C, China is expected to grow at a faster rate than the global mean, and there is a strong warming in the Tibetan Plateau (You et al. 2020). When

studying changes in local climate between 1.5 °C and 2 °C of global warming, non-GHGs influences need to be considered (King et al. 2018; You et al. 2020)." Lines 68-71

Line 90: 'on human body' - suggest 'to human health and well-being' or similar wording

Response: As suggested, we replaced "on human body" with "to human health and well-being". Line 89

Line 93: 'is lacked' – suggest 'is lacking'

Response: As suggested, we replaced "lacked" with "lacking". Line 92

Lines 76-93: thanks to the authors for adding in more detail about previous work, how this work represents a significant advance- I was much more convinced of the novelty/importance of this work after reading this paragraph

Response: Thanks for your positive evaluation.

Line 95: 'increasing' – suggest 'will increase' or better 'will increasingly become a threat to...'

Response: As suggested, we replaced "increasing" with "will increasingly become a threat to…". Line 94

Line 107: 'have become more' - can authors be more specific here about what is better in these ESMs that make it worthwhile to do the analysis? they suggest/hint at this earlier but would be helpful to be more explicit here.

Response: Thanks for your suggestion, we clarified as follows:

"With continuous development, some models have evolved from the original climate system model to more complete ESMs that considers the coupling of Earth system processes (including atmosphere, ocean, land, sea and ice, and biogeochemical cycles) and can provide a more realistic and complete analysis of the whole Earth-atmosphere system." Lines 105-107

Line 113: 'in eastern and southern Asia using' – suggest adding in: 'Asia (here defined as 0-60°N, 70-150°E.) using' and then eliminating the sentence at the end of the paragraph because the last

sentence doesn't seem to fit, and this information can be included in this sentence here.

Response: As suggested, we removed the last sentence and modified this sentence as follows:

"…in eastern and southern Asia (here defined as 0-60°N, 70-150°E) using…" Line 112

Lines 146-147: 'thus, we directly used the CMIP6 multi-model ensemble (MME) mean' - please specify that the authors are referring to the sub-selection of models here so the reader is not confused thinking authors are using the 'entire' ensemble of ~40-50 CMIP6 models

Response: Thanks for your suggestion, we have mentioned that there are seven models carrying out both SSP3-7.0 and SSP3-7.0-lowNTCF experiments in Lines 138-139. Here we emphasized once again and modified this sentence as follows:

"At present, there are few models carrying out both SSP3-7.0 and SSP3-7.0-lowNTCF experiments as mentioned above, thus, we directly used the seven-model ensemble mean…"    Lines 145-146

Line 151: 'high level of resolution' - this is subjective- I suggest removing and simply describing the resolution.

Response: As suggested, we modified this sentence as follows:

"It spans the period from 1979 to the present with a resolution of 0.5° × 0.5°." Line 151

Line 182: 'were are' – typographic error/fix wording

Response: Corrected. Line 179

Lines 209-210: 'Regionally, the greatest warming of more than 1.5 K is found in central and northern Asia and in northern North America, particularly in the Arctic, where warming is greater than 2.5 K.' - might be helpful to cite a paper here that has shown this previously as I'm sure Arctic amplification of warming is not a new finding here

Response: Thanks for your suggestion, we discussed and cited papers here.

"Previous studies have also shown that the Arctic warms at a much higher rate than the rest of the globe in recent decades (Screen and Simmonds, 2010; Pithan and Mauritsen, 2014)" Line 212-214

Lines 216-217: 'The reduction of non-methane SLCF emissions results in an average increase of 0.19

K in global mean SAT in the MME results, ranging from 0.06 K to 0.29 K across different models (Table 3).' – thanks to authors for including the spread in model results to give the reader a sense of the uncertainty- please do so elsewhere where reporting mean model results.

Response: Thanks for your positive evaluation, we added uncertainty/spread to all multi-model mean in the revised manuscript.

Lines 227-233: Where is the model spread/range? Only one number is reported ("27.7%"- this makes the answer sound so certain, but I'm sure there's disagreement in the models in terms of magnitude, correct?)

Response: As suggested, we added uncertainty/spread as follows:

"By 2050, the hottest day and warm days increase by $2.2 \pm 0.7$ K and $27.7 \pm 13.1$ % under the SSP3-7.0 scenario and by $2.6 \pm 0.6$ K and $34.9 \pm 13.3$ % under the SSP3-7.0-lowNTCF scenario, respectively. Compared to the hottest day and warm days, larger increases occur in the coldest day and warm nights, which increase by $2.8 \pm 0.8$ K and $35.9 \pm 14.9$ % under the SSP3-7.0 scenario and $3.2 \pm 1.0$ K and $43.1 \pm 15.6$ % under the SSP3-7.0-lowNTCF scenario, respectively." Lines 230-233

Lines 239-247: In my opinion, this description of previous work and comparison with results from other regions outside the study region seems to fit more in the Introduction or Discussion

Response: Thanks for your suggestion, we compared with results from other regions in the Discussion as follows:

"Such strong local effects of SLCFs on temperature extremes were also revealed in other high emission and population density regions such as Europe and the United States (Sillmann et al., 2013a; Samset et al., 2018; Luo et al., 2020)." Line 415-417

Lines 256-257: 'Concomitant with the increases in TNn and TN90p, TR is also projected to increase over the entire study area in the future.' As far as I can determine from looking at the maps, this is not true- there are blue areas on the map, indicating a decrease in days with tropical nights on the Tibetan Plateau, unless I am misinterpreting the map and colorbar- either way, TR does not consistently increase over the entire study area.

Response: Thanks for your suggestion, we clarified this sentence as follows:

"Concomitant with the increases in the coldest day and warm night, tropical night is also projected to increase over the most entire study area in the future." Lines 256-257

Lines 281-284: please include the model spread here, not just the mean

Response: As suggested, we added model spread as follows:

"The domain-averaged values of the hottest day, warm days, tropical night, and warm spell duration across the entire study area have increases of 1.4 ± 0.4 K, 37.4 ± 11.3 %, 8.9 ± 2.3 days, and 9.1 ± 2.0 days under the SSP3-7.0 scenario, respectively (Fig. 8). Considering the effects of non-methane SLCF reductions, these extreme temperature indices show increases of 1.7 ± 0.4 K, 42.2 ± 11.0 %, 10.6 ± 2.8 days, and 10.1 ± 1.7 days under the SSP3-7.0-lowNTCF scenario, respectively. Future reductions in non-methane SLCFs result in additional increases of 0.3 ± 0.1 K in the hottest day, 4.8 ± 2.2 % in warm days, 1.7 ± 0.8 days in tropical night, and 1.0 ± 0.4 days in warm spell duration for the entire study area. The hottest day, warm days, and warm spell duration all show their largest regional increases in NIN and NC, reaching 0.5 ± 0.2 K, 8.5 ± 2.9 %, and 2.8 ± 1.1 days and 0.4 ± 0.2 K, 5.9 ± 2.7 %, and 3.0 ± 1.3 days, respectively. The models agree on the sign of the change in tropical night across regions, and the largest increases occur in NC and the SCB, at 5.1 ± 2.5 and 4.9 ± 3.3 days, respectively" Line 282-290

Lines 284-285: 'In general, the SCB is most strongly affected by extreme temperatures under both scenarios' - based on which metric? looking forward two sentences, the authors indicate that the other regions are most strongly affected. This wording seems to directly contradict what the authors say later in the paragraph.

Response: Thanks for your suggestion, we removed this sentence in the revised manuscript.

Lines 293-294: 'Figure 7 shows the time series of changes in annual mean precipitation indices averaged across the entire study area under the SSP3-7.0 and SSP3-7.0-lowNTCF scenarios from 2015 to 2050 relative to the reference period' - can authors please show range of difference in gray shading around the grey line? Or does this make the figure too complicated?

Response: Thanks for your suggestion, we omitted the grey lines in Figure 1, Figure 6 and Figure 9 in order not to complicate the figures.

Lines 296-300- here and elsewhere, please include range in model results, not just the mean/median

Response: As suggested, we added model spread as follows:

"By 2050, the values of heavy precipitation days, the maximum consecutive 5-day precipitation, and total wet-day precipitation have increased by $1.6 \pm 0.6$ days, $2.7 \pm 1.0$ days, and $46.7 \pm 16.4$ mm under the SSP3-7.0 scenario, respectively. The increases in precipitation decrease drought events, resulting in a reduction in maximum consecutive dry days of $2.4 \pm 1.5$ days" Lines 298-301

Lines 304-306: 'Under the SSP3-7.0 scenario, R10 decreases by approximately 2 days in central India, the Indo-China Peninsula, and by more than 4 days in the southeastern Qinghai-Tibet Plateau, SWC, and parts of Indonesia, while it increases in all other regions of the entire study area (Fig. 8a).'
- do the models even agree on the sign of change in all these locations? How confident are the authors in projected changes in local precipitation if the models don't even agree on the sign of change?

Response: Thanks for your suggestion, we clarified as follows:

"…However, it is worth noting that the consistency among models is poor." Lines 309-310

Lines 310-311: 'significant'- how is this significant? Based on what metric? Most of the boxplots in Figure 9b show nearly complete overlap in the 25th-75th percentile estimates.
If the authors are going to suggest that the results are 'significantly different' can they at least use a significance test, such as a Mann-Whitney test to determine if the medians/distributions are statistically distinguishable?

Response: Thanks for your suggestion, we removed the word 'significant'.

Line 320: 'significant increase'- see earlier comments on significance

Response: Thanks for your suggestion, we removed the word 'significant'.

Lines 331-334: again, how is significance determined?

Response: Thanks for your suggestion, we removed the word 'significant'.

Line 337: 'However, CDD increases in NIN, the southeastern Tibetan Plateau, and the southern Yangtze River (Fig. 8a and b).' – do the models even agree on the sign of change across all of these

locations, let alone the magnitude? I don't see stippling on the maps in many of these areas.

Response: Thanks for your suggestion, we modified this sentence as follows:

"However, consecutive dry days increases in NIN, the southeastern Tibetan Plateau, and the southern Yangtze River (Fig. 10j and k) with low inter-model agreement" Line 344-345

Lines 338-341: 'Future reductions…'- as far as I can determine, this is the first time that sign agreement has really been acknowledged for precipitation changes- I think the significance and sign agreement needs to be better addressed elsewhere, not just here.

Response: Accepted. We calculated the signal-to-noise ratios (please see the supplement) and added the description of the signal-to-noise ratios and sign agreement of the multi-model ensemble results to elaborate the confidence of the results in the text. Please see the revised manuscript.

Lines 343-344: 'Notably, the increases in CDD in India and eastern China are accompanied by increases in the frequency and intensity of extreme precipitation, which may be related to the probability distribution of future precipitation.' – see previous comments about model agreement on sign of change, significance considerations.

Response: Thanks for your suggestion, we added a description of the sign consistency of the results in the revised manuscript, please see previous response.

Lines 349-351: 'In general, along with the increase in average temperature, future reductions in the emissions of non-methane SLCFs will increase the intensity and frequency of extreme precipitation and decrease the occurrence of extreme droughts in the entire study area.' This and other similar statements seem to be worded as incredibly certain, but don't seem to be backed up by clear evidence from my reading of the figures. For example, when I look at this figure, it looks like this is somewhat true in southern China for R10 and RX5day, and R95p, but not true for CDD, which shows mixed results, and very little model agreement on sign of change.

Response: Thanks for your suggestion, we removed the relevant description and added some descriptions of the sign consistency and the signal-to-noise ratios of the results in the revised manuscript.

Lines 355-356: 'along with a decrease in CDD in northern China' - I see some light blue in Figure 8 with some stippling in isolated areas in eastern China, but a lot of China shows no stippling, which gives me little confidence in the sign, let alone magnitude, of these projections, unless I am misinterpreting the results, if so, please explain.

Response: Thanks for your suggestion, we removed the relevant description

Lines 357-359: 'Consequently, our results indicate that future non-methane SLCF reductions may alleviate the observed precipitation anomaly pattern of "southern flood/northern drought" over eastern China, but may not have apparent mitigating effects on the precipitation reduction in India.' – I am not familiar with the southern flood/northern drought research, but when I look at this figure, I see a mostly east/west, not north/south pattern here with little sign agreement, so I don't know what evidence this statement is based on.

Response: Thanks for your suggestion, we removed the relevant description

Lines 390-391: 'The climate factor is also the largest contributor to the increase in population exposure, followed by the climate- population interaction factor and the population factor in NC, the SCB, and SC' - please be more specific- to precipitation extremes?

Response: Thanks for your suggestion, we clarified as follows:

"For population exposure under extreme precipitation, the climate factor is also the largest contributor to the increase in population exposure, followed by the climate-population interaction factor and the population factor in NC, the SCB, and SC (Fig. 14b-d)." Line 390-392

Lines 401-403: 'Our results show that non-methane SLCF reductions will exacerbate the warming effect caused by GHGs, resulting in increases in extreme temperature and precipitation events compared to the standard SSP3-7.0 scenario.' - if reduced alone, correct?

Response: Yes, we modified this sentence as follows:

"Our results show that non-methane SLCF reductions alone will exacerbate the warming effect caused by GHGs…" Lines 402-403

Lines 420-421: 'Notably, large differences in extreme precipitation changes are found at the regional scale in response to reductions of non-methane SLCFs and only aerosols, with changes in opposite

directions observed in some regions' – yes, I appreciate the authors noted this, and I think this should be noted when individual results are presented if the models don't consistently even agree on sign of change (in the Results section).

Response: Thanks for your suggestion, we noted this in the Results section as follows:

"For heavy precipitation days, the maximum consecutive 5-day precipitation, and total wet-day precipitation, there is a robust increase in the southeastern Qinghai-Tibetan Plateau, SWC and the middle and lower reaches of the Yangtze River, with relatively good inter-model agreement and high confidence. In SWC, these extreme precipitation indices increase by $2.5 \pm 1.9$ days, $2.5 \pm 1.5$ mm, and $37.5 \pm 22.6$ mm, and these increases are greater or comparable to the effects of GHGs. By contrast, consecutive dry days decreases by $1.4 \pm 1.2$ days in SWC due to increases in the frequency and intensity of heavy precipitation events. Notably, there are still large uncertainties in the model's simulations of extreme precipitation, which may attribute to model resolution and natural variation (Li et al., 2014; Deser et al., 2012)." Lines 421-427

"Similarly, the inter-model variation in population exposure is large, which arises from differences in model simulations of extreme precipitation." Lines 433-434

Line 431: 'contrary'- opposite in sign?

Response: Yes, and we replaced "contrary" with "opposite in sign". Line 440

Lines 433-435: 'The latter is usually used as an indicator of flooding, suggesting that heavy precipitation associated with natural disasters will be aggravated in the future due to non-methane SLCFs reduction.' True in southern/SE China, but really not much agreement on sign anywhere else if I am interpreting the maps correctly.

Response: Thanks for your suggestion, we clarified as follows:

"The latter is usually used as an indicator of flooding, suggesting that heavy precipitation associated with natural disasters will be aggravated in the future due to non-methane SLCFs reduction in certain areas, especially in SC and SWC." Lines 443-445

Line 435: what is 'well assessment'?

Response: We modified this sentence as follows:

"Secondly, population exposure is an effective indicator for climate change risk assessment." Lines 445-446

Lines 436-437: 'The reduction of non-methane SLCFs will result in the exposure of millions of people to extreme events, and up to tens of millions in densely populated areas' – This sentence gives me the impression that these extreme events wouldn't have happened in the SSP3-7.0 scenario without NTCF reductions. However, my understanding looking at the SSP3-7.0 vs lowNTCF figures is that under SSP3-7.0 extremes also occur, but the elimination of NTCF just makes them more frequent/likely/worse. Is this correct? If so, can the authors more carefully word these summary findings please?

Response: Thanks for your suggestion, we clarified as follows:

"Compared to SSP3-7.0 scenario, the reduction of non-methane SLCFs will result in additional exposure of millions of people to extreme events, and up to tens of millions in densely populated areas, such as northern India…" Line 446-448

Line 443: 'that some of the extreme indices were not well fitted' – what does this mean?

Response: The linear trend of some indices is not well fitted in Allen et al. (2020). For example, less robust results are generally observed when it comes to wettest day and CDD, with $R^2$ of 0.485 and 0.049. The linear trends for CDD even lack significance.

Lines 452-455: thanks to the authors for contextualizing this.

Response: Thanks for your positive evaluation.

Lines 458-461- this description seems more accurate to me than what is stated on lines 436-437 (see comment above)

Response: We clarified the sentence on lines 446-448, please see the above response.

Line 486: 'has'- have

Response: Corrected.

Lines 493-494: 'which may lead to underestimation of the impact of SLCF emissions reductions in China' - wouldn't this lead to an overestimation if simulated SLCF emissions are higher than they actually are in reality (so future emissions reductions would be less impactful because they already happened, so the models would overestimate the real impacts of SLCF reductions bc we've already experienced these impacts?)? Or am I misinterpreting the impact of lower emissions? Or what the authors intend to say?

Response: What we intended to say is that the CMIP6 database underestimate emission reductions obtained from China's Action Plan in the base year, and the emission bias in the base year would pass to the future and lead to different emission mitigation pathways (Tong et al., 2020). Therefore, future emissions reductions in China are likely to be more intense than those in CMIP6, leading to more impactful to climate change in China compared to present day. We clarified this sentence as follows: "For example, the SSP-RCP global emissions scenarios used in CMIP6 do not fully consider the rapid pollution controls enacted in China since 2013 under the Air Pollution Prevention and Control Action Plan. Consequently, CMIP6 database underestimate emission reductions obtained from China's Action Plan, which may lead to underestimation of the impact of SLCF emissions reductions in China (Wang et al., 2021; Tong et al., 2020), …" Lines 499-502

Lines 496-499: 'What policymakers, the public, or the media need to know'- I will leave this wording choice up to the authors, but in my previous review I was not intending that the authors needed to explicitly address the conclusion to policymakers and the media- instead, I was suggesting the authors contextualize their results as they have done in the Discussion/Conclusion so as not to be misinterpreted by science reporters etc. In my opinion, the authors can eliminate the phrase 'What policymakers, the public or the media need to know is that' and start the sentence with 'Air pollution…'

Response: As suggested, we modified this sentence as follows:

"Air pollution is dangerous to human health, and there is no doubt that we need clean air, but more importantly efforts to reduce GHGs need to be doubled in order to simultaneously mitigate climate change and improve air quality (Quaas et al., 2022; McKenna et al., 2021)." Lines 506-508

Suggestions for figures:

For figures with maps:

In my previous review, I suggested the authors use one colormap for showing changes through time, and another colormap for showing differences among experiments so they can be quickly/easily visually distinguished. The authors combined color bars for the climate change maps, but kept the same color maps (the same colors) to show the differences among experiments- I suggest using different color maps (a different divergent color map for showing differences vs changes through time) so the reader can easily determine that the maps are not showing the same things.

Response: As suggested, we changed the colormap for showing differences among experiments, please see figures in the revised manuscript.

For all line graphs: authors reference 'green' lines in figures, but lines are now grey- please double check figure captions and change where appropriate.

Response: We omitted the grey line in Figure 1, Figure 6 and Figure 9.

Line 775/776 (Figure 1 caption): may be helpful to add a line something like 'if red line only is shown (g,h), this indicates that ssp370 and ssp370-lowNTCF overlap" or other wording to let reader know why there is only one line in bottom panels (g,h).

Response: As suggested, we added "only red line in (g, h) indicates that ssp370 and ssp370-lowNTCF overlap." in caption of Figure 1 in the revised manuscript. Lines 792-793

Figure 2 (line 780): figure panels are so small, I can't see text, etc.

Instead of this being a figure that is 2 (columns) x 8 (rows), can the authors make this two sets of double columns (4x4) so precip is next to temperature with a little space in the middle to separate them visually? This might help fill the white space on the page and improve readability bc the maps could be larger?

Response: We added a third column "CMIP6 minus observations" in response to Referee 2 and reformatted this figure.

Figure 2 (line 785): The stippling in the maps changes from figure to figure - please consistently use stippling- are significant or insignificant regions stippled? Also, the authors changed one figure in

response to my comments, but the same comment could be made for all figures showing warming-showing where models agree on warming in warming scenarios is a low bar that doesn't show meaningful information, and obscures the coloring underneath- this is the case in not only Figure 3, which the authors changed, but also Figure 2, (for ERF changes not warming), Figure 5, Figure 8, etc.

Response: Thanks for your suggestion, we stippled on all maps where models disagree in the revised manuscript.

Figure 3: what about showing the range of differences in grey shading around the grey line as well? Or does this make the plot too complicated to interpret? Also, as noted above, please change 'green lines' to 'grey lines' in caption here and elsewhere where necessary.

Response: We have omitted the grey line in Figure 1, Figure 6 and Figure 9 in order not to make the figures more complicated.

Figure 5: I suggest adding units on colorbar label, or add in units in the figure caption as in earlier figures- the labels in the upper right are hard to find/read

Response: As suggested, we moved the location of the units. Please see Figure 7, Figure 10, and Figure 12 in the revised manuscript.

Figures 6,9,10: Why are country/political boundaries only shown in Figure 5 and 8, and not in other figures with maps (such as maps in Figs 6,9,10)? Can the authors please be more consistent with showing these borders for reader orientation unless there is a specific reason to exclude them?

Response: As suggested, we added country borders/boundaries to all the maps in the revised manuscript.

Figure 6, 7 and other graphs with y axes that cross zero: here and in other figures, a horizontal reference line marking zero on the y axis can be helpful.

Response: As suggested, we added a horizontal reference line marking zero on the y axis in Figure 6, Figure 8, Figure 9 and Figure 11 in the revised manuscript.

Figure 11, Figure 12: legend has an error- 'climte'

Response: Corrected.

Reference:

Deser, C., Knutti, R., Solomon, S., and Phillips, A. S.: Communication of the role of natural variability in future North American climate, Nat. Clim. Chang., 2, 775–779, https://doi.org/10.1038/nclimate1562, 2012.

Li, C., Zwiers, F., Zhang, X., Li, G., Sun, Y., and Wehner, M.: Changes in annual extremes of daily temperature and precipitation in CMIP6 models, J. Clim., 34, 3441–3460, https://doi.org/10.1175/JCLI-D-19-1013.1, 2021.

Li, J., Yu, R., Yuan, W., Chen, H., Sun, W., and Zhang, Y.: Precipitation over East Asia simulated by NCAR CAM5 at different horizontal resolutions, J. Adv. Model. Earth Syst., 7, 774–790, https://doi.org/https://doi.org/10.1002/2014MS000414, 2015.

Pithan, F. and Mauritsen, T.: Arctic amplification dominated by temperature feedbacks in contemporary climate models, Nat. Geosci., 7, 181–184, https://doi.org/10.1038/ngeo2071, 2014.

Samset, B. H., Sand, M., Smith, C. J., Bauer, S. E., Forster, P. M., Fuglestvedt, J. S., Osprey, S., and Schleussner, C. F.: Climate Impacts From a Removal of Anthropogenic Aerosol Emissions, Geophys. Res. Lett., 45, 1020–1029, https://doi.org/10.1002/2017GL076079, 2018.

Screen, J. A. and Simmonds, I.: The central role of diminishing sea ice in recent Arctic temperature amplification, Nature, 464, 1334–1337, https://doi.org/10.1038/nature09051, 2010.

Sillmann, J., Stjern, C. W., Myhre, G., Samset, B. H., Hodnebrog, Ø., Andrews, T., Boucher, O., Faluvegi, G., Forster, P., Kasoar, M. R., Kharin, V. V., Kirkevåg, A., Lamarque, J. F., Olivié, D. J. L., Richardson, T. B., Shindell, D., Takemura, T., Voulgarakis, A., and Zwiers, F. W.: Extreme wet and dry conditions affected differently by greenhouse gases and aerosols, npj Clim. Atmos. Sci., 2, 1–7, https://doi.org/10.1038/s41612-019-0079-3, 2019.

Tong, D., Cheng, J., Liu, Y., Yu, S., Yan, L., Hong, C., Qin, Y., Zhao, H., Zheng, Y., Geng, G., Li, M., Liu, F., Zhang, Y., Zheng, B., Clarke, L., and Zhang, Q.: Dynamic projection of anthropogenic emissions in China: methodology and 2015-2050 emission pathways under a range of socio-economic, climate policy, and pollution control scenarios, Atmos. Chem. Phys., 20, 5729–5757, https://doi.org/10.5194/acp-20-5729-2020, 2020.

Response to Referee #2:

We are grateful to the referees for their time and energy in providing helpful comments and guidance that have improved the manuscript. In this document, we describe how we have addressed the reviewer's comments. Referee comments are shown in black and author responses are shown in blue text.

Li et al compare CMIP6 simulations with/without NTCFs and show that ERF increases due to NTCF reductions. They compare various climate extremes indices over eastern and southern Asia between the two scenarios. The authors have largely responded well to the previous reviewer comments. However, I have some additional points that I think should be considered prior to publication. Line numbers refer to the tracked revised manuscript.

Response: Thank you for your positive evaluations. All the questions and concerns have been carefully answered and the paper has been revised accordingly.

Major comments

1. Are the findings statistically significant? In some places the word 'significant' is used (e.g. 'A significant increase of more than 40 days occurs…', line 257), however no details of statistical testing are given. The standard deviations overlap in figures such as fig. 4, 7, and in fig 6 the boxplots overlap. For the map plots such as fig 5, 8, dots are used to indicate that a given number of models agree on the sign. It would be more meaningful if dots represented regions where the differences due to NTCFs are not statistically significant. Having models agree on the sign is not convincing that there is a significant difference between the two scenarios. Lastly, the uncertainty ranges quoted on lines 395-397 imply that the impact of NTCF reductions does not make a statistically significant impact. I strongly recommend implementing statistical testing on the results.

Response: Thanks for your suggestion,

i) We redescribed sentences with "significant" in the revised manuscript.

ii) We stippled on all maps where models disagree in the revised manuscript and added the signal-to-noise ratio in the Supplements to describe the confidence level of the multi-model results.

iii) We also implemented statistical testing on the MME result. Taking TX90p and R95p for example, we found that the regions with significance at ≥95 % confidence level from the t-test are broadly in line with those with signal-to-noise ratio more than 1 (Figure R1, Figure R2).

Therefore, we chose the signal-to-noise ratio to quantitatively characterize the confidence of the multi-model ensemble signal in the revised manuscript.

[Figure]

Figure R1 Spatial patterns of changes in warm days (TX90p) during 2031-2050 in Asia under the SSP3-7.0 (left column) and SSP3-7.0-lowNTCF (middle column) scenarios relative to 1995-2014. The right column represents changes caused by the non-methane SLCFs mitigation. The slipped regions in (a~c) and (d~f) indicate that the signal-to-noise ratio is less than 1 and significance at <95 % confidence level from the t-test, respectively.

[Figure]

Figure R2 Spatial patterns of changes in total wet-day precipitation (R95p) during 2031-2050 in Asia under the SSP3-7.0 (left column) and SSP3-7.0-lowNTCF (middle column) scenarios relative to 1995-2014. The right column represents changes caused by the non-methane SLCFs mitigation. The slipped regions in (a~c) and (d~f) indicate that the signal-to-noise ratio is less than 1 and significance at <95 % confidence level from the t-test, respectively.

2. I am uncomfortable with the framing that NTCF reductions lead to increased ERF; this is a blanket statement and not true of all possible NTCF mitigation scenarios. At worst, it could lead readers to believe that it is not worth pursuing NTCF emissions reductions because of the extra warming it would incur. As the authors are aware, not all NTCFs behave the same way; tropospheric ozone and black carbon both have warming effects. Indeed, mitigating these emissions has been recommended as a way to complement CO2 reduction measures and improve air quality (see e.g. the UNEP Integrated Assessment of Black Carbon and Tropospheric Ozone, 2011). Clearly it is reductions in non-BC aerosols that are driving the increase in ERF in the SSP3-7.0-lowNTCF scenario; this point needs to be made more clearly. For example, line 222: "the cooling effect of ozone reduction is significantly weaker than the warming effect of aerosol reduction." This is true for the specific scenario analysed here, but would not necessarily be true for alternative low-NTCF scenarios.

Similarly, line 273: "Overall, the combined reductions of aerosols, ozone, and their precursors have an additional warming effect on Earth's climate system, exacerbating the surface warming and extreme temperature events caused by GHGs." Ozone reductions cannot contribute to the additional warming; the additional warming in the low-NTCF scenario can only come from aerosol reductions. More care should be taken throughout the paper to caveat the results appropriately.

Response: Thanks for your suggestion,

i) We have mentioned in the discussion that the important impact of SLCFs on climate change in the short term does not mean that reducing air pollutants is harmful to the climate, but rather that it generates additional warming to the climate, amplifying the temperature and extreme precipitation caused by GHGs changes. Please see Lines 467-480 for more details.

ii) As suggested, we caveated the results more appropriately and clarified as follows:

"Although the reduction of SLCFs leads to a decrease in the aerosol optical depth (AOD) on a global scale (Fig. S1), not all SLCFs behave the same way and it is reductions in non-BC aerosols that are driving the increase in ERF in the SSP3-7.0-lowNTCF scenario." Lines 201-203

"In general, under the SSP3-7.0-lowNTCF scenario, the combined reductions of aerosols, ozone, and their precursors causes further warming, suggesting that the cooling effect of ozone reduction is somewhat weaker than the warming effect of aerosol reduction." Lines 223-225

"Overall, although ozone reductions cause cooling, the combined reductions of aerosols, ozone, and their precursors have an additional warming effect on Earth's climate system, exacerbating the surface warming and extreme temperature events caused by GHGs." Lines 273-275

3. Related to the above point, it would be useful to understand how NTCFs differ spatially over the study region during 2031-2050. Could maps of ground-level ozone concentration and aerosol optical depth be included?

Response: As suggested, we added maps of ground-level ozone concentration and aerosol optical depth in Figure S1 in Supplements.

4. A small number of models performed the SSP3-7.0 and SSP3-7.0-lowNTCF simulations, and the

authors analyse the multi-model mean. How skewed are the model responses? Would you see large differences in the results if you analyse the multi-model median? I am aware, for example, that the BCC-ESM1 model (one of the models analysed) contains large biases in aerosol optical depth relative to other models in certain regions, but do not know if these correlate with biases in the climate extremes examined over southeast Asia.

Response: Thanks for your suggestion,

i) We mentioned that there are large differences among models related to extreme precipitation changes in the revised manuscript.

ii) There are slight differences when analyzing the multi-model median and multi-model mean under the SSP3-7.0 and SSP3-7.0-lowNTCF scenarios, but such differences are small when it comes to the impact of non-methane SLCFs reductions. As shown in Figure 8 and Figure 11, the black center lines and dots almost overlap.

iii) We analyzed the different models of the spatial distribution of AOD changes due to emission reductions of non-methane SLCFs in eastern and southern Asia. In general, the results of the BCC-ESM1 model and other models were in relatively good agreement (Figure R3).

[Figure]

Figure R3 Spatial patterns of changes in ambient aerosol optical depth at 550 nm in Asia during 2031-2050 caused by non-methane SLCFs mitigation for different models.

Minor comments

Line 17: "in terms of the entire study area" note that the study area has not been defined by this point.

Response: Thanks for your suggestion, we modified this sentence as "In terms of the entire study

Line 122: this is unclear wording - why should a lack of climate policies related to GHG reductions and air pollution controls mean that SSP3-7.0 is used as a reference scenario? In reality, SSP3-7.0 and SSP3-7.0-lowNTCF are the only pair of simulations available through CMIP6 for which the effects of NTCFs can be assessed in a controlled manner, so I would simply state this.

Response: As suggested, we revised this sentence as follows:

"The SSP3-7.0 scenario includes no climate policies related to GHG reductions and weak air pollution controls ($\sim$7.0 W m$^{-2}$ at 2100). In this scenario, the future climate change is approximatively influenced by GHG forcing only. The SSP3-7.0-lowNTCF scenario uses the same GHG forcing path but with strong air pollution controls, and thus future climate change in this scenario is influenced by a combination of GHG forcing and reduction of non-methane SLCF emissions." Lines 121-125

Table 1: did all models provide fSST simulations for calculating ERF? And, did they provide the same number of fSST realisations as stated in Table 1?

Response: Yes, all models used in this study provide fSST simulations for calculating ERF and they provide the same number of fSST realizations. We mentioned this in the revised manuscript:

"All models used here provided SST simulations for calculating ERF and they provided the same number of realizations as stated in Table 1" Lines 141-143

Table 3: it would be useful to also report the SAT for the study area. Also note the discrepancy between the caption (SAT) and table header (Tas).

Response: Corrected.

The figure numbers and captions need to be updated; there are two figure 2's, for example. And the captions sometimes refer to green dashed lines, which are now grey (e.g. figure 1).

Response: Thanks for your suggestion, we checked and updated the figure numbers in the revised manuscript.

Figure 1: the grey dashed lines showing the difference between the two scenarios don't really make sense when reading the y axis labels. I got confused and started wondering how negative emissions are possible. I would be tempted to omit the line showing differences in this figure.

Response: As suggested, we omitted the grey line in Figure 1 in the revised manuscript.

Figure 2: great to see the evaluation work done as recommended by previous reviewers. Could I suggest adding a third column: CMIP6 minus observations?

Response: As suggested, we added a third column in the revised manuscript.

Figures 11 and 12 have spelling errors in the legend ('climte'); also please define the acronyms in the headers.

Response: Thanks for your suggestion,

i) Corrected. ii) We defined the acronyms in the headers, please see Figures 13 and 14 in the revised manuscript.

Line 168: representative of what?

Response: Thanks for your suggestion, we clarified this sentence as follows:

"In this study, we focused on the changes in the future (2031–2050) relative to the reference period (1995–2014), the results of CMIP6 MME for future extreme climate projections can be considered representative." Lines 168-169

Section 3.2 and 3.3: I couldn't review these sections properly as there were just too many acronyms that I am unfamiliar with (I am not an expert in climate extremes). Table 2 is a useful addition to the paper, but it is tiring to have to keep cross-referencing it. I suggest keeping table 2, but writing out the acronyms wherever possible so as not to interrupt the flow of text.

Response: As suggested, we write out the acronyms as much as possible in the text in the revised manuscript.

Line 331-332: you don't need the minus signs (e.g. 'significant decreases … <-50 mm')

Response: Corrected.

---

## Author Response (AR3)

Response to Editor #:

We are very grateful to the editor in providing helpful comments and guidance that have improved the manuscript. In this document, we describe how we have addressed the editor's comments. Comments are shown in black and author responses are shown in blue text.

1) Are these +/- 2 std dev?

Response: Thanks for your suggestion, the spreads in the text are +/- 1 std dev and we added some description and modified the description in Figs 6, 9, 13 and 14 in the revised manuscript.

"By 2050, the hottest day and warm days increase by 2.2 ± 0.7 K (mean ± standard deviation) and 27.7 ± 13.1 % under…" Lines 235-236

"The red and blue shading represents the ± 1 standard deviations across models." Line 823 and 838

"The error bars denote the ± 1 standard deviations across models." Line 859 and 865

2) The what in China? Average temperature?

Response: Yes, we modified this sentence as follows:

"In eastern Asia, under global warming of 1.5 °C and 2 °C, the average temperature in China is expected to grow at a faster rate than the global mean…" Lines 68-69

3) I believe this is just referring to the "Earth system" as the atmosphere is part of the system.

Response: Thanks for your suggestion, we deleted "-atmosphere" in this sentence. Line 107

4) Is this incomplete data to the to AerChemMip?

Response: Yes, and we mentioned it in the revised manuscript.

"Due to incomplete data provided by CESM2-WACCM and GISS-E2-1-G in the AerChemMIP…" Lines 172-173

5) Why were these time periods selected? Also, why just 19 years? I assume it is because of the quicker changes and response of climate to SLCP, but it could help to state it.

Response: Thanks for your suggestion, we mentioned it in the revised manuscript as follows:

"We focused on the changes in extreme climate indices in the future (2031–2050) relative to the reference

period (1995–2014), as this is when reductions in aerosol and ozone precursor emissions and the resulting changes in climate are expected to be significant." Lines 175-177

6) Figures S2 and S3 are not referred to in the text, but are key figures added to address the reviewer's comments. These need to be included in the discussion here.

Response: Thanks for your suggestion, we added some description in the revised manuscript as follows:

"It is worth noting that the global spatial distribution of ERFs shows poor inter-model agreement, with large areas where there is disagreement of the sign with at least one of the models and signal-to-noise ratios ($S_{NR}$) less than 1 (Supplementary methods and Fig. S2), especially over the ocean." Lines 199-201

"The largest increases of more than 1.5 W m$^{-2}$ are found over India, southeastern Asia, and surrounding oceans with relatively good inter-model agreement (Fig. 4c and S2c)" Lines 204-205

"The models agree on the sign of the change across regions and have high level of confidence (Fig. 5a, b and Fig. S3a, b)." Lines 216-217

"Moreover, such warming is exacerbated by future non-methane SLCF emissions reductions in most regions of the world, particularly in the Northern Hemisphere with great inter-model consistency (Fig. 5c and 3c)." Lines 219-220

7) I would recommend expanding this to say that there are large areas where there is disagreement of the sign with at least one of the models (identified by the hatched lines or lines etc.).

Response: As suggested, we expand this sentence as follows:

"It is worth noting that the global spatial distribution of ERFs shows poor inter-model agreement, with large areas where there is disagreement of the sign with at least one of the models and $S_{NR}$ less than 1 (Supplementary methods and Fig. S2), especially over the ocean." Lines 199-201

8)As this is part of methods, I recommend also referring to it first in the methods section.

Response: As suggested, we refer it in the methods section as follows:

"We also quantify the confidence of the MME relative to each individual model using sign agreement and Signal-to-noise ratio ($S_{NR}$). The details of the calculation of $S_{NR}$ can be found in Supplementary method." Lines 152-154

9)The SN<1 in TXx is lines over Mongolia, which is part of higher latitudes, so this statement confuses me some. For the models agreeing on sign, I think that is in Figure 7, correct? It should be referred to here.

Response: Thanks for your suggestion, what we referred to was that under SSP3-7.0 and SSP3-7.0-lowNTCF scenarios, all models agree on the sign and $S_{NR}$ larger than 1. We also refer to the figures in the revised manuscript as follows:

"…and the warming is greater at higher latitudes under both scenarios with all models agree on the sign and $S_{NR}$ larger than 1 (Fig 7a, b and Fig. S4a, b)." Line 246

10) In this sentence it can be indicated that there are still areas in the domain where there is difference in the signs of the models and S/N. The description of the uncertainties and differences in the models was a key corrections suggested by both reviewers.

Response: Thanks for your suggestion, we deleted this sentence in the revised manuscript. Line 246

11) But the 90th percentile is relative to each area, correct? So the 90th percentile would be higher for these regions already. So I am not yet following how just "higher temperatures" here than at higher latitudes would lead to more warm days. I think what this is saying is the distribution was tighter (e.g. lower daily variation as noted here), so smaller increases in temperature still put more days into warm days. Is that correct? Please clarify here.

Response: Yes, the 90th percentile calculation in this paper is obtained by calculating each grid points, lower daily temperature variation at low latitudes could result in more days above the 90th percentile of the reference period. We deleted "higher temperatures" in this sentence in the revised manuscript. Line 252

12) These are only shown in Figure 6 correct, and not over the domain? I can't see them at least. I recommend then including "...and warm nigh (Fig 6), tropical night is also projected to increase over much of the study area (red areas in Fig. 7g and h) in the future". Also check on wording as "the" and "entire" were not deleted in the track changes.

Response: As suggested, we modified these sentence as follows:

"Concomitant with the increases in the coldest day and warm night (Fig 6), tropical night is also projected

to increase over most study area (red areas in Fig. 7g and h) in the future…" Lines 260-261

13) However, much of the domain has S/N<1 and they don't agree on sign. This should be included close to this first statement.

Response: Thanks for your suggestion, we added some description in the revised manuscript.

"Concomitant with the increases in the coldest day and warm night (Fig 6), tropical night is also projected to increase over most study area (red areas in Fig. 7g and h) in the future with some difference among models, but the agreement among models is relatively good in regions where tropical night increases, such as eastern China, western northern China, southern Asia and southern eastern Asia." Lines 260-263

14) Not for all as there are blue areas in Fig 7i. Perhaps you can highlight where the magnitude is larger.

Response: Thanks for your suggestion, we highlight some regions with obvious changes in the revised manuscript.

"The spatial pattern of tropical night changes under SSP3-7.0-lowNTCF is similar to that under SSP3-7.0, but the magnitude of the changes is larger, especially in eastern China, western northern China, and Indo-China Peninsula where models show good agreement (Fig. 7g, h and Fig. S4g, h)." Lines 265-267

15) Figure S4l does seem to show that these areas have SN>1, but Fig 7l also shows that models don't agree about the sign is some of these areas. This uncertainty needs to be included in this discussion.

Response: Thanks for your suggestion, we added some uncertainty description in the revised manuscript as follows:

"Notably, the confidence level of the MME in these regions is relatively high, except for some small areas (Fig. S4l), but the variation in sign among models is large (Fig. 7l)." Lines 279-280

16) It will help to refer to the figures that make these points

Response: As suggested, we refer to the figures here. Line 285

17) This is labeled "Asia" correct?

Response: Yes, "the entire study area" in the text refer to "Asia".

18) In NTCF mitigation, correct?

Response: Yes. We modified this sentence as "The hottest day, warm days, and warm spell duration all show their largest regional increases in NIN and NC due to non-methane SLCFs reduction…" Lines 295-296

19) As noted by the reviewers, precipitation predictions have larger uncertainties and disagreement across models. I strongly recommend addressing this up at the front of this section so that results are put into context right away. Right now, this is only really noted in line 309. Providing this general context then puts in place the caveats and uncertainties of the results from the start. I am not recommending to delete the text lower down, just to add some here.

Response: Thanks for your suggestion, we added some description at the front of this section in the revised manuscript.

"However, precipitation projections have larger uncertainties and disagreement across models." Lines 306-307

20) see a larger inc in the mean, correct - as they overlap within 2 std dev.

Response: We modified this sentence as follows:

"Non-methane SLCFs mitigation causes an increase in extreme precipitation indices after 2035 except consecutive dry day, which shows a complex signal in the future." Lines 310-311

21) S5 should also be referenced here

Response: As suggested, we referred to S5 here as follows:

"However, it is worth noting that the consistency among models is poor (Fig. 10b and Fig. S5b)." Line 320

22) As you are referring to Fig 10 and 11 together, it can help readers to follow if the figs are referred to. So this statement is from Fig 11 b, but the next is Figure 10 c, right?

Response: Thanks for your suggestion, we referred to related figs to make it clearer.

"The reductions in non-methane SLCFs cause an increase of $1.0 \pm 0.5$ days in average heavy precipitation days across the entire study area (Fig. 11b). For the selected regions, robust increases in heavy precipitation days occur in SWC, SC, reaching $2.5 \pm 1.9$ days and $1.5 \pm 0.8$ days, respectively (Fig. 10c and Fig. 11b)." Lines 320-323

23) Fig. 11c

Response: done. Line 329

24) Fig 11e

Response: done. Line 355

25) Refer to the figure here.

Response: done.

"Overall, consecutive dry days decreases by $0.02 \pm 0.8$ days and $0.4 \pm 0.5$ days under the SSP3-7.0 and SSP3-7.0-lowNTCF scenarios, respectively (Fig. 11e), with $S_{NR}$ less than 1, which means model projections have a low level of confidence (Fig. S5j and k)." Lines 354-356

26) Neither of these regions have models agreeing with the sign, or with SN>1, so I can't see how they are referred to as "robust".

Response: Thanks for your suggestion, we modified this sentence as follows:

"Also, for most regions, the models do not all agree on the sign of the responses and have low level of confidence. The increases in consecutive dry days in India and eastern China are accompanied by increases…" Lines 361-363

27) I would recommend considering adding just a few sentences on how your results do highlight again the large uncertainties and model disagreements with simulating future precip. I think that using such indices shows really well how hard it is to try to plan for future impacts of changes in precipitation due to the differences in the model. I think that is shown well here and I would recommend highlighting it here and/or in discussion

Response: Thanks for your suggestion, we highlighted in discussion as follows:

"Notably, our results show a large uncertainty and model disagreement in simulating future extreme precipitation. Previous studies also indicate that there are still large uncertainties in the model's simulations of extreme precipitation, which may attribute to model resolution and natural variation (Li et al., 2014; Deser et al., 2012)." Lines 437-439

28) I recommend adding in here something about the acknowledging the model spread and uncertainty, but motivating for the analysis. It may even help to state clearly how these results should be interpreted/used in light of the uncertainties.

Response: As suggested, we added some description here in the revised manuscript.

"It is important to emphasize that there are still uncertainties among models, especially in future projections of extreme precipitation, the difference in model simulations could also lead to uncertainties in population exposure." Lines 375-377

29) Was the ensemble mean used as "c" in equation 1 to get this?

Response: Yes, the ensemble mean used as "c" in equation 1.

30) Fig 13

Response: We have not mentioned Figure 13 here, as we describe it as "more than". We clarified this sentence in the revised manuscript as follows. Also, as suggested, we referred to the relevant figure here in the revised manuscript.

"Compared to the SSP3-7.0 scenario, future non-methane SLCF reduction will increase the population exposed to extreme temperature by more than $3.5 \times 10^8$ person-days in NIN, followed by more than $1.5 \times 10^8$ person-days in NC and the SCB (Fig. 12c)." Lines 385-387

31) Is this the blue or pink bar in NTCF column in four panels in Fig 13?

Response: It is the blue bar in NTCF column. We modified this sentence as "Future non-methane SLCF reductions increase total population exposure to warm days by…" Lines 398-399

32) recommend this statement is re-written as the point is not yet clear. To me, I am not following the first half of the sentence "It is no difficult to find?" - does this mean it is not surprising that the

variation is large?

Response: Thanks for your suggestion, we modified this sentence as "Also, the inter-model variation is large, which arises from differences in model simulations of extreme precipitation." Lines 408-409

33) this is repetitive

Response: Thanks for your suggestion, we delete "but non-linear" in the revised manuscript. Line 465

34) There was a question on this by the reviewer and it is explained in the response but not yet added here.

Response: Thanks for your suggestion, we added some description in the revised manuscript.

"The rate of response increases with warming is not linear (Pendergrass et al., 2019), as shown in Allen et al. (2020) that some of the extreme indices were not well fitted such as the wettest day and CDD, with lower value of $R^2$ or even lack significance." Lines 464-466

35) these are the reductions already obtained, correct. So this statement is that the impact of emission reductions that have already happened may be underestimated, right? I recommend updating this statement to clarify that these are already realized reductions (as I believe they are).

Response: Thanks for your suggestion, we modified this sentence as follows:

"Consequently, CMIP6 database underestimate emission reductions obtained from China's Action Plan (Wang et al., 2021; Tong et al., 2020), which means that the impact of emission reductions that have already happened may be underestimated, thus may lead to underestimation of the impact of SLCF emissions reductions in China." Lines 513-516

36) This almost makes it sound like the blue line isn't there. Perhaps clearer to say "Only the red line is visibile in (g,h) as ssp......overlap"

Response: As suggested, we modified this sentence to make it clearer. Line 802

37) It may help to state that red indicates that model is over-predicting (is that correct? I think from the wording this is model minus obs). I assume the same convention was used for all the anomalies, and thus I think stating it once would be ok.

Response: As suggested, we added some description in the revise manuscript.

"The right column represents difference between CMIP6 and observations, so the red area in this column indicates that model is over-projecting." 806-807

38) Does this mean the areas with the lines? I am not used to seeing them called "slipped". Perhaps consider "areas with hatching" or "areas with lines".   This comment is for all figures with this in this manuscript and in supplementary materials

Response: Thanks for your suggestion, we replaced "slipped" with "areas with hatching" in the revised manuscript.

39) g and h are the only ones that have lines in the individual scenarios (i.e. a and b don't have them).

Response: Thanks for your suggestion, we checked these figs, it is because in Figures 7a, b, d, e and k all models agree on sign across the entire study area. Also, the change of WSDI under SSP-3.70 had lines in small part of northern Asia and northern India (Fig. 7j).

[Figure]

40) the numbers on person days for colour bar R95d is missing.

Response: Thanks for your suggestion, we corrected it in the revised manuscript

41) Comment on S/N method in supplementary material: I think stating that you did both 95th percentile confidence level with t-test and SN test here and they are the same so you are just showing one (as you explained well in author's response) would also strengthen this section. I recommend adding it in.

Response: Thanks for your suggestion, we added some description in the revised supplementary as follows.

"In addition, we also performed the test of confidence level on the MME result. The regions with

significance at ≥ 95 % confidence level from the t-test were broadly in line with those with signal-to-noise ratio larger than 1. So we only show the results of the signal-to-noise ratio here." Lines 20-22

42) Change word "slipped" in figure captions as I believe that is a typo.

Response: As suggested, we replaced "slipped" with "areas with hatching" in the revised manuscript.

43) Figure S4 g and h: This is the only one that shows the lines about S/N for the individual scenarios. Why is it not consistent with others (e.g. a, b)?

Response: Thanks for your suggestion, we checked these figs, it is because in Figures S4a, b, e ,j and k $S_{NR}$ > 1 across the entire study area. Also, the change of TX90p under SSP-3.70 had lines in small part of southern Asia (Fig. S4d).